# Functional ultrasound neuroimaging reveals mesoscopic organization of saccades in the lateral intraparietal area

Whitney S. Griggs [1,2,13] ✉, Sumner L. Norman [1], Mickael Tanter[3,4], Charles Liu [1,5,6,7], Vasileios Christopoulos[8,9], Mikhail G. Shapiro[10,11,12] & Richard A. Andersen[1,8] ✉

The lateral intraparietal cortex (LIP), contained within the posterior parietal cortex (PPC), is crucial for transforming spatial information into saccadic eye movements, yet its functional organization for movement direction remains unclear. Here, we used functional ultrasound imaging (fUSI), a technique with high sensitivity, large spatial coverage, and good spatial resolution, to map movement direction encoding across the PPC by recording local changes in cerebral blood volume within PPC as two male monkeys performed memory-guided saccades. Our analysis revealed a heterogeneous organization where small patches of neighboring LIP cortex encoded different directions. These subregions demonstrated consistent tuning across several months to years. A rough topography emerged where anterior LIP represented more contralateral downward movements and posterior LIP represented more contralateral upward movements. These results address two fundamental gaps in our understanding of LIP's functional organization: the neighborhood organization of patches and the stability of these populations across long periods of time. By tracking LIP populations over extended periods, we developed mesoscopic maps of direction specificity previously unattainable with fMRI or electrophysiology methods.

The posterior parietal cortex (PPC) integrates visual information with other sensory modalities, represents possible action plans, and decides upon the optimal action for downstream execution[1–3]. Separate PPC regions preferentially encode different movement types[4,5], or effectors. The lateral intraparietal area (LIP) preferentially encodes saccades[6], the parietal reach region (PRR) preferentially encodes limb reaches[2], and the anterior intraparietal area (AIP) preferentially encodes grasping movements[7]. These areas reveal an effector-dependent functional organization within PPC.

[1]Biology and Biological Engineering, California Institute of Technology, Pasadena, CA, USA. [2]Medical Scientist Training Program, David Geffen School of Medicine, University of California Los Angeles, Los Angeles, CA, USA. [3]Physics for Medicine Paris, INSERM, CNRS, ESPCI Paris, PSL Research University, Paris, France. [4]INSERM Technology Research Accelerator in Biomedical Ultrasound, Paris, France. [5]Department of Neurological Surgery, Keck School of Medicine of USC, Los Angeles, CA, USA. [6]USC Neurorestoration Center, Keck School of Medicine of USC, Los Angeles, CA, USA. [7]Rancho Los Amigos National Rehabilitation Center, Downey, CA, USA. [8]T&C Chen Brain-Machine Interface Center, California Institute of Technology, Pasadena, CA, USA. [9]Biomedical Engineering, University of Southern California, Los Angeles, CA, USA. [10]Chemistry & Chemical Engineering, California Institute of Technology, Pasadena, CA, USA. [11]Andrew and Peggy Cherng Department of Medical Engineering, California Institute of Technology, Pasadena, CA, USA. [12]Howard Hughes Medical Institute, Pasadena, CA, USA. [13]Present address: Department of Neurological Surgery, The University of Chicago Medicine, Chicago, IL 60605, USA. ✉e-mail: wsgriggs@gmail.com; richard.andersen@vis.caltech.edu

The mesoscopic (-100–1 mm scale;[8–11]) functional organization of saccade direction within LIP remains poorly explored[5,12], with most studies focusing on spatial receptive field topography[13–16] with differing results. The visual receptive fields and saccadic motor fields have been shown to be largely overlapping[17]. While Savaki et al. 2010[18] provided evidence for topographic organization for saccade direction, their single-axis saccade paradigm and quantitative [$^{14}$C]deoxyglucose method left unresolved how multiple saccade directions are represented in LIP. This debate[12] continues in part due to the limited field of view, sensitivity, and/or spatial resolution of existing recording techniques (Fig. 1A). fMRI can record whole brain activity, but lacks the spatial resolution and signal sensitivity to further refine our knowledge of LIP's spatial organization (Fig. 1B). Intracortical electrophysiology with high-density microelectrode arrays (16–400 µm inter-electrode spacing), such as Neuropixels or Utah arrays, can measure single neuron activity but cannot sufficiently sample or simultaneously record from large brain volumes, such as primate PPC (Fig. 1D). Furthermore, it is difficult to align and reconstruct data recorded over many months, limiting our ability to observe anatomical patterns across many recordings. Recent advances in distributed microelectrode arrays, such as the Gray Matter array[19] or MePhys system[20], now allow simultaneous multi-region recordings across primate hemispheres. While these distributed microelectrode array platforms achieve unprecedented temporal resolution (<1 ms) and volumetric coverage, the spacing between electrode shafts (2–5 mm) remains an order of magnitude coarser than the -100–500 µm functional organization observed in cortical columns and microcircuits[21]. This spacing fundamentally limits continuous spatial sampling of fine-grained population activity patterns. These trade-offs in existing recording techniques highlight the need for a sensitive technique to bridge the gap in spatial resolution between microscopic (e.g., single neurons) and macroscopic (e.g., whole brain) views of the primate cortex.

Here, we use an emerging technique, functional ultrasound imaging (fUSI), to determine the mesoscopic, i.e., between microscopic and macroscopic, spatial organization of saccadic response fields within LIP. fUSI's large field of view, excellent sensitivity, and high spatial resolution (Fig. 1A, C) are ideally suited to this task[22–25]. We recorded fUSI while two rhesus macaque monkeys (Monkey L and P) performed an oculomotor task. We found functionally distinct subregions within (dorsal-ventral) and across (anterior-posterior) coronal LIP planes where small mesoscopic patches of neighboring cortex

encoded different movement directions consistently across months to years. These results fill a gap in our understanding of LIP's functional organization and demonstrate that fUSI is a powerful tool for elucidating mesoscopic function in the brain.

## Results

Using fUSI, we recorded high-resolution changes in cerebral blood volume (CBV) from multiple PPC subregions as two rhesus macaque monkeys (Monkey L and P) performed memory-guided saccades (Fig. 2A). These areas included lateral intraparietal area (LIP), ventral intraparietal area (VIP), medial intraparietal area (MIP, a subregion of PRR), Area 5, Area 7, and medial parietal cortex (MP). During the task, each monkey fixated on a center cue, was cued with one of eight peripheral directions, remembered the cue location, and executed a saccade to the remembered location once the central fixation point extinguished.

We used a miniaturized linear ultrasound transducer array capable of high spatial resolution (100 µm × 100 µm in-plane) and a large field of view (12.8 mm width, 128 elements, 100 µm spatial pitch, 16 mm depth penetration, 400 µm plane thickness)[24,25], i.e., each voxel is -100 × 100 × 400 µm. We recorded 1 Hz fUS images by positioning the transducer surface normal to the brain above the dura mater. We recorded from multiple evenly-spaced coronal planes of the left PPC (Fig. 2B, C). During each session (Table S1), we recorded from a single coronal plane. We centered the recording chamber over the intraparietal sulcus to record from as much of the posterior parietal cortex as possible, both medial-lateral, but also anterior-posterior (Fig. 2B, and Supplementary Movie 1, 2).

### Are there mesoscopic populations tuned to different directions?
To identify mesoscopic directional tuning patterns across PPC, we used a general linear model (GLM) to identify voxels that responded differently to the eight directions. In both monkeys, mesoscopic PPC populations had clear directional tuning. In an example session from Monkey P (Fig. 3A-C), most voxels within the anatomically-defined LIP (>75% of voxels) showed directional tuning while regions outside of the LIP, such as VIP, MIP, Area, 5, Area 7, and MP, contained fewer voxels with directional tuning (<10% of voxels within a given region) (Figure S1). All region boundaries were estimated based upon a histological rhesus monkey atlas[26]. Within LIP, we observed heterogenous response patterns in different subpopulations, where we define

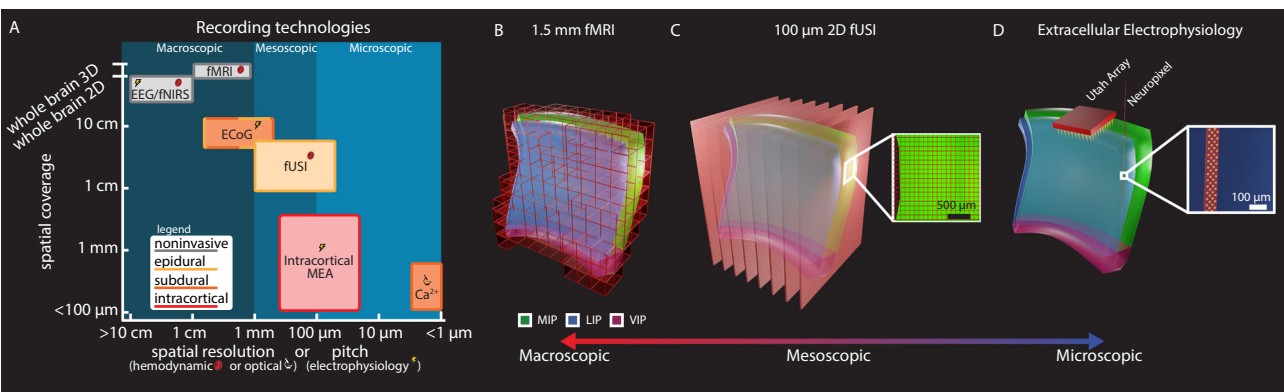

**Fig. 1 | Functional ultrasound enables mesoscopic imaging of neural populations. A** Spatial coverage, invasiveness, and spatial resolution for different large animal recording technologies. Spatial coverage: Largest dimension of brain volume sampling. MEA: multi-electrode array; Ca$^{2+}$: calcium imaging; ECoG: electrocorticography; EEG: electroencephalogram; fNIRS: functional near-infrared spectroscopy; fMRI: functional magnetic resonance imaging; fUSI: functional ultrasound imaging. Panel modified with permission from Griggs and Norman et al. 2024[25]. **B** 1.5 mm isotropic fMRI of intraparietal sulcus and adjacent cortex. Each red box represents one voxel. **C** 15.6 MHz 2D fUSI. Each red sheet represents one coronal imaging plane. Inset shows 100 µm × 100 × 400 µm voxel size. **D** Utah array and Neuropixel 1.0 recording methods for recording from intraparietal sulcus. Inset shows size of Neuropixel 1.0 electrodes (yellow). Panel a is adapted from Griggs, W.S., Norman, S.L., Deffieux, T. et al. Decoding motor plans using a closed-loop ultrasonic brain–machine interface. Nat Neurosci 27, 196–207 (2024). (https://doi.org/10.1038/s41593-023-01500-7) under a CC BY license: (https://creativecommons.org/licenses/by/4.0/).

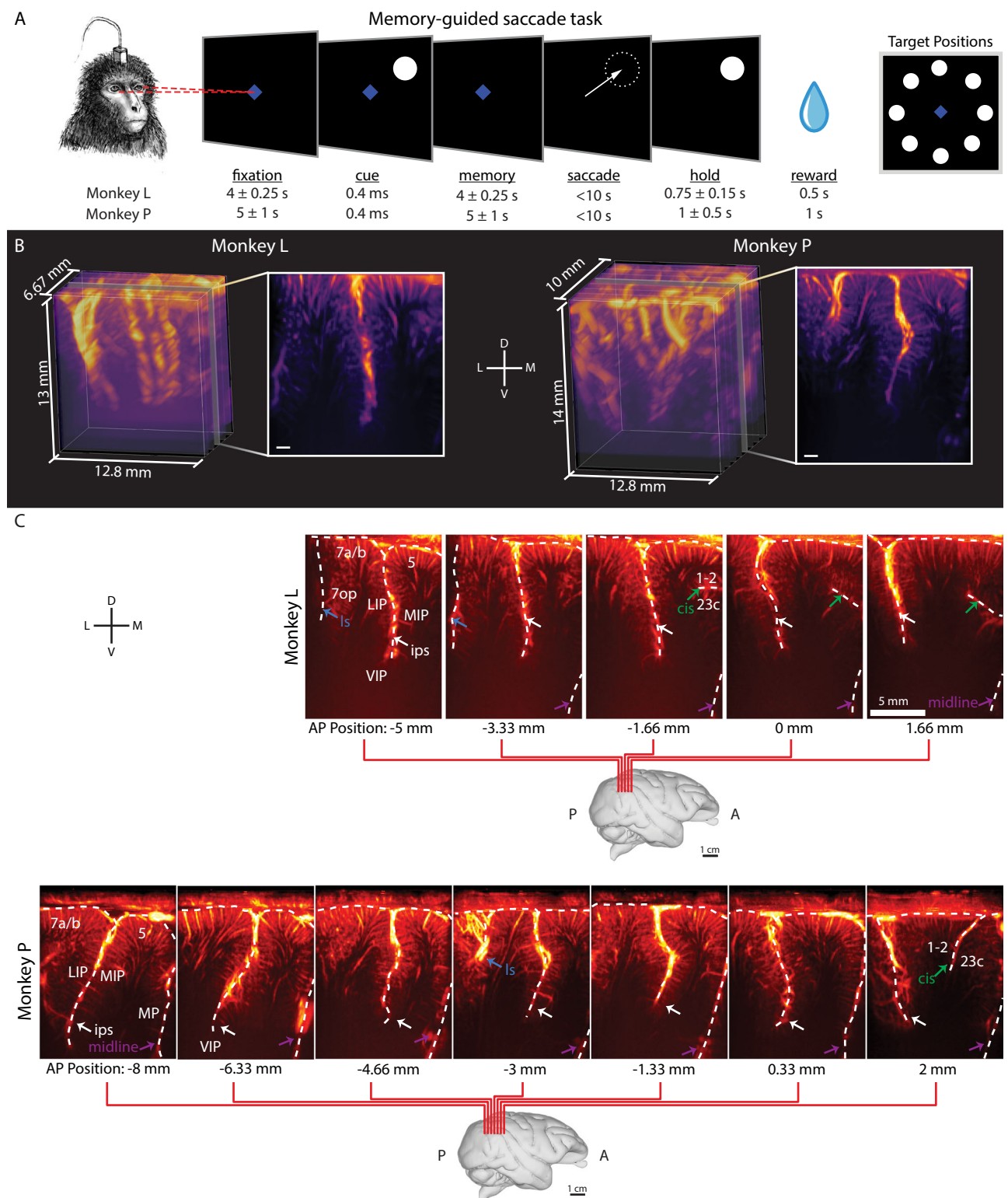

**Fig. 2 | Monkeys performed memory-guided saccade task during fUSI acquisition. A** Memory-guided saccade task. A trial began with the monkey fixating on a center blue diamond. After the monkey fixated, a white circular cue was flashed for 400 ms in one of 8 peripheral locations. Once the center fixation diamond extinguished, the monkey made a saccade to the remembered cue location and maintained fixation on the peripheral location. If the saccade was to the correct location, the peripheral cue reappeared and the monkey received a liquid reward. Example session median [IQR] saccade latency Monkey P-399 [363, 430] ms, Monkey L-530 [500, 569] ms. **B** 3D vascular maps for Monkey L and Monkey P. The

field of view included the intraparietal sulci for both monkeys. White scalebar−1 mm. *D* dorsal. *V* ventral. *L* lateral. *M* medial. **C** Coronal imaging planes in Monkey L and P. Position relative to estimated ear-bar zero (EBZ) overlaid on a NHP brain atlas[91]. Anatomical labels based upon Saleem et al. [26]. LIP lateral intraparietal area, VIP ventral intraparietal area, MIP medial intraparietal area, MP medial parietal area, ls lateral sulcus, cis cingulate sulcus, ips intraparietal sulcus. Monkey graphic in panel a was drawn by Krissta Passanante. Brain graphic in panel b generated through the Scalable Brain Atlas[92] and a publicly available MRI rhesus macaque atlas[91].

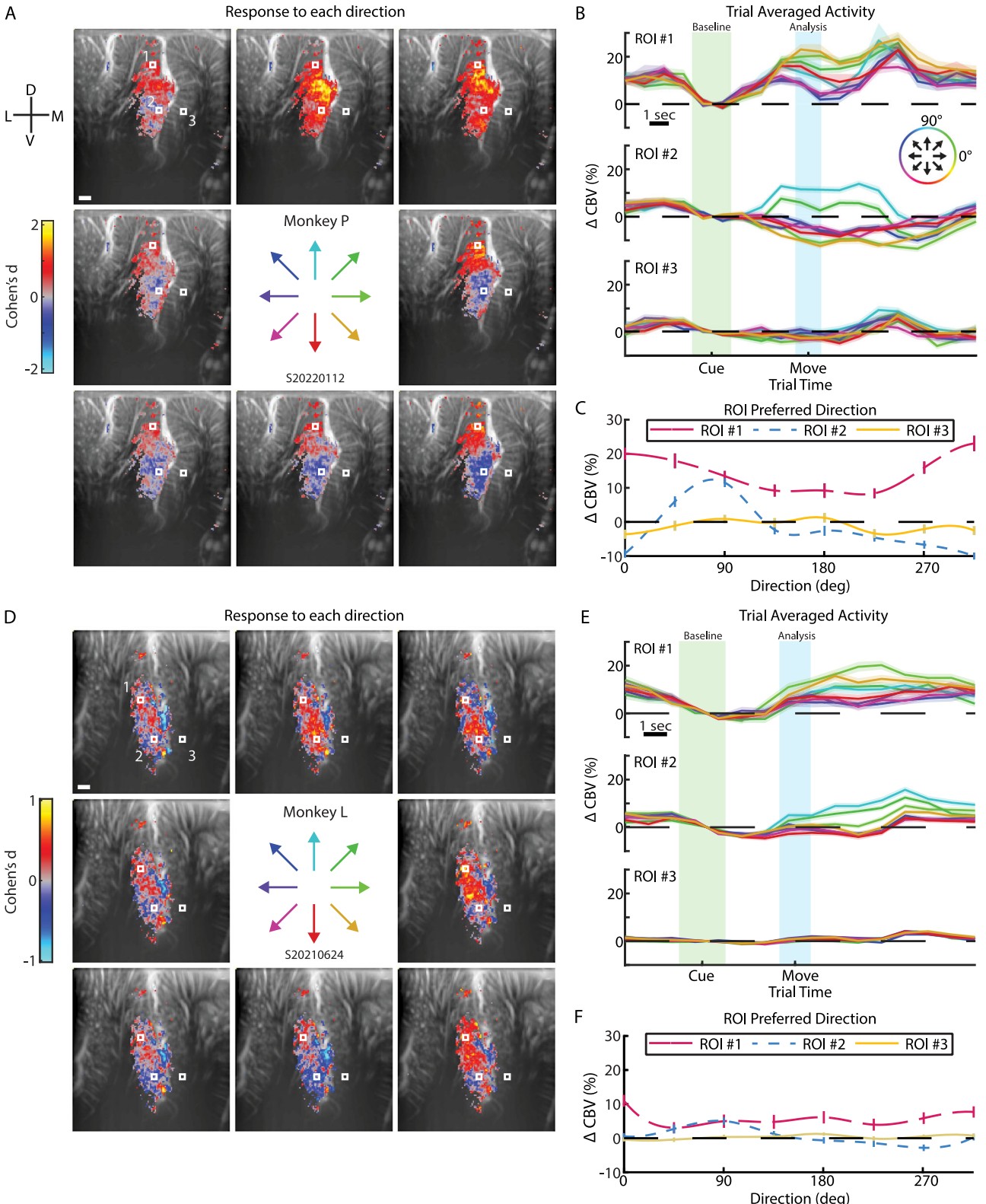

**Fig. 3 | PPC contains multiple distinct directionally-tuned mesoscopic populations. A** Statistical parametric maps showing the average activity during the memory period. Voxel threshold determined by GLM F-test for voxels where $p < 0.00001$ (FDR-corrected). ROI size is $400 \times 400 \, \mu m$. White scale bar−1 mm. Center arrows indicate the 8 directions tested. Cohen's d is a standardized measure of response strength for each direction where positive (negative) values correspond to increase (decrease) in CBV from the baseline. **B** Event-related average of activity within each ROI. Each line represents one direction. The circular color scale indicates the direction of each line. Error shading shows SEM. Green shading shows timepoints used for calculating baseline and blue shading shows timepoint used for analyzing memory response to the different directions. **C** Tuning curves. Each line shows a cubic spline fit to the directional responses at the end of the memory period within each ROI. Error bars show SEM. For A–C, plots based upon $n = 468$ trials. **D–F** Example session for Monkey L. Plots based upon n = 555 trials. Same format as (**A**–**C**) Source data are provided as a Source Data file.

'subpopulation' as a group of one or more adjacent voxels with highly similar responses to the different directions, i.e., highly similar tuning properties. Some LIP subpopulations exhibited substantial increases in CBV from baseline (>20%), with the magnitude depending on movement direction, while others showed direction-dependent suppression (Fig. 3B). To highlight this diversity of directional responses, we defined several example 400 × 400 µm regions-of-interest (ROIs) within LIP and MIP (Fig. 3A-white boxes, Fig. 3B) and averaged the response to each of the directions in the last second of the memory period (single timepoint within +/−0.5 sec of memory end) across all the ROI's voxels to create regional tuning curves (Fig. 3C). Different populations within a single coronal plane had different preferred directions and widths of their tuning curves. Some subpopulations were broadly tuned to the entire contralateral hemifield (Fig. 3B, C – ROI 1) while other subpopulations were tightly tuned to a narrow window of directions (Fig. 3B, C – ROI 2). The example session from Monkey L displayed similar phenomena (Fig. 3D–F). Most voxels (>70% of voxels) within LIP displayed directional modulation and these voxels were clumped into multiple LIP subpopulations with different tuning curves. Regions outside of the LIP, such as VIP, MIP, and Area 7, contained far fewer voxels with directional tuning (<10% of voxels within a given region). Similar to Monkey L, the regional tuning curves within LIP had different preferred directions and tuning curve widths. For example, ROI 1 displayed broad tuning whereas ROI 2 displayed narrower tuning to 45° and 90° (Fig. 3D–F). In Monkey L, the mid-MIP directly adjacent to the sulcus had some directionally tuned vascular response, but this activity did not penetrate deeper cortical layers.

To better understand the response fields at each voxel, we calculated several measures of data distribution for each voxel (Figure S2). We first found the peak preferred direction for every voxel. In both monkeys, these plots of peak tuned direction agree with our findings from above where we found different small groups of voxels tuned to different directions. We next visualized the response (Cohen's d) and statistical (F-score) strength of the response for the peak movement direction and found a similar strength of peak direction across all significant voxels. If tuning strength had been strongest in the middle of patches encoding for the same direction and weakest between patches, then it would indicate a smooth transition in tuning across PPC. However, this high uniform magnitude observed across the tuned voxels in LIP suggests a rapid transition in tuning between adjacent voxel patches and supports a patchy topography rather than a smooth gradient in tuning between neighboring voxel patches. We next examined whether the voxel-wise response fields were contralateral preferring[27,28], i.e., more active for saccades to the right because we recorded from the left PPC. As expected, most of the voxels in LIP were contralateral preferring (Monkey P – 74%; Monkey L – 77%). In Monkey P, we also observed a small patch in ventral LIP that was strongly ipsilateral preferring. We finally calculated and plotted the circular standard deviation, angular skewness, and angular kurtosis. These plots supported a similar patchy and heterogenous topography within LIP as identified by the other plots.

Due to large variability between recording sessions in Monkey L, we repeated the above analyses for an example session from a different coronal plane (Figure S3). We observed many of the same patterns observed in the first two example sessions. Many voxels within the anatomically-defined LIP (>30% voxels) showed directional tuning while few voxels in non-LIP regions (≤2% of voxels in any given region) showed directionally-modulated activity (Figure S1, S3A). In LIP, there were closely neighboring voxel patches with different tuning curves (Figure S3B, C). The statistical overlays (direction, response strength, statistical strength, circular standard deviation, angular skewness, angular kurtosis, and laterality index) showed a similar patchy topography identified in the other example sessions (Figure S3D).

## How consistent is this directional tuning within a session?

Having observed clear mesoscopic populations with directional preference in both monkeys, we aimed to better understand the information content within these voxel subpopulations and the consistency of their responses from trial to trial. To this end, we performed decoding analyses for each example session. We trained a model to decode the intended movement direction on a subset of each example session's trials and then tested how well the model could predict the intended movement direction on held-out test trials. If the model has statistically significant decoding accuracy on the test trials, it would demonstrate that the encoding of direction within the imaging field of view is consistent from trial to trial. For this decoding analysis, we used principal component analysis (PCA) to reduce the dimensionality of the fUSI data and linear discriminant analysis (LDA) to predict one of the eight movement directions using the PCA-transformed data. We examined the ability to decode the intended movement direction throughout the trial (Fig. 4) and found that we could begin decoding the intended movement direction significantly above chance ($p < 0.01$; 1-sided binomial test; leave-one-out cross-validation) within 3 sec of the directional cue onset (Fig. 4A, D). In both monkeys, the percent correct exceeded 50% (leave-one-out cross-validation; Monkey P – 59.6% correct, Monkey L – 54.1%). Missed predictions typically bordered the true movement direction (Fig. 4B, E). To quantify this, we present the mean absolute angular error between the predicted and true movement direction. As with the percent correct, the mean angular error reached significance ($p < 0.01$; 1-sided permutation test) within 3 sec of the directional cue. The mean angular error converged to <35° for both monkeys (Monkey P – 23.7°, Monkey L – 32.8°, Fig. 4A-bottom, D-bottom). The second example session in Monkey L (Figure S3) showed similar trends to the other example sessions but had worse decoding performance. The decoding performance still reached significance within 3 sec of the directional cue, but the percent correct and angular error only reached 30% and 55° respectively (Figure S3E). Missed predictions still bordered the true movement direction but had higher spread (Figure S3F) than observed in the first example session from Monkey L.

We used the entire image at each timepoint to decode the intended movement direction on individual trials. One possibility is that the prediction is being driven by a few voxels that stay consistent while the other voxels fluctuated. To understand which portions of the image, i.e., the vascular anatomy, contributed the most to the decoding performance, we performed a searchlight analysis. We moved a pillbox (200 µm radius) across the entire image and assessed the ability of each pillbox to decode the intended movement direction (Fig. 4C, F, and S3G). In other words, we serially examine how each unique group of voxels contained within a 200 µm radius pillbox can individually decode the intended movement direction. To separate information contained within different brain regions, we only analyzed voxels on the same side of the sulcus for a given searchlight pillbox. As an example, a pillbox centered on an LIP voxel only contained LIP voxels whereas a pillbox centered on an MIP voxel only contained MIP voxels. We found that many of these 200-µm radius pillboxes, or voxel patches, could robustly decode the intended movement direction ($p < 0.01$) with many of the voxel patches approaching 30° angular error in our example sessions (Fig. 4C, F). In the supplemental example session (Figure S3G), the searchlight still identified voxel patches that could decode intended movement direction, but the angular error only approached 75°. In other words, these searchlight results demonstrated that many PPC voxels encoded the intended movement direction and could drive accurate single-trial predictions. LIP contained the greatest number of informative voxels and these informative voxels had high overlap with the same voxels identified with the previous GLM analyses (Figs. 4G, and S3H) as measured by Dice-Sørenson similarity. Additionally, voxels in all the example sessions displayed a strong and significant correlation between F-score (GLM

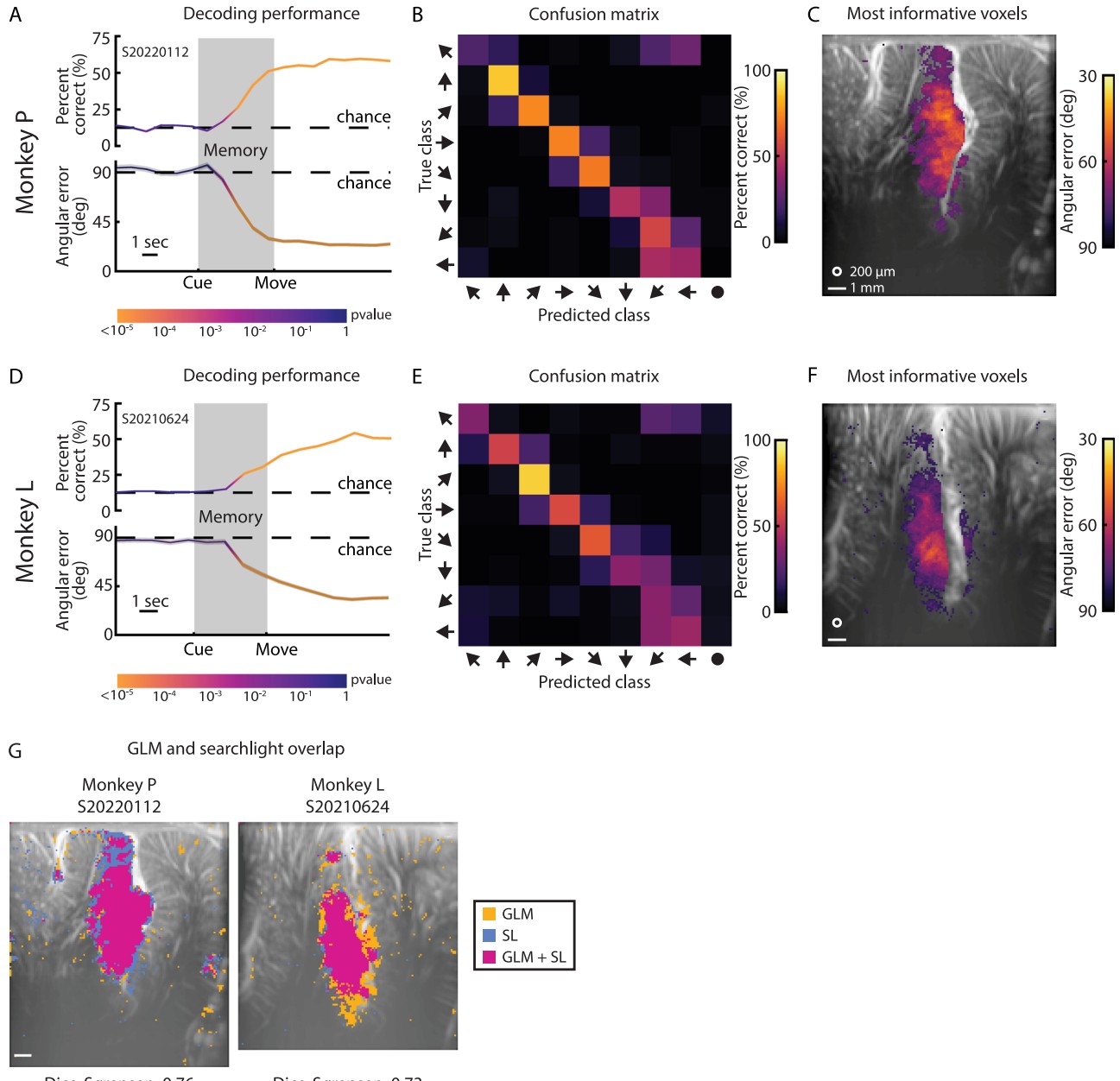

**Fig. 4 | Single-trial decoding of eight intended movement directions with high accuracy. A** Decoding performance as a function of time. Top plot shows percent correct. Bottom plot shows mean angular error. Shaded error bars around mean angular area shows standard error across the folds of the leave-one-out analysis. Dashed lines show chance level performance. Color of the line shows statistical significance (1-sided binomial test or permutation test). **B** Confusion matrix of decoding from last timepoint in trial. Performance represented as percentage (rows adds to 100%). The circle predicted class corresponds to the 'center' position that emerges from the multicoder approach. **C** Searchlight analysis at end of trial.

Top 10% of voxels with the lowest mean angular error. White circle – 200 μm searchlight radius. White line – 1 mm scalebar. Masked voxels correspond to threshold of $p < 10^{-5}$ (1-sided binomial test with FDR correction) For (**A**–**C**), plots based upon $n = 468$ trials. **D**–**F** Decoding performance for Monkey L. Same format as (**A**–**C**). Masked voxels corresponds to threshold of $p < 0.005$ (1-sided binomial test with FDR correction). Plots based upon $n = 555$ trials. **G** Overlap of GLM and searchlight analysis statistical masks ($p < 0.001$, FDR-corrected). Overlap calculated using Dice-Sørenson similarity. Source data are provided as a Source Data file.

analysis) and angular error (searchlight analysis) (Figure S4). In both animals, voxels in regions outside of LIP were within the 10% most significant voxels (Fig. 4C, F). In Monkey P, these regions included Area 7 (14% of the most significant voxels), MIP (11%), VIP (5%), and MP (2%). In Monkey, these regions included MIP (8%), VIP (5%), and Area 7 (4%). The significant voxels within MIP were in the superficial cortical layers along the sulcus and did not extend into deeper cortical layers, matching the results from Monkey L's GLM analysis. To understand how these searchlight results evolved across trial time, we repeated the

searchlight analysis for each timepoint after fixation (Figure S5). Like the decoder performance using the entire image, the searchlight plots began showing significant voxels within LIP within three seconds after fixation. Significant voxels appeared within other regions, including Area 7, VIP, MIP, and MP, in later trial timepoints. The number of significant voxels and decoder performance plateaued by the end of the movement period.

The searchlight analyses identified voxels contributing to decoding performance but did not specify which voxels were critical

for distinguishing specific movement directions. To address this, we examined the weights of the PCA-LDA decoder trained on full fUSI images (Figure S6). LDA weights represent boundaries between pairwise movement directions (e.g., 0°–45°, 45°–315°) and define the features that best discriminate between them. Since we trained the LDA model on PCA-transformed data, we projected the LDA weights back into the original vascular image space using the inverse PCA transform to visualize the importance of different voxels in the LDA model (Figure S6). Voxels with stronger weights were more important for distinguishing specific directional pairs, while those with near-zero weights were less relevant. We found that different PPC subregions specialized in discriminating specific directional pairs rather than uniformly contributing to all pairs. For instance, mid-LIP voxels in both monkeys showed the strongest weights for neighboring directions (e.g., ipsilateral upwards in Direction 1 vs. other directions in Direction 2; Figure S6). This pattern demonstrates distinct populations within PPC with strong tuning for specific directions. Additionally, the voxels with the strongest LDA weights aligned closely with GLM-identified voxels (Figure S6 – Dice-Sørenson plots), further supporting that mesoscopic LIP subpopulations robustly encode movement directions. These findings highlight the synergy between the PCA + LDA decoder and GLM at revealing specialized subregions within PPC for movement direction encoding.

## Are these mesoscopic populations stable across multiple days?

In the example sessions, PPC subpopulations were robustly tuned to individual movement directions, but is the function in each population stable across time? In a previous paper, we showed that populations in PPC could be used to control an ultrasonic brain-machine interface even after 60+ days since training the decoder model, suggesting that PPC populations are stable across 1-2 months[25]. To extend this result and better understand the stability across many months to years, we collected data from the same coronal plane across 4–30 months. We then trained our decoder on one session's data and tested its performance on other sessions from the same plane without retraining the model. We tested all combinations of sessions. If the subpopulations' functions were constant across time, a decoder trained on one session would accurately predict intended movement directions on another session's data from the same coronal imaging plane.

In Monkey P, the decoder performed above chance level for over 100 days (Fig. 5A) and across all pairs of training and testing sessions ($p < 10^{-5}$; 36/36 pairs) (Fig. 5B). In Monkey L, the decoder performed above chance level for more than 900 days between the training and testing sessions (Fig. 5D), an effect that persisted across nearly all pairs of training and testing sessions ($p < 0.01$; 117/121 pairs) (Fig. 5E). Cross-validated decoding performance varied within each training session (diagonal of performance matrices; Fig. 5B, E), so we also present cross-session decoding accuracy normalized to each training session's cross-validated accuracy. We did not observe any clear differences between the absolute and normalized accuracy measures. Interestingly, in Monkey L, the decoder trained on the Day 119 session performed the best for three directions (contralateral up, contralateral down, and ipsilateral down) in the training set and continued to decode these same three directions the best consistently throughout the test sessions (Fig. 5D). We saw this pattern where the decoder could best predict certain directions, even when the training session had poor cross-validated performance by itself (Figure S7). We also observed in both monkeys that temporally adjacent sessions exhibited better performance (Figs. 5C, F, and S8).

In Monkey L, the performance was clumped into two temporal groups (before and after Day 900). Was this change in performance due to physical changes in the imaging plane or due to changes in subpopulation function? Although we did our best to align our recordings to the exact same imaging plane from day to day, occasionally our alignment was imperfect and out-of-plane compared to

previous sessions. Visual inspection revealed consistent macrovasculature, e.g., arterioles, but inconsistent mesovasculature (Figure S9A, D). Importantly, the physical changes would suggest that (a) we were decoding from slightly different neural populations and (b) small neighboring neural populations encode different directional information. To test our hypothesis that physical differences in imaging plane (and therefore differences in vascular anatomy) led to the decrease in decoder performance, we measured the similarity of the vascular anatomy across time using an image similarity metric: the complex-wavelet structural similarity index measure (CW-SSIM)[29]. The CW-SSIM clumped the vascular images into discrete groups (Figure S9B, E), matching our qualitative assessment of image similarity. The similarity grouping also matched the pairwise decoding performance grouping in Monkey L (Fig. 5E). The decoder performance and image similarity were correlated (Figure S9C, F). As image similarity decreased between the training session and each test session, so too did decoder performance. This supports our hypothesis that the decrease in decoder performance resulted from changes in the imaging plane rather than drift in each subpopulation's tuning.

Together, these longitudinal decoding results demonstrate subregions tuned to specific movement directions remain consistent across many months to years. Our analysis revealed three key findings. First, the whole image decoder relies upon the LIP subregions most strongly tuned to specific movement directions (Figure S5). Second, voxels identified by the searchlight and GLM analyses showed substantial overlap (Fig. 4G, and S5H). Third, the whole-image decoder maintained stability across extended time periods, indicating that the voxels supporting decoder performance are similarly stable over time.

## How does mesoscopic population tuning change across anterior and posterior portions of PPC?

Having demonstrated that, within an imaging plane, there are PPC subpopulations robustly tuned to individual movement directions and these subpopulations' tunings are consistent across many months to years, we next asked how direction tuning varied across different anterior to posterior coronal imaging planes. We repeated the same GLM analysis for data acquired from coronal planes evenly spaced throughout the PPC (Fig. 2B, C, and Supplementary Movie 1, 2) and found the peak preferred direction for every voxel (Fig. 6A). Several patterns appeared. First, each coronal plane contained LIP voxels with directional modulation. Second, each anatomical plane in both monkeys contained multiple LIP subpopulations with different tuning properties. Third, posterior planes encoded more contralateral upward movements while anterior planes encoded more contralateral downward movements. Fourth, in both monkeys, regions outside of the LIP contained directionally modulated voxels. In Monkey P, these regions included Area 7, MIP, MP, and Area 5 cortex. In Monkey L, we observed a small region in ventral Area 7a. We did not observe any activity within Area 5 of Monkey L and only observed very superficial activity within MIP of a single coronal plane (−3.33 mm of EBZ). Unfortunately, Monkey L's recording chamber was more lateral and did not contain the same posterior portion of MP where we observed activity in Monkey P.

To further understand the directional encoding across different coronal planes, we extracted the directional-modulated voxels within LIP and created a beeswarm chart containing each voxel's directional preference (Fig. 6B). As expected, certain directions within a given plane were over-represented, i.e., clumps of similarly tuned neurons in the beeswarm plot. For both monkeys, we found a statistically significant relationship between anterior-posterior location and preferred direction ($p < 1e-40$) where more anterior planes had more LIP voxels tuned for downwards directions while posterior planes had more LIP voxels tuned for upwards directions. Most LIP voxels encoded for contralateral movements (−90° to +90°) although there were some voxels that responded most strongly, i.e., a CBV increase, for

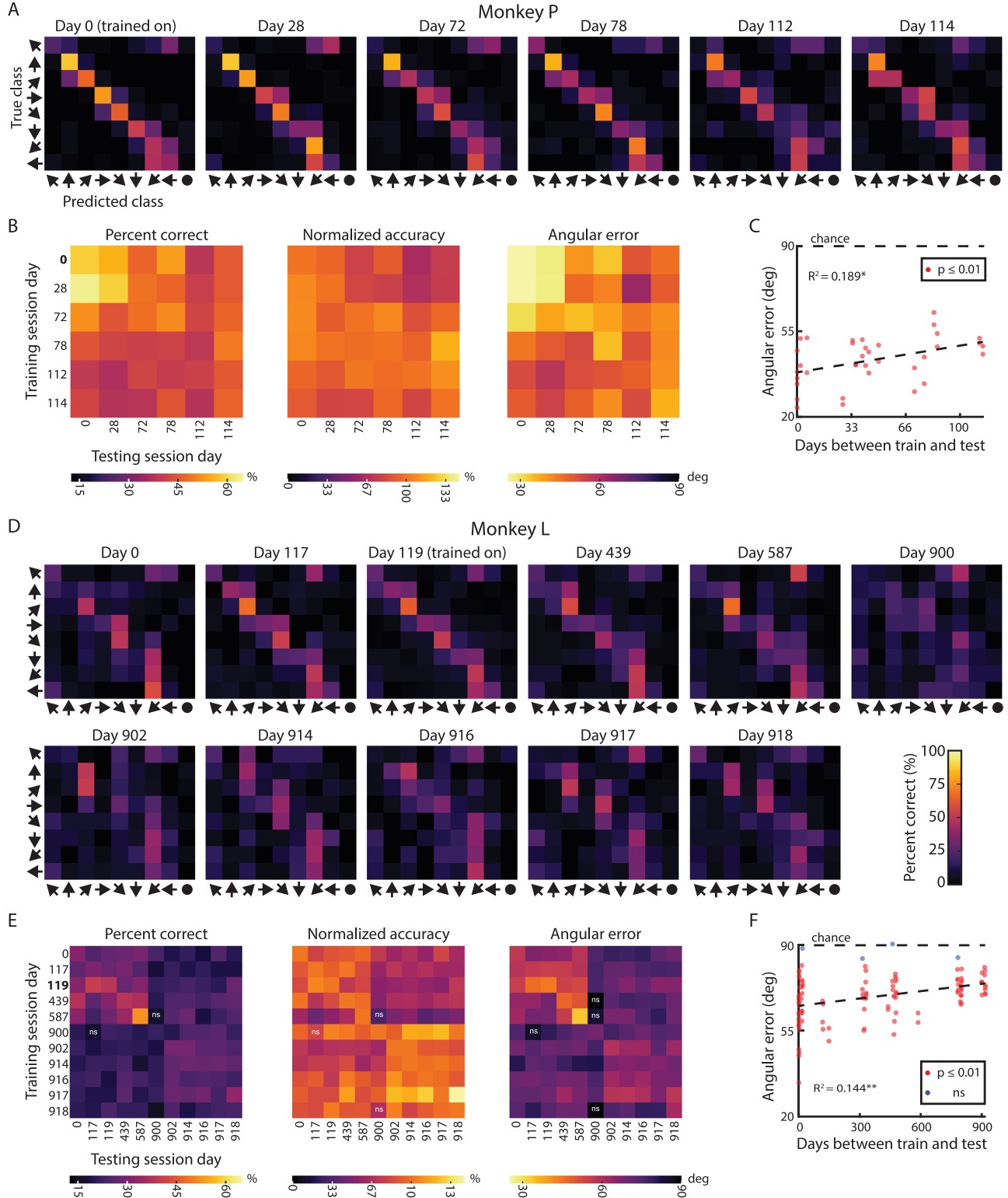

**Fig. 5 | PPC stably encodes movement direction across many months to years.**
**A** Example decoder stability for Monkey P. Trained the decoder on Day 0 data and tested the trained decoder on other sessions from the same imaging plane without any retraining. **B** Decoder stability for training and testing on each session for Monkey P. ns – nonsignificant decoding performance ($\alpha = 0.01$). Bold text represents example session shown in Figure 5A. **C** Mean angular error as a function of days between the training and testing session (absolute difference in time) for Monkey P. Dashed line – Linear fit to data. *$p = 0.0079$, **$p = 1.76\text{e-}5$ (2-sided t-test). **D** Example decoder stability for Monkey L. Trained the decoder on Day 119 data and tested the trained decoder on other sessions from the same imaging plane without any retraining. **E** Decoder stability for training and testing on each session for Monkey L. Same format as Figure 5B. **F** Decoder stability for Monkey L. Same format as Figure 5C. Source data are provided as a Source Data file.

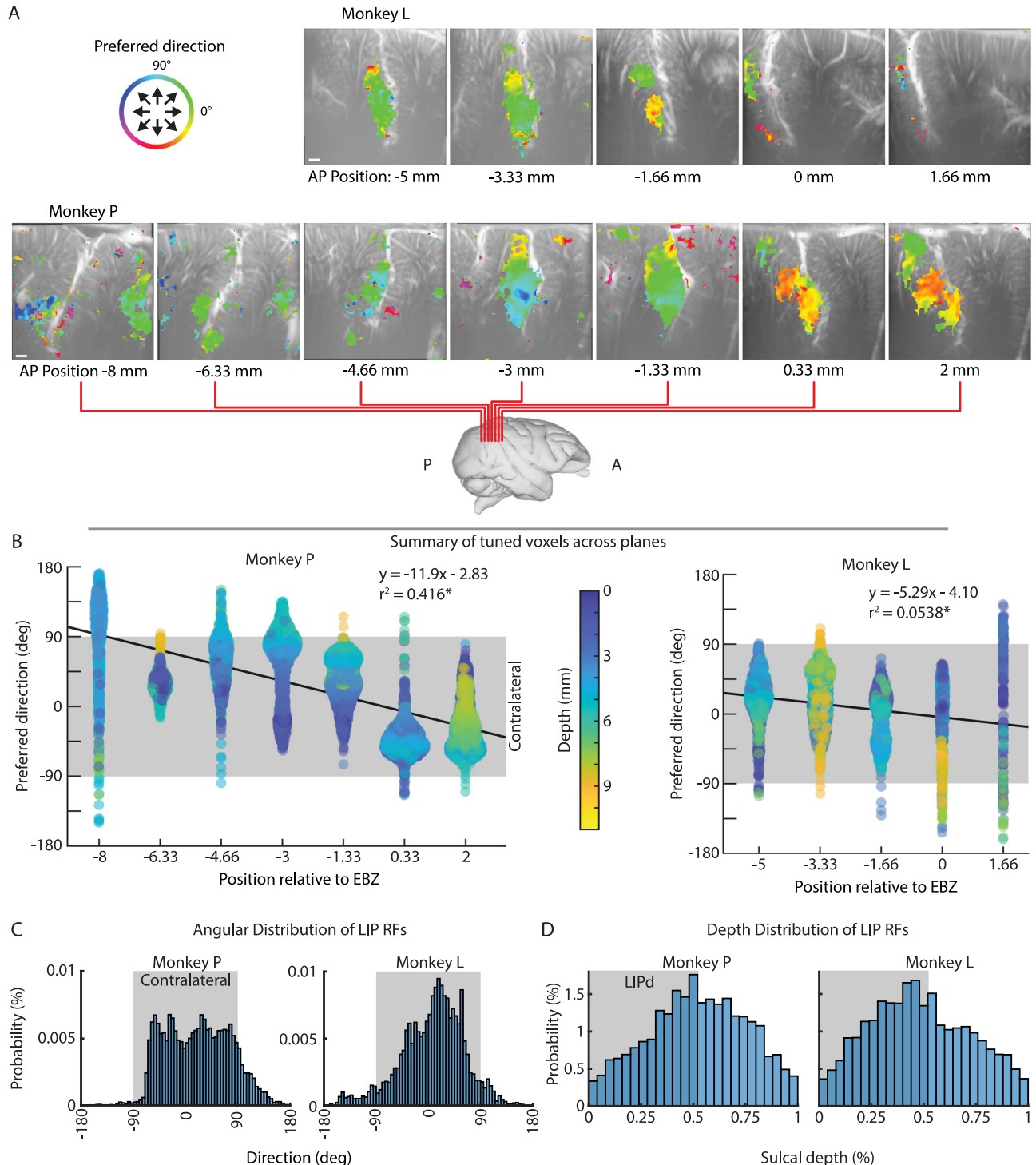

**Fig. 6 | Polar direction is topographically organized along anterior-posterior axis of LIP. A** Color overlays showing preferred direction for voxels with statistically significant difference in response for different movement directions. Threshold based upon GLM 2-sided F-test where $p < 0.01$ (FDR-corrected). **B** Preferred direction for all significant voxels within each coronal plane. Color represents depth from brain surface. Gray shaded area shows contralateral angles. Black line = Line of best fit with equation shown. *$p < 1e-40$ (2-sided F-test with FDR correction). **C** Angular distribution of response fields within LIP. Gray shaded area shows contralateral angles. **D** Depth of tuned LIP voxels. Gray shaded area shows approximate LIPd. Source data are provided as a Source Data file. Brain graphic in panel a generated through the Scalable Brain Atlas[92] and a publicly available MRI rhesus macaque atlas[91].

ipsilateral movements. Each of the planes had broad and overlapping representation of contralateral movements. To better quantify these observations, we collapsed the voxels across planes (Fig. 6C) and found that >85% of tuned LIP voxels were contralateral preferring (Monkey P – 87.7%; Monkey L – 89.4%). We next examined the data

distribution across each coronal plane (Figures S10, S11). Similar to the example sessions (Figures S2, S3), the tuning and statistical strength metrics (Figure S10A, B) displayed a high uniform magnitude across tuned voxels, suggesting a rapid transition in tuning between adjacent voxel patches and supporting a patchy topography rather than a

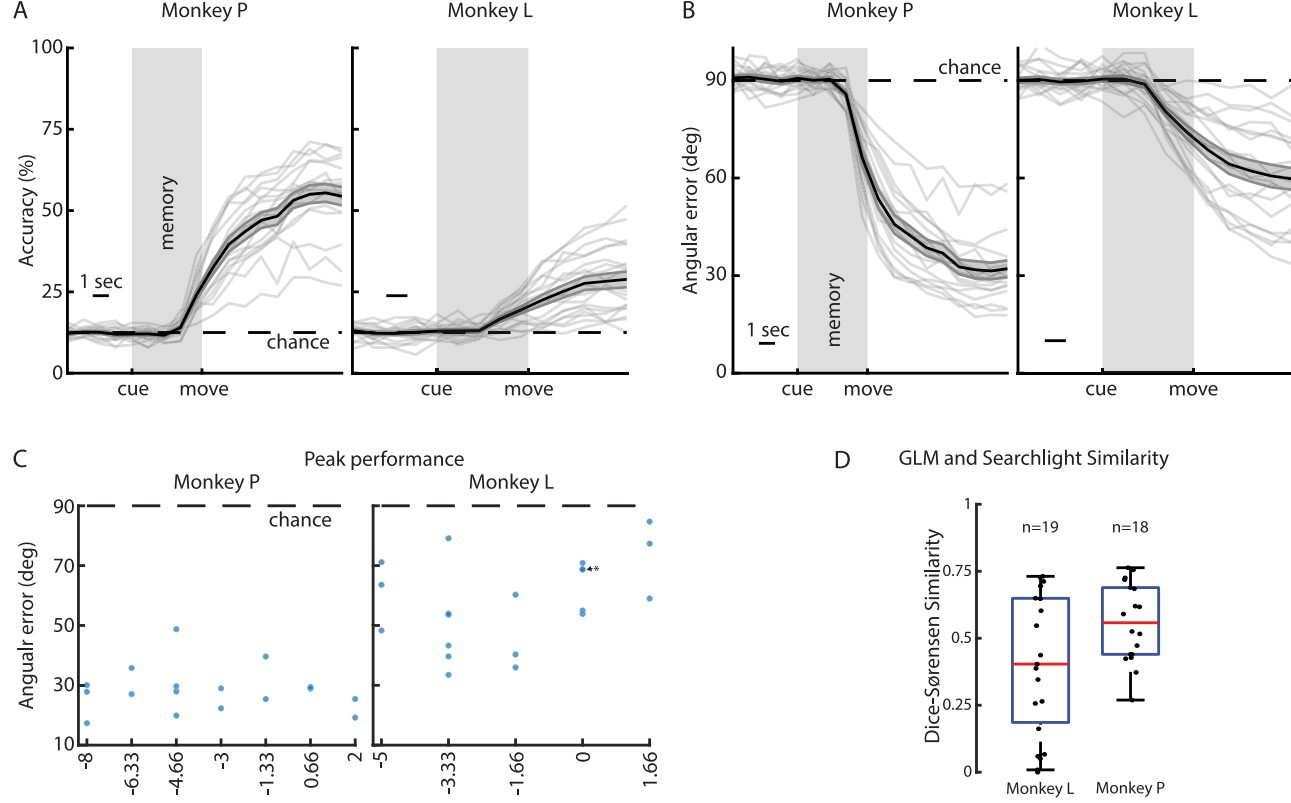

**Fig. 7 | Linear decoders can decode intended movement direction from most fUSI sessions, regardless of PPC plane.** Percent correct for each session. Solid black line with gray envelope show mean ± SEM. Each gray line shows performance on single session. Dashed line shows chance level. **B** Mean angular error for each session. Same format as in (**A**). **C** Mean angular error as function of coronal plane. *=two closely overlapping points (**D**). Summary of overlap of GLM and searchlight

analysis across all sessions. Each dot represents 1 session. Box shows 25th-75th quartiles with red line showing median Dice-Sørenson value. Whiskers extend to furthest non-outlier point (± 1.5*IQR beyond the 25th or 75th quartiles). Minima [maxima] are -1.5*IQR [ + 1.5*IQR] beyond the 25th [75th] percentile. One session from Monkey L excluded due to no significant voxels in GLM or searchlight analysis. Source data are provided as a Source Data file.

smooth gradient in tuning between neighboring voxel patches. As the main analyses showed, most voxels were contralateral preferring (Figure S10C). As with the example sessions (Figures S2, S3), the circular standard deviation, angular skewness, and angular kurtosis supported a patchy and heterogenous topography within LIP.

Certain anterior/posterior coronal planes had better representation of specific directions, so we asked whether there would be any performance difference in fUSI decoders trained on the different anatomical planes. This would have translational implications if specific anterior/posterior regions were better for directional decoding. We applied our decoding analysis to every recorded session (Fig. 7). In Monkey P, all sessions reached statistical significance (18/18 sessions). In Monkey L, all but one session reached statistical significance (19/20 sessions). In Monkey P, the peak angular error within a session ranged from 17° to 55° (29.97° ± 2.32° mean ± SEM). In Monkey L, the angular error ranged from 33° to 85° (57.98° ± 3.35° mean ± SEM). There was no statistical difference (1-way ANOVA, α = 0.01) between the percent correct or angular error depending on the plane (Fig. 7C). These results suggest that all anatomical PPC planes we sampled contained sufficient information to accurately decode at least eight intended movement directions on a single-trial basis.

We next compared the similarity between the significant voxels identified by the GLM and searchlight analyses (Fig. 7D). As with the example sessions, there was a high degree of similarity between the voxels identified in the two analyses. In Monkey L, the median [IQR] Dice-Sørenson similarity was 0.40 [0.19, 0.65]. In Monkey P, the median [IQR] Dice-Sørenson similarity was 0.56 [0.44, 0.69]. This

further demonstrates how the decoder and GLM analyses effectively complement each other in revealing specialized subregions within PPC for movement direction encoding.

### Do dorsal and ventral LIP display different tuning properties?
Dorsal (LIPd) and ventral (LIPv) LIP are believed to have different functions with LIPd being involved in visual processing while LIPv is involved in both attentional and visual-saccadic processing[30,31]. According to theories of topographic encoding[12], we would expect separate representations of movement directions within LIPd and LIPv. We observed no clear split in function at the middle portion of LIP, so we relied upon a previous definition of 53% sulcal depth to compare activity within LIPd and LIPv[30]. We found that anterior LIPd was more active than posterior LIPd. Middle LIP, i.e., junction of defined LIPd and LIPv, consistently demonstrated the most activity across all planes. To quantify this observation, we labeled the beeswarm chart with the depth of each LIP voxel from the brain surface (Fig. 6B). We did not observe any clear trends in the data to clearly distinguish functional differences between LIPd and LIPv. We additionally collapsed all the tuned voxels across planes and looked at their percent depth within the sulcus (Fig. 6D). Instead of observing a clear separation between LIPd and LIPv, we observed one homogenous group with the most activity peaking at the boundary between LIPd and LIPv, suggesting that LIPd and LIPv may share a topographic representation and/or that they receive common shared inputs.

## Discussion

Our results demonstrate that PPC contains subregions tuned to different directions. These tuned voxels were predominantly within LIP and grouped into contiguous mesoscopic subpopulations. Multiple subpopulations existed within a given coronal plane, i.e., there were multiple preferred directions in each plane. A rough topography exists where anterior LIP had more voxels tuned to contralateral downwards saccades and posterior LIP had more voxels tuned to contralateral upwards saccades. These populations remained stable across more than 100–900 days.

### fUSI-specific features

**Sensitivity of fUSI.** We observed large effect sizes with changes in CBV on the order of 10–30% from baseline activity (Fig. 3). This is much larger than observed with BOLD fMRI where the effect size was ~0.4–2% on similar saccade-based event-related tasks[27,32]. Our results support a growing evidence base that establishes fUSI as a sensitive neuroimaging technique for detecting mesoscopic functional activity in a diversity of model organisms, including pigeons, rats, mice, non-human primates, ferrets, and infant and adult humans[23–25,33–40].

**Mesoscopic functional imaging–balancing spatial resolution and field-of-view.** Several studies have reported a patchiness in direction selectivity with many neighboring neurons tuned to approximately the same direction followed by an abrupt transition to a patch of a different preferred direction[13,14,41]. These results match very closely with the results observed in this study where we found clusters within LIP tightly tuned to one direction with differently tuned clusters in close proximity within a given plane. These results further emphasize the high spatial resolution of fUSI for functional mapping of neuronal activity. These results also closely match a previous study that used fUSI to identify the tonotopic mapping of the auditory cortex and inferior colliculus in awake ferrets where the authors found a functional resolution of 100 μm for voxel responsiveness and 300 μm for voxel frequency tuning[34].

### Stability of directional tuning across time

fUSI allows for repeated, longitudinal imaging of the same brain regions across months to years, facilitating studies of the stability of functional populations over time. Here, we used this feature to demonstrate that we could decode intended movement direction using a decoder trained on data from a different session many months to years apart. This strongly suggests that the directional preference for the LIP subpopulations remains stable at the mesoscale. The decoder performed best when the training and testing sessions were close in time. We have three possible interpretations for this. First, the representations of direction within subpopulations drift in difficult-to-predict ways across time. Under this interpretation, we would expect that the decoders' predicted movement directions would become increasingly random as more time elapses as the tuned voxels used for the model decorrelate. A second interpretation is that the subpopulations drift, but they drift at the same rate and in the same directions. This would lead to the tuned voxels staying correlated but encoding for different directions. Under this interpretation, we would expect to see the decoder make increasingly more mistakes, but in a consistent manner. For example, the decoder might develop an error bias where instead of predicting the correct class, it consistently predicts a different direction in its place. A third interpretation is that the vascular placement relative to our recording plane changed across time (consistent with changes in the imaging plane) and that we are decoding from slightly different neural populations. Under this interpretation, the decoding errors should increase for neighboring directions because the tuned mesoscopic populations observed in this study extend in both anterior-posterior directions and smoothly transition to encoding different directions rather than having sharp transitions where neighboring voxels encode for completely different directions. This smooth transition means that the populations used for decoding will still be similar to the original populations being measured, consistent with changes in the imaging plane.

Our data best supports the third interpretation: the recording plane physically shifted over time. Rather than the errors becoming increasingly random as time progresses, the confusion matrices still had strong diagonal components, i.e., correct predictions, but with higher variance about that diagonal. Additionally, image similarity metrics drifted across time, confirming that the imaging plane changed despite our best attempts at acquiring the exact same imaging plane during each session (Figure S9). Finally, the decoder performance and image similarity were positively correlated. This supports the interpretation that the subpopulations are stable across time with our decoder performance decreasing because of our imaging plane changing.

### Preference for contralateral space

Previous studies found that LIP responds strongest to contralateral stimuli and movements. At the single neuron level, ~80–90% of LIP neurons are tuned to contralateral directions[13,14]. Of note, Platt and Glimcher 1998[42] reported no bias towards contralateral or ipsilateral in their recorded LIP neurons. At the macroscopic population level, the BOLD response in LIP is also almost exclusively contralateral preferring[15,16,27]. In the present study, we also found that LIP has strongly lateralized responses with ~88% of LIP voxels preferred contralateral directions. The reasons for the apparent discrepancy with Platt and Glimcher 1998[42] remain unclear.

### Anatomical considerations

**Anterior-posterior gradient.** Our results extend previous studies' evidence of topography within LIP[12]. Two previous fMRI[15,16] and one electrophysiology[14] studies found an anterior-posterior gradient for encoding of visual field where visual objects in the lower visual field evoked activity within anterior LIP and visual objects in the upper visual field evoked activity within posterior LIP. Our fUSI data (Fig. 6) found a similar anterior-posterior gradient but for the planning and execution of eye movements instead of for the location of visual objects. However, visual receptive fields and saccadic motor fields have been shown to overlap[43], suggesting a similar anterior-posterior gradient for saccade directions.

Two studies[13,18] found different patterns. Blatt et al. 1990[13] examined visual receptive fields for both direction and distance from the fovea and sampled mostly regions within the lateral bank of the intraparietal sulcus that was more posterior to the current study with only a small range of overlap with monkey P (overlap of approximately 4 mm, −4 to −8 mm of EBZ). Savaki et al. 2011[18] found that the upper part of oculomotor space is represented in dorsal-anterior LIP while the lower part of oculomotor space is represented within ventral-posterior LIP. In this range, Blatt et al. 1990[13] found visual receptive fields in the upper and lower visual field evoked activity within anterior LIP with lower visual fields more dorsal. Visual receptive fields in the upper visual field were found ventrally and lower visual field dorsally within posterior LIP. Savaki et al. 2011[18] found that the upper part of oculomotor space is represented in dorsal-anterior LIP while the lower part of oculomotor space is represented within ventral-posterior LIP. Arcaro et al. 2011[16] reconciled their fMRI data with the two electrophysiology studies by suggesting that the differences result solely because of differences in recording site location, i.e., the two electrophysiology papers recorded from different overlapping anterior-posterior ranges of LIP. Our combined range of recording agreed with the results that Arcaro et al. 2011 showed for visuotopic LIP (LIPvt) and caudal intraparietal cortex (CIP-2). Over the range of overlap with Blatt et al. 1990, we observed a similar tuning of ventral LIP for contralateral upward. Taken together, our results support the

interpretation that differences in recording site location explain the difference in anterior-posterior gradients.

**Differences between dorsal and ventral LIP.** Previous studies found that peripheral targets were represented within the LIPv while foveal and parafoveal targets were represented within the LIPd[13–16]. In our study, we focused on movement preparation to a single eccentricity (20°) and observed activity within both LIPd and LIPv. In both monkeys, we observed less LIPd activity in the more posterior planes (<−3.33 mm of EBZ). The overall distribution of tuned LIP voxels did not demonstrate a clear separation between LIPd and LIPv. Our study was not designed to interrogate the encoding of eccentricity, so future fUSI studies with foveal, parafoveal, and peripheral targets will be needed to explore the mesoscopic representation of saccade eccentricity within LIP and PPC.

Previous studies have found that LIPd was primarily involved in processing vision-related information while LIPv contributed to attentional, vision, and oculomotor processes[30,31]. Our task was aimed at understanding the oculomotor representation of different directions within LIP but was not designed to separate the effects of attention from oculomotor planning. In our study, we did not demonstrate a separation in directional representation between LIPd and LIPv with the most activity peaking at the boundary between LIPd and LIPv. This suggests that LIPd and LIPv may share a common topographic representation rather than having separate duplicated representations of angular direction.

**Directional saccadic activity outside of LIP.** In both monkeys, we observed directionally-modulated activity outside of the LIP in the example sessions and aggregated data in VIP, MIP, and Area 7. In Monkey P, we additionally observed directionally modulated activity in MP. MP has been previously identified as a saccade-related area in single-unit electrophysiology and fUSI studies[24,44]. Monkey L's recording chamber was located lateral of MP areas. Nevertheless, our results in Monkey P, combined with observations in previous studies, support that MP may be an underexplored oculomotor planning region. The MP voxels preferred contralateral directions and did not display any clear organization of their response fields.

The posterior-medial MIP also contained directionally tuned activity. Some of this activity (−8 mm of EBZ, Monkey P) is in superficial cortical layers, perhaps reflecting inputs to MIP that relay directional information from upstream brain regions. We also observed activity within the deeper layers of posterior-ventral MIP in both monkeys, agreeing with previous literature that found visual and saccade-related activity within MIP[45,46]. However, previous work also found the functionally-defined parietal reach region (PRR), overlapping with the anatomically-defined MIP, responds predominately to reach movements[2,4,45]. Our task was a memory-guided saccade task with no reach component. Monkey P sat in an open chair with his hands and arms free while Monkey L sat in an enclosed chair with his hands and arms confined. Despite Monkey P being free to move his arms, we did not observe any arm movements related to the task itself. Future fUSI studies where we use a task with intermingled reaches and saccades will be useful in elucidating the contribution of visual stimuli, saccades, and reaches to the observed directionally-modulated activity within MIP.

Different subregions within VIP also contained directionally tuned activity in both monkeys with no consistent pattern of which VIP subregions showed directionally-modulated activity across coronal planes or between monkeys. Previous research has shown that VIP has a high degree of selectivity for the direction of a moving stimulus with minimal, if any, activity to a static stimulus presented within its receptive field[47]. Additionally, VIP is selective for smooth pursuit eye movements rather than saccadic eye movements[48]. Future work with moving stimuli and smooth pursuit eye movements will be necessary to elucidate what components of our task elicited activity within VIP.

In our experiments, putative Area 7a displayed directionally tuned activity to contralateral movement directions in both monkeys. This is consistent with previous literature that found Area 7a neurons have visual receptive fields and display saccade-related activity[49–52]. We do not know why only a small region of Area 7 A in Monkey P showed directional tuning or why fewer coronal planes in Monkey L displayed directionally tuned activity in Area 7a. One possibility is that the neurons within individual voxels of Area 7a display high heterogeneity in their response fields such that no consistent tuning appears at the mesoscopic population level.

### Applications to ultrasonic brain-machine interfaces
We previously showed that we could decode movement timing (memory/not-memory), direction (contralateral/ipsilateral), and effector (hand/eye) simultaneously on a single-trial basis with high accuracy[24]. We recently also demonstrated that we could train monkeys to use a real-time fUSI brain-machine interface (BMI) for up to eight directions of eye movements[25]. Here, we extended these papers' results in several aspects.

First, we demonstrated that we could achieve better decoding performance using offline recorded data (50–60% correct) than the accuracy reported for the online real-time fUSI-BMI data (~38% correct). One explanation for this performance increase is motion correction. In the present study, we used *post hoc* motion-correction to minimize movement of the imaging plane across a session. In the real-time fUSI-BMI study, we did not implement motion correction. In the present study, we showed why mesoscopic populations are tolerant to a small amount of motion: similarly tuned voxels are more often spatially contiguous. However, even modest amounts of motion would alter the information available to the decoder, decreasing performance. One future method that may be well-suited to this problem is convolutional neural networks that can utilize local structure in images to maintain high performance rather than our existing decoder algorithms that assume features do not move across time.

Second, we demonstrated that we could decode above chance level with a static decoder model even after several years. In Griggs and Norman et al. 2024[25], we collected data over 79 days, far fewer than the 900 days reported here. This suggests that future ultrasonic BMIs can constantly update an existing model rather than needing to be recalibrated daily. This is one current advantage of imaging-based BMIs over intracortical electrode-based BMIs. Intracortical electrode-based BMIs typically require frequent calibration or retraining due to their inability to record from the same neurons across multiple days[53]. By simply combining imaging-based BMIs with image alignment (2D plane and potentially 3D volume) to a previous session's field of view, we can stabilize BMIs over long periods of time.

In Monkey L, we observed high variability in decoder performance across sessions (Fig. 7). We explored several factors including anatomical plane, number of error trials in a given session as a proxy for motivation, day of the week, and amount of brain movement within a session, but never identified any factors that explained the day-to-day variability in Monkey L. Future work will be needed to identify the causes of this across session variability, such as whether differences in fixational eye movements between the two monkeys might explain the decoder variability.

### Limitations of study
**Anatomical labeling of PPC subregions.** Our anatomical labeling relied on a standardized NHP atlas and prior literature[26,30] rather than subject-specific histology. Subtle boundary variations (e.g., LIPd-LIPv transitions) may exist and future studies combining histology and functional connectivity could refine these parcellations.

 

**Effect of contrast changes**. LIP in monkeys is known to be driven by motor as well as contrast-based changes over time[16] and visual changes in LIP are known to be organized topographically, i.e., LIP is retinotopically arranged[16]. In our task design, when the monkeys move their eyes toward the periphery of their visual field, the visual input changes dramatically, especially close to the border of the screen. To address the potential influence of contrast-based changes on CBV signals, we focused our GLM analyses on a time window at the end of the memory period, prior to saccade onset. This timing minimizes contamination from visual input changes associated with saccades, as saccade latencies were typically ~500 ms after the go cue. Additionally, significant decoding was observed prior to movement initiation (Fig. 4A, D), and similar decoding performance was reported in conditions without saccades in a previous study[25], further supporting that contrast changes are unlikely to be the primary driver of the observed CBV patterns. However, we cannot eliminate the effect of contrast changes on our observed results and this warrants investigation in future studies.

**Visual versus motor planning**. While our task design temporally separated visual cue processing and saccade preparation phases by having a cue that is short (400 ms) and a memory period that is long (> 4 sec), fUSI faces inherent constraints in fully dissociating these components. Neurovascular coupling introduces an ~2–4 sec smoothing effect on CBV signals[54–56], creating a temporal overlap between the visual and motor components. We saw the CBV increase early after the visual stimuli and then stay elevated throughout the memory period suggesting that we may have both an early visual response and a sustained movement preparation response. Future experiments with modified tasks or simultaneous electrophysiology will be needed to better disentangle the visual and motor components.

## Future studies

**Record along intraparietal sulcus axis**. In our study and most of the previous studies of LIP response fields, the topography changed along an anterior-posterior axis. Future studies could use 3D fUSI or align a 2D fUSI imaging plane along the intraparietal sulcus to acquire a larger anterior-posterior slice of LIP. This was not possible in the current animals due to the size of the ultrasound transducer and how our chamber was positioned off-axis relative to the intraparietal sulcus. This future study could significantly increase the longitudinal resolution compared to the current study and simultaneously improve effect sizes thanks to the ability to record anterior-posterior populations synchronously (in contrast to the current study that reconstructed these data over many sessions).

**Spatial autocorrelation of fUSI voxels**. Each voxel ( ~ 100 μm x ~ 100 μm x ~ 400 μm) contains ~65 neurons and 130 glia[57], whereas each 1–1.5 mm³ fMRI voxel contains ~16,000–24,000 neurons and 32,000–48,000 glia. This suggests that fUSI can detect very local activity within neural circuits, including from within different cortical layers. However, fUSI measures changes in CBV and neurovascular coupling is complex[58–60]. Although every neuron within the brain is positioned within 15 μm of a blood vessel[61], the contributions of different cell types are not sufficiently well understood to disentangle their contribution to the CBV signal. Additionally, neighboring voxels are supplied oxygen and nutrients by the same neurovasculature. This could contribute to an unknown extent to spatial autocorrelation of fUSI encoding (Figure S12), confounding our ability to precisely identify the size and spatial separation of tuned populations. Motion of our imaging plane and spatial smoothing further increases this spatial autocorrelation. There have been a variety of methods proposed for fMRI to handle the statistical consequences of spatial autocorrelation and calculate accurate statistical thresholds for cluster-wise inference[62–65]. However, to the best of our knowledge, no methods have been devised to separate the various contributors to the spatial autocorrelation, including correlated neuronal activity. Future experiments are required to disambiguate the contribution of correlated neurons versus other contributors to the size of neurovascular patches with similar tuning. Each voxel most likely contains neurons with a mixture of response fields with a bias towards specific response fields. Simultaneously recording fUSI signals, local field potentials, and single neurons will be crucial for understanding the response properties within individual voxels and patches of similarly tuned voxels. Recently, there has been development of new electrophysiology methods[19,20] designed to access mesoscale organization that would aid greatly in understanding the relationship between the mesoscopic neurovascular populations identified in this paper and the underlying neuronal activity of neurons distributed across the PPC.

**Cluster-wise inference**. In this paper, we employed voxel-wise FDR correction, but cluster-wise inference methods would likely offer improved control for false positives[63,66], especially given the spatial autocorrelation present in fUSI data and many other similarities of fUSI data to fMRI data. For example, the smaller dispersed clusters in Fig. 3A, D or Fig. 6A may be false positives. Future studies exploring the adaptation and validation of existing cluster-wise inference techniques[67–70] for fUSI, taking into account the unique characteristics of this modality, will be invaluable.

**Eccentricity axis**. Many studies have found a topography along an eccentricity axis with foveal and parafoveal targets being anterior of the peripheral targets representation[13–16,18]. In the current study, we presented our stimuli at a single eccentricity. Future studies could compare the representation of foveal, parafoveal, and peripheral targets within the LIP, potentially improving the field's understanding of how angular direction and eccentricity are spatially organized. Exploring foveal representation of saccades may also require a different experimental task than used in this study due to the difficulties associated with tracking and measuring small saccades[71]. Furthermore, implementing continuous fUSI decoding (as opposed to predicting discrete directions or eccentricities) could extend our findings by examining whether trial-to-trial variations in mesoscopic LIP activity predicts endpoint error distributions in saccadic eye movements. This would enable investigation of how mesoscopic LIP populations encode movement precision in addition to categorical direction or eccentricity.

**Directional tuning of cortical layers**. Ultra-high field (UHF) fMRI has enabled sub-millimeter voxel resolution and allowed researchers to study cortical layer-specific activity[72–74], especially with CBV-based fMRI[75,76]. Similar laminar analyses are possible with fUSI because it measures CBV and has higher spatiotemporal resolution and sensitivity than UHF fMRI. To date, only one fUSI study has begun to explore this possibility. Blaize et al. 2020[77] inferred cortical layer based upon cortical depth from the sulcus and found layer-specific ocular dominance within the deep visual cortex. In the present study, we observed broad activity within LIP that did not appear to respect any laminar boundaries within the cortex. In both monkeys, we detected some directionally specific activity within the shallower layers of MIP (Figs. 3D, 4C, F, 6A). This may reflect activity within superficial input layers. We could qualitatively estimate the boundary between white matter and gray matter based on the amount of organized mesovasculature observed in our vascular maps. However, the thickness of cortex varied within and across imaging planes, which prevented reliable estimates of cell layer based upon cortical depth. Future studies will be needed to better understand how to define cortical layers with fUSI, including studies to identify layer-specific properties detectable by ultrasound.

## Discussion summary

Here, we used fUSI to demonstrate that the posterior parietal cortex (PPC) contains mesoscopic populations of neurons tuned to different movement directions. This organization changed along an anterior-posterior gradient and remained stable across many months to years. These results unify previous findings that examined the topographic organization of LIP at the macroscopic (fMRI) and microscopic (electrophysiology) levels. In one monkey, we additionally found robust saccade-related activity within the medial parietal (MP) cortex, a parietal area that warrants further investigation. Using the methods established here for tracking the same populations across many months to years, it will be possible to apply brain-machine interfaces and other technologies that are advantaged by stable recordings across time.

## Methods

### Experimental model and subject details

All training, recording, surgical, and animal care procedures were approved by the California Institute of Technology Institutional Animal Care and Use Committee and complied with the Public Health Service Policy on the Humane Care and Use of Laboratory Animals. We worked with two rhesus macaque monkeys (*Macaca mulatta*; 14-years old, male, 14–17 kg). Monkey L participated in two previous fUSI experiments[24,25]. Monkey P participated in one previous fUSI experiment[25].

### General

**Animal preparation and implant.** We implanted a titanium headpost and custom square recording chamber on each monkey's skull under general anesthesia and sterile surgical conditions. We printed or machined a 24 × 24 mm (inner dimension) chamber using Onyx filament (Markforged) for Monkey L and a similar chamber made of PEEK for Monkey P. We placed the recording chamber over a craniectomy centered above the left intraparietal sulcus.

**Behavioral setup and task.** Each monkey sat head-fixed in a dark enclosed recording booth in custom-designed primate chairs facing an LCD screen ~30 cm away (HP L2025; 250 nits; 350:1 contrast ratio). We used a custom Python 2.7 software based upon PsychoPy[78] to control the behavioral task and visual stimuli. We tracked their left eye position using an infrared eyetracker at 500 Hz (EyeLink 1000, Ottawa, Canada). Eye position was recorded simultaneously with stimulus information for offline analysis.

Monkeys performed a memory-guided saccade task (Fig. 2A) where they fixated on a center dot of radius 2° (fixation state), maintained fixation while a peripheral cue (2° radius) was flashed for 400 ms in one of eight locations (20° eccentricity, equally spaced around a circle), continued to maintain fixation on the center dot (memory state), and finally made a saccade to the remembered cue location (movement state). If they correctly made a saccade to the cued location, the peripheral cue was redisplayed and the monkey maintained fixation on the peripheral target until the liquid reward (30% juice; 0.35 mL monkey L and 0.75 mL monkey P) was delivered. During each task state, the monkey had to keep their eyes within 6° of the fixation or target, i.e., the tolerance window was 6°. To avoid the monkeys predicting state transitions, we used variable durations sampled from a uniform distribution for each task state. In Monkey L, the fixation and memory phase were 4 ± 0.25 sec, the target hold phase was 0.75 ± 0.15 sec, and the intertrial interval (ITI) was 5 ± 1 sec. For Monkey P, the fixation and memory phase were 5 ± 1 sec, the target hold phase was 1 ± 0.5 sec, and the ITI was 8 ± 2 sec. Both monkeys were allowed up to 10 sec to make the saccade following the memory phase. For each task state, the duration was randomly drawn from the respective uniform distribution, meaning that the fixation and

memory lengths might be different in the same trial, e.g., fixation length of 4.15 sec and memory length of 3.9 sec for one trial in Monkey L.

### Functional ultrasound imaging

We used a programmable high-framerate ultrasound scanner (Vantage 256; Verasonics, Kirkland, WA) to drive the ultrasound transducer and collect pulse echo radiofrequency data. We used a custom plane-wave imaging sequence to acquire the 1 Hz Power Doppler images. We used a pulse repetition frequency of 7500 Hz with 5 evenly spaced tilted angles (−6° to 6°) with 3 accumulations to create one high-contrast compounded ultrasound image. We acquired the high-contrast compound images at 500 Hz and saved the images for offline construction of Power Doppler images. We constructed each Power Doppler image using 250 compound images acquired over 0.5 sec. To separate the blood echoes from background tissue motion, we used an SVD clutter filter[79]. For more details on the functional ultrasound imaging sequence and Power Doppler image formation, please see previous literature[24,80].

We used a 15.6 MHz ultrasound transducer (128-element miniaturized linear array probe, 100 µm pitch, Vermon, France). This transducer and imaging sequence provided us with a 12.8 mm (width) and 13–20 mm (height) field of view. The in-plane resolution was ~100 µm × 100 µm with a plane thickness of ~400 µm. During each recording session (Table S1), we placed the ultrasound transducer on the dura with sterile ultrasound gel and acquired images from a single imaging plane. We held the transducer using a 3D-printed slotted chamber plug that minimized motion of the transducer relative to the brain. The slots were spaced 1.66 mm apart. This slotted chamber plug allowed us to acquire specific imaging planes consistently across sessions. To help with later offline data concatenation, we acquired vascular maps using a single Power Doppler image and adjusted the transducer until the acquired vascular map closely matched a previously acquired vascular image for that chamber slot.

### Across session alignment and concatenation

We concatenated data across multiple sessions for each imaging plane. We first performed a semi-automated intensity-based rigid-body registration to align the vascular anatomy between sessions. As described above, during the acquisition, we minimized out-of-plane movement between sessions by matching each session's imaging plane to a previously acquired template image for each chamber slot. See Griggs and Norman et al. 2024[25] for more details. After aligning the anatomy, we concatenated all data together from a given coronal plane in temporal order, i.e., sessions recorded on later days are later in the concatenated dataset. No rescaling of data was performed during concatenation.

### 3D visualization

We used MATLAB to export the vascular images to NIFTI format. We used Napari[81], the 'napari-nifti' plugin[82], and custom Python code to visualize the 3D reconstruction and save as images. The images were combined to form a video using Da Vinci Resolve 17.4.4 Build 7 (Blackmagic Design). For Fig. 1B–D, we used Blender v3.5.0 on MacOS and v2.92.0 on Microsoft Windows to create representations of the cortex and different recording technologies.

### Quantification and statistical analysis

Unless reported otherwise, summary statistics are reported as mean ± SEM.

**Anatomy.** All labels for PPC subregions, such as LIP, VIP, MIP, Area 5, Area 7, and MP, were anatomically defined based upon a rhesus macaque neuroanatomy atlas[26]. This avoided circular logic in using

functionally defined boundaries to assess how much activity was within a region.

**General linear model (GLM).** We applied several pre-processing steps before creating the GLM to explain the data. We first applied a Gaussian spatial filter (FWHM – 100 µm). We then applied a voxel-wise high-pass temporal filter (1/128 Hz) to remove low-frequency drift. We finally used grand mean scaling to scale each voxel's intensity to a common scale[83,84]. To build the general linear model, we convolved the regressors of interest with a hemodynamic response function (HRF). We used a single gamma function with a time constant ($\tau$) of 1 sec, a pure delay ($\delta$) of 1 sec, and a phase delay ($n$) of 3 sec based upon a previous monkey event-related fMRI study[27]. The regressors of interest were fixation period, memory period, movement period, and reward delivery. For the memory and movement periods, we used separate regressors for each direction. We then fit the GLM model using the convolved regressors and scaled fUSI data. We used an F-test to identify voxels that had a statistically significant difference to the eight directions during the memory period.

**Multiple comparison correction.** For all voxel-wise p-values used and reported, we used false-discovery rate correction (FDR) to correct for the simultaneous multiple comparisons. This was implemented using MATLAB's 'mafdr' function.

**Preferred direction.** We used a center-of-mass approach to find the preferred tuning of each voxel. For each voxel, we first calculated the Cohen's d measure of effect size by comparing the response in the last second of the memory period (single timepoint within ±0.5 sec of memory end) to the baseline (−1 to 1 sec relative to cue onset). Positive (negative) values correspond to increase (decrease) in CBV from the baseline. This gave us a standardized measure of response strength for each direction. We then scaled the peak response at each voxel to be 1. We then found the centroid for each voxel, which provided both a direction and magnitude. The direction represents the peak tuning direction while the magnitude represents the strength of that tuning. A value close to zero means no tuning while a value close to 1 means highly tuned to a specific direction. This method minimizes assumptions about shape of the response field, such as whether it is Gaussian. We then smoothed the resulting statistical map using a pillbox spatial filter (1-voxel radius).

**Statistical measures of data distribution.** We calculated the laterality index based upon the formula used in previous papers[27,28].

$$\text{Laterality index} = \frac{\text{Response}_{\text{contra}} - \text{Response}_{\text{ipsi}}}{|\text{Response}_{\text{contra}}| + |\text{Response}_{\text{ipsi}}|} \quad (1)$$

We calculated the circular standard deviation, angular kurtosis, and angular skewness using the Circular Statistics Toolbox[85] (available at https://github.com/circstat/circstat-matlab).

**Across-session statistical analyses.** For each coronal plane, we would concatenate all sessions recorded from that plane. We then performed the same methods for statistical analyses as described above. We examined the last timepoint when calculating the preferred tuning and other statistical measures of data distribution at each voxel. For the multiple comparison correction using FDR, we applied the correction across all voxels in all coronal planes simultaneously rather than just applying within a single coronal plane.

**Within-session decoding analysis**
Decoding intended movement direction on a single trial basis had five steps: 1) aligning the fUSI data and behavioral data, 2) preprocessing, 3)

selecting data to analyze, 4) dimensionality reduction and class separation, and 5) cross-validation.

First, we created the behavioral labels by temporally aligning the fUSI data with the behavioral data. We could then label each fUSI timepoint with its corresponding task state and movement direction.

Second, we preprocessed the data by applying several operations. The first operation was motion correction. We used NoRMCorre to perform rigid registration between all the Power Doppler images in a session[86]. We then applied temporal detrending (50 timepoints) and a pillbox spatial filter (2-voxel radius) to each Power Doppler image.

Third, we would then select what spatial and temporal portions of the data to use in the decoder model. We always used the entire image where each voxel is a single feature. We used the entire image rather than decoding from specific anatomically-defined subregions to avoid introducing any bias by our approximation of anatomical boundaries. By including the entire image, we did not actively bias the decoder to rely upon specific regions but rather let the data itself tell us the most important features. We used a dynamic time window. At each timepoint before the cue, we used all timepoints since the start of the trial. For example, to test our ability to decode at 3 sec after the trial start, we used the fUS images at 0, 1, 2, and 3 sec. At each timepoint after the cue, we used all previous timepoints after the cue in the trial. For example, to test our ability to decode at 2 sec after cue onset, we concatenated the data from 0, 1, and 2 sec after the cue. We treated these timepoints as additional features in the decoder model. In other words, the input to our decoder model had N*T features, where N is the number of voxels in a single Power Doppler image and T is the number of timepoints.

Fourth, we split the data into train and test folds according to a leave-one-out or 10-fold cross-validation scheme. For the test sets, we stripped the behavioral labels. We then scaled the train and test splits by applying a z-score operation fit to the train data. We used the entire image for our features, i.e., each voxel's activity was a single feature. To train the linear decoder on the training data, we used principal component analysis (PCA) for dimensionality reduction and linear discriminant analysis (LDA) for class separation. For the PCA, we kept 95% of the variance. The number of components kept depended on the training dataset, typically varying between 330–450 components (Figure S13). For the LDA, we used MATLAB's 'fitcdiscr' function with default parameters. We used a multicoder approach where the horizontal (left, center, or right) and vertical components (down, center, or up) were separately predicted and combined to form the final prediction. As a result of this separate decoding of horizontal and vertical movement components, center predictions are possible (horizontal−center and vertical−center) despite this not being one of the eight possible peripheral target locations. We then calculated the percent correct and absolute angular error for each sample in the test data.

Fifth, we then repeated the model training and testing for each consecutive fold of data. We finally found the mean accuracy metrics across all the folds, i.e., mean accuracy and mean angular error. To correct for testing the performance at every trial timepoint, we used a Bonferroni correction.

We used a 1-sided binomial test to calculate the p-values associated with the percent correct results and used a permutation test with 100,000 replicates to calculate the p-values associated with the angular error results. Although the multi-coder architecture can generate 9 possible classes, due to class imbalances in the training sets, the empiric chance level of the decoder accuracy was between [1/8, 1/9]. We therefore picked the more conservative threshold of 1/8 chance level for the binomial test. For the permutation test, each replicate was created by sequentially drawing X directional guesses from a uniform distribution of the eight possible directions, where X is the number of trials in the session. We then calculated the 1-sided p-value of each of

our results by finding how many of the replicates were less than our observed mean angular error.

See Norman and Maresca et al. 2021[24] and Griggs and Norman et al. 2024[25] for more details on these methods.

**Visualization of LDA weights.** To visualize the LDA boundaries between pairwise classes, we projected the boundaries for the horizontal and vertical LDA decoders through the inverse PCA transform. For each class-class pair, we obtained the LDA boundary weights (constant and linear components) which has a single weight for each principal component. We discarded the constant components and weighted each principal component by the linear component of the LDA weights and added all the principal components together. We then multiplied the resulting horizontal and vertical decoder weight images together. To keep a similar scale between all pairwise class combinations, we took the square root of the resulting absolute value for pairwise classes where the vertical or horizontal class was different for both classes and then multiplied by the sign of the original value to preserve the sign prior to the square root. We performed this square root operation because otherwise pairwise classes where the vertical or horizontal classes were both identical would be approximately a square root larger. We finally applied a threshold to each image where we kept the 10% most extreme values (positive or negative).

### Across-session decoding analysis
To test whether we could use a decoder trained on a separate session's data to decode movement intent in a different session, we applied the same steps as for the within-session decoding analysis with two differences. First, the training set was all the data from a specific session and the testing set was all the data from a different specific session. Second, to assess performance within the same train and test session, we used 10-fold cross-validation instead of leave-one-out cross-validation. The later sessions' data (after March 25, 2022; Table S2) were used in a previous publication and were acquired at a 2 Hz imaging rate with slightly different acquisition parameters. See Griggs and Norman et al. 2024[25] for more details about the acquisition of these data. For the across-session decoding analysis, we down-sampled this 2 Hz data to 1 Hz to allow us to easily compare the same trial timepoints between the two sets of data.

### Image similarity
We compared the pairwise similarity of vascular images from different sessions by using the complex wavelet structural similarity index measure (CW-SSIM). The CW-SSIM quantifies the similarity of two images, where 0 is dissimilar and 1 is the same image[29]. We used the CW-SSIM over other forms of SSIM because it is more flexible in incorporating variations in image resolution, luminance change, contrast change, rotations, and translations. We used an implementation freely available from the MATLAB Central File Exchange[87] with 4 levels and 16 orientations.

### Searchlight analysis
We defined a circular region of interest (ROI) and, using only the voxels within the ROI, we performed the within-session decoding analysis using 10-fold cross-validation. We assigned that ROI's percent correct and angular error metrics to the center voxel. We then repeated this across the entire image, such that each image voxel is the center of one ROI. To visualize the results, we overlaid the performance metric (angular error or percent correct) onto a vascular map and kept up to the 10% most significant voxels. As part of this searchlight analysis, we ignored activity within the sulcal fold or activity on the other side of the sulcal fold. To do this, we defined the boundaries of the sulcal folds using a custom GUI in MATLAB and only used voxels on the same side of the sulcal fold as the searchlight center. This is similar in principle to the cortical surface-based searchlight decoding developed for fMRI[88].

### GLM and searchlight similarity
To assess the similarity between the GLM and searchlight results, we computed the overlap of the statistical masks for both analyses. To maintain similarity with the GLM analysis, we used a modified searchlight analysis where analyzed the decoding performance at the end of the movement period (as opposed to end of trial), did not ignore activity within the sulcal folds, and allowed activity on both sides of the sulcal folds to be in the same searchlight window. After this modified searchlight analysis, we created threshold masks for both the GLM and searchlight results using FDR-corrected p-values and a threshold of $p \le 0.001$. We then computed the Dice-Sørenson coefficient (DSC) to measure the similarity between the two threshold masks, where A and B are the two masks.

$$DSC = \frac{2|A \cap B|}{|A| + |B|} \qquad (2)$$

We performed this analysis for every session. We excluded one session where there were no voxels in either the GLM or searchlight masks, i.e., no voxels showed significant directional encoding in the GLM or searchlight analyses.

### Spatial autocorrelation
For every voxel in the image, we examined voxels at different distances from the seed voxel. For each distance tested, we identified voxels that were between [max(0, i-0.1) mm, i mm] away. We then performed Pearson linear correlation between these identified voxels and the seed voxel. We then assigned the mean correlation to the seed voxel. To calculate the mean correlation for each distance, we took the mean and standard deviation across the entire image.

### Reporting summary
Further information on research design is available in the Nature Portfolio Reporting Summary linked to this article.

## Data availability
The processed functional ultrasound neuroimaging data generated in this study have been deposited in the CaltechDATA database under the accession code (https://doi.org/10.22002/p5jan-02r60)[89]. Source data for figures are provided with this paper. Source data are provided with this paper.

## Code availability
Code used to generate key figures and results is available at (https://github.com/wsgriggs2/PPC_directional_tuning) and archived[90] on Zenodo at (https://doi.org/10.5281/zenodo.15122174).

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

## Acknowledgements

We thank Kelsie Pejsa for assistance with animal care, surgeries, and training. We thank Tyson Aflalo, Claire Rabut, Ernesto Criado Hidalgo, Di Wu, and Geeling Chau for their helpful discussions and insights. W.S.G was supported by an NEI F30 (NEI F30 EY032799), the Josephine de Karman Fellowship, and the UCLA-Caltech MSTP (NIGMS T32 GM008042). S.L.N. was supported by the Della Martin Foundation. This research was supported by the National Institute of Health BRAIN Initiative (grant 1R01NS123663-01 to R.A.A., M.G.S., and M.T.), the T&C Chen Brain-Machine Interface Center (R.A.A.), and the Boswell Foundation (R.A.A.).

## Author contributions

W.S.G., S.L.N., V.C., M.G.S., and R.A.A. conceived the study; W.S.G. trained the NHPs and acquired the data; W.S.G. and S.L.N. performed the data processing and analysis; W.S.G. drafted the manuscript with substantial contributions from S.L.N., M.G.S., and R.A.A., and all authors edited and approved the final version of the manuscript. V.C., C.L., M.T., M.G.S., and R.A.A. supervised the research.

## Competing interests

M.T. is a co-founder and shareholder of Iconeus company. M.T. is the co-inventor of several patents in the field of neurofunctional ultrasound and ultrafast ultrasound. M.T. does not have any other financial conflict of interest, nor any non-financial conflict of interests. All the other authors do not have any financial or non-financial conflicts of interest.
