## [Transparent Peer Review file · Nature Communications]

Functional ultrasound neuroimaging reveals mesoscopic organization of saccades in the lateral intraparietal area

Corresponding Author: Dr Whitney Griggs

Version 0:

Reviewer comments:

Reviewer #1

(Remarks to the Author)

This is a very interesting paper, investigating saccade direction tuning in LIP in non-human primates. The results clarify previous reports showing contradicting findings, and shed light on the mechanisms of movement direction encoding in non-human primates.

To achieve this, the authors adopt a novel technique to image cerebral blood volume over time with an unprecedented level of detail, and the images of cortical vasculature are insightful as well as beautiful.

I have several questions and clarifications that I believe should be addressed before granting publication.

Potential contrast-based responses in the memory guided saccade task.

LIP in non-human primates is known to be driven by motor as well as by contrast-based changes over time (Arcaro et al., 2011). Moreover, visual responses in LIP are organized topographically (i.e. LIP is retinotopically arranged, Arcaro et al., 2011). When monkeys move their eyes towards the periphery of the visual field, the visual input changes dramatically, especially close to the border of the screen.

Given the changes in visual input associated with each saccade, it is fair to ask whether potential contrast-based changes might have contributed to the measured CBV changes in the study, especially considering that the analyses focus on a time-window when the monkey was performing the saccade.

Also, it would be important to know more details about the task setting: whether monkeys were performing the task in complete darkness, and the specs of the LCD monitor used (e.g. luminance). This would give an idea of the contrast change between the border of the LCD and the background.

ROI definition:

From the text it is not clear whether LIP was defined based on functional criteria (e.g. response to performing saccades), or structural criteria. Please clarify.

Figure 2, panel A. please report the duration in seconds of each phase of the trial (fixation, cue, memory, saccade, hold, reward).

Figure 3

The ROI labels (1,2,3) on panel A are quite hard to read, please make them more conspicuous.

The interpretation of Cohen's d is not clear. In particular, it is not clear how to interpret negative Cohen's d values, do they represent stronger activation during baseline than the memory period, or do they represent locations where the memory – baseline response was simply weaker? It would be beneficial to clarify this point in the figure caption.

Reviewer #2

(Remarks to the Author)

Griggs et al. used functional ultrasound imaging to investigate the mesoscopic layout of lateral intraparietal area (LIP) during a memory-guided saccade task. The task required the two monkeys to fixate centrally and make outward memory-guided

saccades. The authors tested for (directional) selectivity of subregions in LIP and adjacent areas (e.g., MIP, MP) to determine if the topographic layout extends to neighbouring regions as well. They found such selectivity, as well as lateralized responses in LIP, with additional saccade-related directional responses in adjacent areas. They also tested whether this organization lasts on a scale of months to years, which they were able to find.

This manuscript was a pleasure and fascinating to read. It advances the field in the use of fUSI to better understand saccade-related neural activity at a scale that bridges the gap between single-unit recordings and fMRI spatial resolution. In addition, it provides longitudinal information about the oculomotor organization, which is also an important addition to the field.

My minor comments relate to the mentioning of specific dates - instead, please specify the duration of time that has passed when describing the temporal organization results.

UHF fMRI also needs to be abbreviated on line 485.

Reviewer #3

(Remarks to the Author)

Griggs et al. use functional ultrasound imaging (fUSI) to map subregions of the lateral intraparietal cortex (LIP) based on tuning to saccade direction. Taking advantage of the high spatial resolution and wide field of view the authors show heterogeneous patches with tuning to distinct saccade direction. These results show the utility of fUSI as a complementary method to more established imaging techniques commonly used in neuroscience. The patchy tuning of LIP further adds valuable insight to the functional organization of this brain area and to the debate on the extent of retinotopic organization in higher-level visual association areas. The demonstration of the increased stability of the decoder for up to 900 days, by introduction of motion-correction, is a methodological advancement over previous work by the authors.

However, some claims of the paper are not strongly supported by the analysis. The organization of LIP into small patches tuned to saccade direction could partly be a spurious result of the statistical method. That the tuning of these patches is stable over a long period of time is suggested by the data, but not clearly demonstrated. Lastly, the anterior-posterior gradient of movement direction encoding in the LIP has not been statistically tested. The analysis of the paper can be improved as outlined below to support these claims.

Major comments

1. The statistical analysis of responsive patches is prone to false positives. As described in the discussion (lines 466-484) spatial autocorrelation exists in fUSI, similar to other imaging modalities. Voxel-wise FDR correction can introduce a high number of false positives in the presence of spatial autocorrelation (Friston et al. 1991; Chumbley and Friston 2009). Cluster-wise inference is not limited to fMRI analysis and also commonly used with other imaging modalities, such as MEG or EEG (Maris and Oostenveld 2007). In this work, smaller dispersed clusters as in Figures 3A,D or 6A could potentially be false positives. Cluster-wise inference could mitigate this issue and is a better control for false positives. This would support the claim that the tuning to saccade direction is organized in heterogeneous patches (lines 26-27).
2. The claim that the tuning of subregions to specific movement directions are consistent across time, is not clearly supported by the results (lines 27-28; 312-313). While the decoder is stable over a long time period, it is not clear that the decoder relies on the same spatial locations that are most strongly tuned to movement directions. If the reasoning relies on the statement in line 179-181 that the maps of most informative locations for the decoder and GLM analysis overlap, quantifying this effect would bolster the claim. This could be done by computing image overlap, for example with the Dice-Sørensen coefficient. This analysis would still leave some uncertainty about the weights of the decoder used in the stability analysis. Visualization of the PCA weights of the decoder could also support the claim that the decoder relies on subregions that are tuned to specific movement directions.
3. One of the main claims of the paper, that posterior LIP encodes more contralateral upward movement, while anterior LIP encodes more contralateral downward movements (lines 28-30, 256-257, 311-312) is not supported by a formal statistical analysis. A possible test could be a correlation of the median preferred direction across significant voxels within each plane, with the anterior-posterior position of the planes. This would be a summary of Figure 6. Figure 6A and 6B suggest that this correlation might reach significance for Monkey P, however, data for Monkey L looks less convincing.
4. In addition, the literature discussed to support the claim of an anterior-posterior tuning gradient is not accurately described. References 10-13 and 33 (lines 332-335, 360-361) are cited as support for the notion that anterior-posterior gradients encode eye movement directions (Ben Hamed et al. 2001; Wardak, Olivier, and Duhamel 2004; Patel et al. 2010; Arcaro et al. 2011). However, these references do not report gradients of eye movement direction, but gradients of location in the visual field. Most of the experiments in these studies were performed during fixation (Wardak, Olivier, and Duhamel 2004; Patel et al. 2010; Arcaro et al. 2011). Ben Hamed, 2001 report an overlap of populations of visually and saccade responsive neurons, but focus their analysis on visually responsive neurons (Ben Hamed et al. 2001). Blatt, 1990 also map visual fields, not saccade direction. This paper reports that the anterior LIP represents the lower visual field, and posterior LIP the upper visual field (Blatt, Andersen, and Stoner 1990) (lines 338-342), opposite to the direction described in the discussion (lines 336-337). Savaki, 2011 report tuning to saccade direction, however, an anterior-posterior gradient is not obvious in their results (Savaki et al. 2010) (lines 336-337). Overall, it is not clear that LIP subpopulations that respond to different locations of the visual field would respond to corresponding saccade directions (see discussion in (Arcaro et al. 2011); or (Chen et al.

2016)). This literature does not discount the findings of the current paper, but needs to be discussed accurately. In fact, if the anterior-posterior gradient can be convincingly shown this would support the novelty of this work.

Minor comments

1. Line 24: It is not immediately clear that the LIP is part of the PPC.
2. Figure 2A: Adding the time duration of different parts of trials would be helpful.
3. Lines 118, 127, 196: It would be useful to clarify the example session that was used for the analysis. Is this first session that is used in the analysis of the stability of the decoder?
4. Line 168: remove "around"
5. Line 639: How many PCA components were kept?

References

- Arcaro, Michael J., Mark A. Pinsk, Xin Li, and Sabine Kastner. 2011. "Visuotopic Organization of Macaque Posterior Parietal Cortex: A Functional Magnetic Resonance Imaging Study." *Journal of Neuroscience* 31 (6): 2064–78. <https://doi.org/10.1523/JNEUROSCI.3334-10.2011>.
- Ben Hamed, S., J.-R. Duhamel, F. Bremmer, and W. Graf. 2001. "Representation of the Visual Field in the Lateral Intraparietal Area of Macaque Monkeys: A Quantitative Receptive Field Analysis." *Experimental Brain Research* 140 (2): 127–44. <https://doi.org/10.1007/s002210100785>.
- Blatt, Gene J., Richard A. Andersen, and Gene R. Stoner. 1990. "Visual Receptive Field Organization and Cortico-Cortical Connections of the Lateral Intraparietal Area (Area LIP) in the Macaque." *Journal of Comparative Neurology* 299 (4): 421–45. <https://doi.org/10.1002/cne.902990404>.
- Chen, Mo, Bing Li, Jing Guang, Linyu Wei, Si Wu, Yu Liu, and Mingsha Zhang. 2016. "Two Subdivisions of Macaque LIP Process Visual-Oculomotor Information Differently." *Proceedings of the National Academy of Sciences* 113 (41): E6263–70. <https://doi.org/10.1073/pnas.1605879113>.
- Chumbley, J, and K Friston. 2009. "False Discovery Rate Revisited: FDR and Topological Inference Using Gaussian Random Fields." *NeuroImage* 44 (1): 62–70. <https://doi.org/10.1016/j.neuroimage.2008.05.021>.
- Friston, K. J., C. D. Frith, P. F. Liddle, and R. S. J. Frackowiak. 1991. "Comparing Functional (PET) Images: The Assessment of Significant Change." *Journal of Cerebral Blood Flow & Metabolism* 11 (4): 690–99. <https://doi.org/10.1038/jcbfm.1991.122>.
- Maris, Eric, and Robert Oostenveld. 2007. "Nonparametric Statistical Testing of EEG- and MEG-Data." *Journal of Neuroscience Methods* 164 (1): 177–90. <https://doi.org/10.1016/j.jneumeth.2007.03.024>.
- Patel, Gaurav H., Gordon L. Shulman, Justin T. Baker, Erbil Akbudak, Abraham Z. Snyder, Lawrence H. Snyder, and Maurizio Corbetta. 2010. "Topographic Organization of Macaque Area LIP." *Proceedings of the National Academy of Sciences* 107 (10): 4728–33. <https://doi.org/10.1073/pnas.0908092107>.
- Savaki, Helen E., Georgia G. Gregoriou, Sophia Bakola, Vassilis Raos, and Adonis K. Moschovakis. 2010. "The Place Code of Saccade Metrics in the Lateral Bank of the Intraparietal Sulcus." *Journal of Neuroscience* 30 (3): 1118–27. <https://doi.org/10.1523/JNEUROSCI.2268-09.2010>.
- Wardak, Claire, Etienne Olivier, and Jean-René Duhamel. 2004. "A Deficit in Covert Attention after Parietal Cortex Inactivation in the Monkey." *Neuron* 42 (3): 501–8. [https://doi.org/10.1016/S0896-6273\(04\)00185-0](https://doi.org/10.1016/S0896-6273(04)00185-0).

Reviewer #4

(Remarks to the Author)

I've attached a word document of the review that has formatting that might be easier to digest. Apologies for the inconvenience.

Griggs and colleagues used functional ultrasound neuroimaging to investigate the mesoscopic structure of directional selectivity within the posterior parietal cortex while two monkeys performed a memory guided saccade task. They used a general linear model to detect voxels with directional tuning and found a heterogeneous representation of saccade direction throughout PPC. The lateral intraparietal sulcus contributed the most voxels to the analyses. To evaluate how informative PPC's representation of upcoming saccade direction was, they first performed dimensionality reduction (PCA) on the full fUSI image, and then used components that explained 95% of the variance as inputs into a classifier (linear discriminant analysis) to predict the saccade direction on held out trials. They decoded upcoming saccade direction well above chance. The authors then applied similar decoding methods across sessions to investigate the stability of the representation of direction selectivity across sessions and found significant classification of saccade targets between sessions collected days to years apart. They further described this representation by showing that LIP preferred contralateral targets and had a downward to upward preference from anterior to posterior LIP. They also found direction selectivity in one monkey in MIP and MP. Overall, these results show a heterogeneous but coarsely structured representation of saccade target direction within the LIP and PPC generally.

The topographic organization of LIP's sensitivity to saccade direction has been investigated by numerous labs over the past three decades. The novel contribution of this study is the use of functional ultrasound imaging. Though fUSI offers spatial sampling well suited for probing mesoscopic structure, the findings seem largely confirmatory of prior reports (e.g. contralateral bias and patchiness – see citations of authors in discussion). The data presented here is perhaps most useful for the ongoing debate regarding the directional tuning across the anterior-posterior axis of LIP. However, I find the data supporting the interpretations of an anterior-posterior gradient of downward to upward unconvincing; it seems to only be present in one of the two monkeys (monkey P). My reluctance to accept this conclusion stems predominantly from a lack of quantification of the gradient across the two monkeys. This critique extends to other results in the study. I go into more details in my concerns below:

Major Concerns:

1. Although Griggs and colleagues use standard analytical tools to quantify their effects, the presented results are largely example sessions with lack quantifications across sessions. There are also no quantifications for some conclusions. The strength of the conclusions are limited by the lack of summary data or specific quantifications of specific effects. Here is each instance:

a. Claim: PPC contains multiple distinct directionally tuned mesoscopic populations (Figure 3) Evidence: The authors present tuning curves from two ROIs from one imaging plane in one session for each monkey. They also present an effect size heat map from that one session for each direction directions.

What is Mesoscale: It's unclear how these ROI were selected (not the same way as the searchlight analysis presented later). Why was a square (vs. circle as in searchlight) of that size chosen? What was the size of the square? Is there something intrinsic about this shape/size that makes a population mesoscopic?

Majority of data not discussed: Two planes (one from each monkey) out of 226 planes (7 planes * 18 sessions in monkey p + 5 planes*20 sessions in monkey L) were presented. I suspect much of the other data would also show direction selectivity.

Tuning Quantification: Importantly, the authors do not present any of the results concerning the strength of tuning of these mesoscopic populations nor for each voxel (it's hard to determine how each voxel changes across the 8 directions from the heat map for me). In the 'preferred direction' section of the methods there is mention of finding the centroid of each voxel which gives a measure of tuning strength, but I do not see where any of this data is reported. For the plotted mesoscopic tuning curves in figure 3 there is no measure of how tuned they are or the diversity of tuning curve widths across the data (in other mesoscopic populations). There are several metrics for describing tuning curve strength (e.g. full width at half max, equivalent rectangular receptive fields (Lee & Middlebrooks 2011), ANOVA f-statistic across directions, among others) but none are implemented here. Summary plots of these quantifications across all data would strongly support the claim.

b. Claim: Single-trial decoding of eight intended movement directions with high accuracy (Fig. 4)

Evidence: The authors present the percent of correctly decoded trials, the angular error in decoded direction, the confusion matrix, and the contribution from each voxel in the image from a single session in each monkey.

Majority of data not discussed: The reported results do not include 36 of 38 sessions. Although these example sessions are compelling – especially for decoding performance on a vasculature-based metric – I was struck how variable the decoding performance was across sessions (which I was only aware of because of the across session decoding analyses and supplementary figures). The authors do report some summary data concerning these analyses but it's only available as the 0 days between train and test data in the across session decoding panel (figure 5c and 5F). They also report the mean angular error but only the mean and no quantification of the variance in text. It would be quite simple to present a histogram of the percent of trials correctly decoded and the angular error across sessions. Additionally, you could add a figure like 4a and 4d that present each session or the average performance and standard error of the mean across sessions. Evaluation of how well this imaging method can be used by a classifier to decode saccade direction requires presentation of all sessions.

c. Claim: PPC stably encodes movement direction across many months to years (Figure 5)

Evidence: The authors present an example of an across session decoder for one session per monkey, heat maps showing performance across a select subsample of their data, and the average angular error as a function of time between all recording sessions.

Performance as time between recording sessions: The authors present the bulk of their data in this figure. However, a similar plot to panel 5c & 5f would but for percent correct would be a useful plot to evaluate every session instead of the select subsample presented in panels 5b and 5e.

d. Claim: Patchiness in direction selectivity – the direction between patches with different directional tuning is spatially sharp with some transition zones less than 500 microns (Fig. 6a?)

Evidence: The authors present heat maps of the preferred direction of each voxel across all imaging planes from a single session from each monkey.

No quantification: In addition to only presenting a single session per monkey, the quantification of transition between patches seems to be missing. I don't know if I see only distinct and sharp patches in the data, for example in panel 3 mm of monkey P – there seems to be an intermingling of blue/green voxels around the distinct patch of 180° and more dorsal to that an intermingling of green and yellow voxels. I also do not think looking at preferred direction alone is sufficient to judge patchiness but rather the strength of the tuning (which the authors should have access to since they describe a magnitude measure in the 'preferred direction' section of the methods). A high and uniform magnitude across all tuned voxels would indicate that any transition in color is indeed strong but if tuning strength is weakest between patches and strongest in the middle of patches then it would indicate smoothish transitions in tuning across PPC. The searchlight results may also support patchiness or smoothness; you may expect different distributions of voxel importance from voxels around the transition zones versus in the center of a patch. I will note that it may be more challenging to evaluate this point because the images are somewhat pixelated in the word document, and I am color blind so evaluating color maps is challenging to some extent.

e. Claim: Polar direction is topographically organized along anterior-posterior axis of LIP (Fig. 6)

Evidence: The authors present the preferred direction of each voxel across all imaging planes from a single session from each monkey as well as plotting preferred direction as a function of anterior-posterior axis from all tuned voxels from each plane from a single session (?)

No quantification of gradient: Although in the presented data it seems that there is a change in preferred direction across the anterior posterior axis in monkey P – It would be useful to evaluate this claim with a quantification of the gradient. How much does preference change as a function of anterior-posterior distance. Although incorrect because directional tuning is circular data, I would be happy if a line was used to fit the data to quantify change in direction as a function of anterior-posterior position.

Majority of data missing: The authors present 2 of 38 sessions. It would be of useful to evaluate how consistent the gradient is across all sessions.

2. I am reluctant to conclude that there is an anterior-posterior gradient given the data presented in figure six. Although this is

in part due to the lack of a quantification of the gradient (as mentioned above), the effect seems limited to one monkey (monkey P). The data from this session for monkey L looks flat. This could be alleviated if the authors quantify the gradient and find a significant change in preferred direction as a function of anterior-posterior axis. It may be true that there is a consistent mapping across animals and the explanation for the reported differences are due to sampling differences like the authors suggest. Alternatively, there may not be consistent mappings across animals. For example, in Patel et. al. 2010 (ref. # 8) there seems to be 1 of the 4 imaged LIPs with altered mappings (Fig 3, monkey Y, right hemisphere).

Minor concerns:

1. Abstract:

a. Line 28-29: "A rough topography emerged where anterior LIP represented more contralateral and upward movements and posterior LIP represented more contralateral downward movements" – This is opposite of the reported anterior-posterior gradient.

2. Introduction:

a. It may be worth noting some modern electrophysiology methods designed to access mesoscale organization: Dotson, Hoffman, Goodell & Gray, *Neuron*, 2017 and Tiechert et. Al. *bioRxiv* 2024 (<https://doi.org/10.1101/2024.05.13.593946>). Although these have less continuous spatial sampling they have better temporal resolution. Though maybe this would be a discussion point.

b. Fig. 1 Panel C: Are the spaces between the planes just for visualization or does fUSI have large gaps between planes? Is this related to the 1.66 mm between transducer slots?

3. Methods:

a. Memory guided saccade task:

i. What was the size of the fixation dot, the target, the fixation window, and the target window?

ii. Did the visual cue always appear in the middle of the fixation/memory phase? For example, if the fix/memory phase for 1 trial was 5 seconds would it appear at 2.5 seconds and if in another trial the fix/memory period was 4 seconds the cue would appear at 2 seconds? Or was there some other distribution of cue presentation times?

iii. Why were the intertrial intervals different between the two monkeys – was there something intrinsically different about monkey P's vasculature that they needed 2-4 more seconds between trials? Could the longer intertrial interval contribute to the better performance of decoding in Monkey P?

b. General Linear Model (GLM)

i. I'm unfamiliar with vasculature-based methodologies, but convolution of a time-series can lead to leakage of signal into earlier bins. Is there potential for movement related signal to leak into the memory time bin? I think this is of little concern given the sluggishness of the blood volume responses.

c. Preferred Direction

i. What was the time-bin used at the end of the memory period? Was it the last second of the memory period?

d. Within-Session Decoding

i. Line 618 Typo: fUS > fUSI

ii. How many repeats of the leave-one-out validation were done? Or was every trial left out once?

4. Results:

a. General

i. Line 96-97: "high spatial resolution 100-micron x 100 micron in-plane" – I found this to be a bit misleading. It is important to know the voxels used in all analysis will be done on are 100x100x400 microns. Especially for interpretation of voxel-by-voxel heat maps and anterior-posterior gradients.

ii. Fig. 2a: The dashed circle isn't defined in the legend. I assume it is the target window.

iii. Fig. 2c: Medial Parietal (MP) is not defined in the figure legend.

b. Figure 3:

i. Panel B/E: Does 'Move' correspond to the go cue or the actual start of the saccade?

ii. Which time bin is used for the analysis? From the methods I thought it was the last bin of the memory phase, but "analysis" appears aligned with "Move" on the x-axis. It would be nice to have as shaded area of the plot for the analysis bin (like for the baseline).

iii. I found it interesting there was some ROIs that had much higher baseline volume. It made me think about the CBV response – are there non-linearities in the CBV response? Are there saturations if there's a significantly larger visual response to a saccade target that could occlude an additional movement response?

c. How consistent is the directional tuning within a session?

i. I'm interested in the components that explain the most variance of the CBV response across the PPC. How many components are needed to reach 95% variance? Also do the top PCs correspond to direction tuning? A quick way to look at this would be to correlate each voxel's directional tuning strength to their weight onto each component. It would be interesting to know if saccade direction explained a large amount of variance across all components or specific components.

ii. Although potentially out of scope for this study, an interesting future direction for this data would be to use a continuous decoder to see if fUSI could predict single-trial variance in movement direction (E.g. decoded endpoint error correlates with actual saccade endpoint error on single trials).

iii. Line 179-181: "LIP contained the most informative..." – It may be a nice addition to this analysis to correlate the GLM weight for each voxel with the angular error from the searchlight analysis.

d. Figure 4:

i. 'Decode movement direction': With the sluggishness of the CBV response is it possible to differentiate the visual response to the saccade target versus the movement preparation for the saccade? Are you decoding the saccade target location or are you decoding the saccade direction?

ii. I know the circle corresponds to the 'center' position which emerged from the multicoder approach, but it's not listed anywhere in the main text or figure legend. It turns out to be a nice internal control that 'center' never seems to be decoded.

1. Is the chance listed in panel A and D $1/8$ (the directions) or $1/9$ (the directions + center)?
- iii. What time bin corresponds to the decoder for the confusion matrix?
- iv. What time bin corresponds to the searchlight analysis? Additionally, it would be interesting to look at what timepoint the voxels are most informative.
- e. Are the mesoscopic populations stable across multiple days? And Figure 5:
 - i. It is impressive to get above chance decoding 900 days later.
 - ii. I found the explanation that the implant likely shifted and correlating image similarity with accuracy and angular error convincing.
 - iii. This section is when I started to realize that, especially for monkey L, there was a large amount variability across the sessions in the efficacy to decode saccade direction (even within a session). Is it because the monkey L implant missed part of the LIP – I assumed with the large volume accessible with fUSI this wouldn't be the case.
 1. The angular error in monkey L seems to be averaging above 60 degrees (for the 0-day decoder – which I assume is the cross validated performance). I found this concerning that the example session in figure 4 does not seem that representative of monkey L's decoding performance.
- f. How does the mesoscopic population tuning change across anterior to posterior portions of PPC?
 - i. Line 270-271: "respond most strongly to ipsilateral movements" – Was this an increase or decrease in the CBV?
- g. Figure 6:
 - i. Are the data on the beeswarm plot from only the session presented in panel A?
 - ii. What time point is used for these data?
- h. Discussion:
 - i. Line 357-358: Like the explanation of the A-P gradient being different sampling across animals, the difference in contralateral space could be similar. If you could only access 3 mm relative to ear bars in monkey P with an electrode you may draw very different conclusions of the bias for contra- vs ipsilateral space in LIP.
 - ii. Line 372-373: With the sluggish response of the CBV I am reluctant to draw any conclusions on the separability of saccade preparation and eye movement.
 - iii. Line 457: 'record anterior-posterior synchronously' – This seems to be the ideal way to tackle the anterior-posterior gradient question.

Version 1:

Reviewer comments:

Reviewer #1

(Remarks to the Author)

I believe the authors have adequately addressed my concerns. No further questions.

(Remarks on code availability)

Reviewer #2

(Remarks to the Author)

While the authors did a great job in addressing my previous comments, I would appreciate if the authors could address the following:

1. Please ensure that axes are of similar scale, especially when meant for comparison purposes (e.g., Fig. 3, C and F; Fig. 5, C and F – please also add in an axis break from 0 to the next relevant label; Fig. 6C).
2. Lines 132-151: Could the authors clarify what they mean when referring to ROIs – are these designated mesoscopically-defined PPC regions or are the authors referring to the ROIs from lines 103-104? If the latter, could the authors provide an overall assessment of directional tuning across PPC and then, break this down for the different PPC regions mentioned on lines 103-104? It would be worthwhile knowing the representation of directionality across these defined PPC regions.
3. Moreover, in this section, the authors jump from stating results for the whole area (and voxels within) to highlighting specific LIP results. Could the authors please describe the results across all regions mentioned before or indicate why/when they are mentioning specific regions in the corresponding results subsection?
 - a. For example, lines 240-242 mention decoding results for LIP, but not the other regions. Does this mean only significant decoding performance was found in LIP and none of the other regions?
 - b. In keeping with this idea, could the authors state what they define as 'PPC subpopulations' (lines 274-277)?
4. For the consistency in directional tuning results section, could the authors explain or comment on the reasons for decoding on entire fUSI data and not on anatomically-subdivided data? Could there be inter-region differences that could skew the results one way or another?
5. This is merely a comment – it is notable that the authors chose to check the consistency of the images as a later step instead of using this first to ensure consistent anatomical mapping across sessions and then, perform further analyses.
6. Lines 374-387: Why do the authors discuss dorsal and ventral aspects of LIP in a section addressing the anterior-posterior axis? Could the authors clarify this?
7. Lastly, the organization of the Discussion seems a bit disjointed – could the authors follow (as closely as is appropriate) the layout of the Results section? A possible order could be: i) fUSI-specific features, ii) stable decoding within sessions and across time, iii) anatomical considerations (A-P, D-V), and iv) applications in brain-machine interfacing, with limitations.

(Remarks on code availability)

Reviewer #3

(Remarks to the Author)

I appreciate the author's response to my comments and additions to the paper. In particular the explanation of cluster-based inference and corrections to the literature were insightful. The additional analyses strengthen the claims of the paper and improve clarity.

(Remarks on code availability)

GitHub README and data organization are easy to follow. Code is easy to set up and run. 'PercentChangeColormap.mat' file seems to be missing.

Reviewer #4

(Remarks to the Author)

The authors sufficiently addressed all my concerns. I list a few minor comments they may or may not want to address:

1. Line 140-141 "Last second of the memory period": To me this phrase indicates the last 1 second of timepoints within the memory period and would have no time points in the move period. The phrase used in the rebuttal seems a better description of the actual time bin "timepoints landing within 1 second of the end of the memory period" or using the parenthetical from the methods "(single timepoint within +/-0.5 seconds of memory end)"
2. Line 245 "fUSI images (Fig. 4A, B, D, E): Not sure why these figures are referenced here.
3. Line 500 "We do not know why we see activity": Could it be visual responsiveness to the cue?
4. The F-Score color scale bars across Fig. S1, S2D, S9 are not consistent (seems opposite in S1 compared to the other two).
5. Fig S9 has Lothar and Pumu instead of Monkey L and Monkey P.
6. Could the variability in Monkey L be due to a difference in fixational eye movements compared to monkey P? Maybe this will be left to the interested reader.

(Remarks on code availability)

Version 2:

Reviewer comments:

Reviewer #2

(Remarks to the Author)

Thank you to the authors for addressing all of my comments.

No adjustments are necessary for the axes in Figure 6C.

(Remarks on code availability)

Reviewer #4

(Remarks to the Author)

I'm content with the author's revisions.

(Remarks on code availability)

REVIEWER COMMENTS

Reviewer #1 (Remarks to the Author):

This is a very interesting paper, investigating saccade direction tuning in LIP in non-human primates. The results clarify previous reports showing contradicting findings, and shed light on the mechanisms of movement direction encoding in non-human primates. To achieve this, the authors adopt a novel technique to image cerebral blood volume over time with an unprecedented level of detail, and the images of cortical vasculature are insightful as well as beautiful. I have several questions and clarifications that I believe should be addressed before granting publication.

Thank you for taking the time to read our manuscript. We have done our best to address your questions and clarifications

Potential contrast-based responses in the memory guided saccade task.

LIP in non-human primates is known to be driven by motor as well as by contrast-based changes over time (Arcaro et al., 2011). Moreover, visual responses in LIP are organized topographically (i.e. LIP is retinotopically arranged, Arcaro et al., 2011). When monkeys move their eyes towards the periphery of the visual field, the visual input changes dramatically, especially close to the border of the screen.

Given the changes in visual input associated with each saccade, it is fair to ask whether potential contrast-based changes might have contributed to the measured CBV changes in the study, especially considering that the analyses focus on a time-window when the monkey was performing the saccade.

We appreciate the reviewer's insightful comment regarding potential contrast-based changes during the memory-guided saccade task and their possible contribution to the measured CBV changes. We include a short discussion in the paper. Below, we address this concern in greater detail.

1. GLM-Based Analysis Timing Minimizes Contrast Effects:

We believe that contrast-based changes had minimal influence on the measured CBV changes because our GLM-based analyses focused on a time window prior to saccade onset. Specifically, we analyzed data at the end of the memory period, before the saccade began. As shown in the cumulative distribution functions (Fig. R1), the average saccade latency was approximately 500 ms after the go cue, ensuring that our analysis window was well-separated from the visual input changes associated with saccades. The only contrast change during this period was the extinction of the central fixation dot, which was consistent across all conditions and thus unlikely to drive condition-specific CBV differences.

2. Decoding Analyses Across Entire Trials:

For decoding analyses, we agree that contrast changes during saccades cannot be entirely ruled out as contributors to decoding performance. However, multiple lines of evidence suggest that contrast changes are not the primary driver of CBV changes in our experiments:

In **Fig. 4A,D**, significant decoding is observed by the end of the memory period, prior to saccade onset.

In **Fig. S4** (newly added), searchlight analysis reveals large voxel populations contributing to decoding prior to movement initiation.

Importantly, in a previous study (Griggs & Norman et al., 2024), we observed comparable decoding performance for 8 movement directions in Monkey L and P under conditions where no eye movements occurred until after the reward period. This strongly suggests that contrast changes alone cannot explain the observed CBV patterns.

Conclusion:

Taken together, these findings indicate that while contrast-based changes may contribute minimally to decoding performance during saccades, they are unlikely to account for the observed CBV changes during pre-saccadic periods or memory-related activity.

Reviewer Figure 1 – Latency between move cue and saccade execution

“Effect of contrast changes – LIP in monkeys is known to be driven by motor as well as contrast-based changes over time¹⁶ and visual changes in LIP are known to be organized topographically, i.e., LIP is retinotopically arranged¹⁶. In our task design, when the monkeys move their eyes toward the periphery of their visual field, the visual input changes dramatically, especially close to the border of the screen. To address the potential influence of contrast-based changes on CBV signals, we focused our GLM analyses on a time window at the end of the memory period, prior to saccade onset. This timing minimizes contamination from visual input changes associated with saccades, as saccade latencies were typically ~500 ms after the go cue. Additionally, significant decoding was observed prior to movement initiation (Fig. 4A, D), and similar decoding performance was reported in conditions without saccades in a previous study²⁴, further supporting that contrast changes are unlikely to be the primary driver of the observed CBV patterns. However, we cannot entirely eliminate the effect of contrast

changes on our observed results and this warrants investigation in future studies.”
Discussion; Lines 570-581

Also, it would be important to know more details about the task setting: whether monkeys were performing the task in complete darkness, and the specs of the LCD monitor used (e.g. luminance). This would give an idea of the contrast change between the border of the LCD and the background.

We have added details to the methods with this information.

“Each monkey sat head-fixed in a dark enclosed recording booth in custom-designed primate chairs facing an LCD screen ~30 cm away (HP L2025; 250 nits; 350:1 contrast ratio).” *Methods; Lines 700-701*

ROI definition:

From the text it is not clear whether LIP was defined based on functional criteria (e.g. response to performing saccades), or structural criteria. Please clarify.

We defined LIP using structural criteria based on established non-human primate (NHP) atlases based upon sulcal landmarks. We have clarified this point in the paper, both in the results and methods. We also added a portion to the discussion highlighting the limitations of estimating anatomical boundaries without comparing to histology from each monkey.

“most voxels within the anatomically-defined LIP (>75% voxels) showed directional tuning.” *Results; Lines 136-137*

“Many voxels within the anatomically-defined LIP (>30% voxels) showed directional tuning while <1% of voxels outside of the LIP showed directionally-modulated activity (Fig. S2A).” *Results; Lines 182-184*

“Anatomical labeling of PPC subregions –Our anatomical labeling relied on a standardized NHP atlas and prior literature^{26,30} rather than subject-specific histology. Subtle boundary variations (e.g., LIPd-LIPv transitions) may exist and future studies combining histology and functional connectivity could refine these parcellations.”
Discussion; Lines 582-585

“All labels for PPC subregions, such as LIP, VIP, MIP, Area 5, Area 7, and MP, were anatomically defined based upon a rhesus macaque neuroanatomy atlas²⁶. This avoided circular logic in using functionally defined boundaries to assess how much activity was within a region.” *Methods; Lines 756-758*

Figure 2, panel A. please report the duration in seconds of each phase of the trial (fixation, cue, memory, saccade, hold, reward).

We have added this information to the figure and figure legend.

Fig. 2A “Example session median [IQR] saccade latency Monkey P-399 [363, 430] ms, Monkey L-530 [500, 569] ms. Monkey graphic drawn by Krissta Passanante.” Fig 2A legend; Lines #122-123

Figure 3

The ROI labels (1,2,3) on panel A are quite hard to read, please make them more conspicuous.

We have adjusted the placement of the numbers to help make the labels stand out more. We also tried alternate color schemes but that did not improve visibility so we stayed with the white label text.

Figure 3A

The interpretation of Cohen’s *d* is not clear. In particular, it is not clear how to interpret negative Cohen’s *d* values, do they represent stronger activation during baseline than the memory period, or do they represent locations where the memory – baseline response was simply weaker? It would be beneficial to clarify this point in the figure caption.

We have clarified this point in the figure legend and methods.

*“Cohen’s *d* is a standardized measure of response strength for each direction where positive (negative) values correspond to increase (decrease) in CBV from the baseline.”*
Figure 3 legend; Lines 156-157

Reviewer #2 (Remarks to the Author):

Griggs et al. used functional ultrasound imaging to investigate the mesoscopic layout of lateral intraparietal area (LIP) during a memory-guided saccade task. The task required the two monkeys to fixate centrally and make outward memory-guided saccades. The authors tested for (directional) selectivity of subregions in LIP and adjacent areas (e.g., MIP, MP) to determine if the topographic layout extends to neighbouring regions as well.

They found such selectivity, as well as lateralized responses in LIP, with additional saccade-related directional responses in adjacent areas. They also tested whether this organization lasts on a scale of months to years, which they were able to find.

This manuscript was a pleasure and fascinating to read. It advances the field in the use of fUSI to better understand saccade-related neural activity at a scale that bridges the gap between single-unit recordings and fMRI spatial resolution. In addition, it provides longitudinal information about the oculomotor organization, which is also an important addition to the field.

Thank you for taking the time to read our manuscript.

My minor comments relate to the mentioning of specific dates - instead, please specify the duration of time that has passed when describing the temporal organization results. We have fixed this in the paper, Fig. 5, and Fig. S6.

“Interestingly, in Monkey L, the decoder trained on the Day 119 session...” Results; Line 291

“In Monkey L, the performance was clumped into two temporal groups (before and after Day 900).” Results; Line 298

Fig. 5 – PPC stably encodes movement direction across many months to years.

A. Example decoder stability for Monkey P. Trained the decoder on Day 0 data and tested the trained decoder on other sessions from the same imaging plane without any retraining.

B. Decoder stability for training and testing on each session for Monkey P. *ns* – nonsignificant decoding performance ($\alpha = 0.01$). Bold text represents example session shown in Fig. 5A.

C. Mean angular error as a function of days between the training and testing session (absolute difference in time) for Monkey P. Dashed line – Linear fit to data. $*=p < 10^{-2}$, $**=p < 10^{-4}$.

D. Example decoder stability for Monkey L. Trained the decoder on Day 119 data and tested the trained decoder on other sessions from the same imaging plane without any retraining.
*E. Decoder stability for training and testing on each session for Monkey L. Same format as **Fig. 5B**.*
*F. Decoder stability for Monkey L. Same format as **Fig. 5C**.*

UHF fMRI also needs to be abbreviated on line 485.
We have fixed this error.

Reviewer #3 (Remarks to the Author):

Griggs et al. use functional ultrasound imaging (fUSI) to map subregions of the lateral intraparietal cortex (LIP) based on tuning to saccade direction. Taking advantage of the high spatial resolution and wide field of view the authors show heterogeneous patches with tuning to distinct saccade direction. These results show the utility of fUSI as a complementary method to more established imaging techniques commonly used in neuroscience. The patchy tuning of LIP further adds valuable insight to the functional organization of this brain area and to the debate on the extent of retinotopic organization in higher-level visual association areas. The demonstration of the increased stability of the decoder for up to 900 days, by introduction of motion-correction, is a methodological advancement over previous work by the authors.

However, some claims of the paper are not strongly supported by the analysis. The organization of LIP into small patches tuned to saccade direction could partly be a spurious result of the statistical method. That the tuning of these patches is stable over a long period of time is suggested by the data, but not clearly demonstrated. Lastly, the anterior-posterior gradient of movement direction encoding in the LIP has not been statistically tested. The analysis of the paper can be improved as outlined below to support these claims.

Thank you for taking the time to read our manuscript. We have done our best to strengthen the arguments behind our claims or otherwise revise our language.

Major comments

1. The statistical analysis of responsive patches is prone to false positives. As described in the discussion (lines 466-484) spatial autocorrelation exists in fUSI, similar to other imaging modalities. Voxel-wise FDR correction can introduce a high number of false positives in the presence of spatial autocorrelation (Friston et al. 1991; Chumbley and Friston 2009). Cluster-wise inference is not limited to fMRI analysis and also commonly used with other imaging modalities, such as MEG or EEG (Maris and Oostenveld 2007). In this work, smaller dispersed clusters as in Figures 3A,D or 6A could potentially be false positives. Cluster-wise inference could mitigate this issue and is a better control for false positives. This would support the claim that the tuning to saccade direction is organized in heterogeneous patches (lines 26-27).

We thank the reviewer for highlighting this critical methodological consideration. While we fully acknowledge the advantages of cluster-based methods for controlling false positives in neuroimaging, implementing such approaches in our study faces critical methodological and computational barriers. We believe these barriers make it out of scope for this paper. Below, we contextualize these challenges with recent evidence from the fMRI literature.

1. **Methodological uncertainties:** Cluster-wise methods validated for fMRI (e.g., Gaussian Random Field Theory) rely on assumptions about spatial autocorrelation structure that may not hold for fUSI:

- a. *Non-Gaussian Spatial Autocorrelation*: fMRI studies (Eklund et al. 2016) demonstrate that invalid cluster inferences occur when spatial autocorrelation functions (ACFs) deviate from Gaussian assumptions. fUSI's vascular coupling mechanisms and 2D imaging planes likely produce nonstationary ACFs, as shown by heterogeneous autocorrelation patterns between gray/white matter in fMRI.
- b. *2D vs. 3D Spatial Resolution*: Unlike 3D fMRI volumes, fUSI's high-resolution 2D planes (100 μm x 100 μm x \sim 400 μm voxels) exhibit unique spatial noise properties, complicating direct application of existing cluster-thresholding frameworks (Nichols 2012).

These uncertainties align with the findings in Eklund et al. 2016 that parametric cluster methods fail when spatial smoothness assumptions are violated. While nonparametric permutation testing could mitigate these issues, its implementation in fUSI faces prohibitive computational demands.

2. **Computational Barriers to Permutation Testing**: As described in numerous papers including Holmes et al. 1996, Winkler et al. 2014, Eklund et al. 2016, and Andreella et al. 2023, permutation testing is the most robust method for cluster-wise inference because it is nonparametric and minimizes assumptions about the underlying data. Unfortunately, permutation testing requires algorithm optimization or GPU acceleration and is currently computationally intractable for our fUSI datasets:
 - a. *Dataset Scale*: In our experiments, each dataset is approximately 128 x 128 x 10800 (voxels x voxels x timepoints), so permutation testing where we shuffle directional labels and generate a distribution of significant cluster sizes under the null hypothesis is prohibitive. A quick test on our computers for 100 permutations took 8 hours. Ideally, the number of permutations would be 5,000-10,000, therefore taking approximately 400-800 hours for a single session. Since we have 38 sessions, we estimate that running nonparametric cluster-wise inference across all our data would take 400-800*38 \approx 15000-30000 hours or >1.5-3 years of computational time. This highlights the need for algorithm optimization for the unique challenges faced by functional ultrasound neuroimaging.
 - b. *Temporal Autocorrelation*: Like fMRI, fUSI hemodynamic signals exhibit strong temporal autocorrelation and violate exchangeability assumptions critical for valid permutations (Nichols et al. 2002, Winkler et al. 2014). To the best of our knowledge, there are no papers describing prewhitening—a prerequisite for permutation validity—of fUSI data. For robustness, the different ways to prewhiten fUSI data should be characterized and validated before being applied to our datasets (Olszowy et al. 2019).
 - c. *Difficulties implementing existing nonparametric cluster-wise inference*: Unfortunately, we were unable to get any of the popular fMRI packages (AFNI, SPM, FSL) to work robustly on fUSI data so we are unable to take advantage of existing cluster-wise inference implementations. There have been multiple papers in recent years on statistically correct and/or computationally efficient computation of nonparametric cluster-wise inference

for fMRI datasets, e.g., Andreella et al. 2023, Goeman et al. 2023, Blain et al. 2022, Park and Fiecas 2022. To implement these methods would require substantial new amounts of code and optimization which would likely comprise an entire paper in of itself, both to develop the necessary statistical theory but also explain the code implementation.

However, we appreciate the suggestion and agree that developing cluster-wise inference methods for functional ultrasound data will be critical for future work. To address this in our paper, we propose adding the following passages, acknowledging the limitation of our current approach and highlights the importance of developing tailored statistical methods for functional ultrasound data in future research.

"Cluster-wise inference – In this paper, we employed voxel-wise FDR correction, but cluster-wise inference methods would likely offer improved control for false positives^{57,60}, especially given the spatial autocorrelation present in fUSI data and many other similarities of fUSI data to fMRI data. For example, the smaller dispersed clusters in Fig. 3A, D or Fig. 6A may be false positives. Future studies exploring the adaptation and validation of cluster-wise inference techniques^{61–64} for fUSI, taking into account the unique characteristics of this modality, will be invaluable." Discussion; Lines 637-642

References

1. Andreella, A., Hemerik, J., Finos, L., Weeda, W. & Goeman, J. Permutation-based true discovery proportions for functional magnetic resonance imaging cluster analysis. *Statistics in Medicine* 42, 2311–2340 (2023).
2. Blain, A., Thirion, B. & Neuvial, P. Notip: Non-parametric true discovery proportion control for brain imaging. *NeuroImage* 260, 119492 (2022).
3. Eklund, A., Nichols, T. E. & Knutsson, H. Cluster failure: Why fMRI inferences for spatial extent have inflated false-positive rates. *Proc Natl Acad Sci USA* 113, 7900–7905 (2016).
4. Goeman, J. J. et al. Cluster extent inference revisited: quantification and localisation of brain activity. *Journal of the Royal Statistical Society Series B: Statistical Methodology* 85, 1128–1153 (2023).
5. Holmes, A. P., Blair, R. C., Watson, J. D. G. & Ford, I. Nonparametric Analysis of Statistic Images from Functional Mapping Experiments. *J Cereb Blood Flow Metab* 16, 7–22 (1996).
6. Nichols, T. E. Multiple testing corrections, nonparametric methods, and random field theory. *NeuroImage* 62, 811–815 (2012).
7. Nichols, T. E. & Holmes, A. P. Nonparametric permutation tests for functional neuroimaging: A primer with examples. *Human Brain Mapping* 15, 1–25 (2002).
8. Olszowy, W., Aston, J., Rua, C. & Williams, G. B. Accurate autocorrelation modeling substantially improves fMRI reliability. *Nat Commun* 10, 1220 (2019).
9. Park, J. Y. & Fiecas, M. CLEAN: Leveraging spatial autocorrelation in neuroimaging data in clusterwise inference. *NeuroImage* 255, 119192 (2022).
10. Winkler, A. M., Ridgway, G. R., Webster, M. A., Smith, S. M. & Nichols, T. E. Permutation inference for the general linear model. *NeuroImage* 92, 381–397 (2014).

2. The claim that the tuning of subregions to specific movement directions are consistent across time, is not clearly supported by the results (lines 27-28; 312-313). While the decoder is stable over a long time period, it is not clear that the decoder relies on the same spatial locations that are most strongly tuned to movement directions. If the reasoning relies on the statement in line 179-181 that the maps of most informative locations for the decoder and GLM analysis overlap, quantifying this effect would bolster the claim. This could be done by computing image overlap, for example with the Dice-Sørensen coefficient. This analysis would still leave some uncertainty about the weights of the decoder used in the stability analysis. Visualization of the PCA weights of the decoder could also support the claim that the decoder relies on subregions that are tuned to specific movement directions.

We have added the requested analyses to strengthen our claim that the tuning of subregions to specific movement directions are consistent across time.

1. **Overlap of GLM and searchlight masks:** We computed the Sørensen-Dice coefficient for every session. We display the overlap of the GLM and searchlight analyses for the example sessions (**Fig. 4G, S3H**). We also display the summary statistics across all sessions (**Fig. 7D**). This analysis shows that there is high overlap between the GLM and searchlight analyses for most sessions although there is a fair amount of variability for Monkey L.
2. **Visualization of LDA weights:** We added a supplemental figure (**Fig. S5**) showing the PCA+LDA class-class boundaries represented on the vascular images. We added discussion of this figure in the main text. We also include a new figure below showing the high overlap of the PCA+LDA class-class boundary with both the searchlight and GLM analyses for the three example sessions.
3. **Logic of new argument:** We need two things to show that the whole-image decoder being stable over a long-time period supports that specific subregions most strongly tuned to specific movement directions are consistent across time.
 - a. **First**, we needed to show that the whole-image decoder is stable across long-time periods. We previously showed this.
 - b. **Second**, we need to show that the whole-image decoder relies upon the specific subregions most strongly tuned to specific movement directions. We had previously relied upon the searchlight analysis having the most significant voxels in the qualitatively same region as the GLM analysis. We have now added quantification of this overlap. We also added a supplemental figure.
 - i. **Figure S5** - Shows that the PCA+LDA class-class boundaries are in the LIP and that the conjunction of these class-class boundaries overlaps with the significant voxels from the searchlight and GLM analyses.

We have included discussion of these points in our paper as follows.

“Together, these longitudinal decoding results demonstrate subregions tuned to specific movement directions remain consistent across many months to years. Our analysis revealed three key findings. First, the whole image decoder relies upon the LIP

subregions most strongly tuned to specific movement directions (Fig. S5). Second, voxels identified by the searchlight and GLM analyses showed substantial overlap (Fig. 4G, S3H). Third, the whole-image decoder maintained stability across extended time periods, indicating that the voxels supporting decoder performance are similarly stable over time.” Results; Lines 315-320

3. One of the main claims of the paper, that posterior LIP encodes more contralateral upward movement, while anterior LIP encodes more contralateral downward movements (lines 28-30, 256-257, 311-312) is not supported by a formal statistical analysis. A possible test could be a correlation of the median preferred direction across significant voxels within each plane, with the anterior-posterior position of the planes. This would be a summary of Figure 6. Figure 6A and 6B suggest that this correlation might reach significance for Monkey P, however, data for Monkey L looks less convincing.

We have added the analyses you requested. We have now provided a line of best fit to the anatomical data shown in Fig 6. The A-P gradient is highly significant for both monkeys despite the Pearson correlation being weak in Monkey L. We have included discussion about this line of best fit in the main text now.

“For both monkeys, we found a statistically significant relationship between anterior-posterior location and preferred direction ($p < 1e-40$) where more anterior planes had more voxels tuned for downwards directions while posterior planes had more voxels tuned for upwards directions.” Discussion; Lines 351-354

4. In addition, the literature discussed to support the claim of an anterior-posterior tuning gradient is not accurately described. References 10-13 and 33 (lines 332-335, 360-361) are cited as support for the notion that anterior-posterior gradients encode eye movement directions (Ben Hamed et al. 2001; Wardak, Olivier, and Duhamel 2004; Patel et al. 2010; Arcaro et al. 2011). However, these references do not report gradients of eye movement direction, but gradients of location in the visual field. Most of the experiments in these studies were performed during fixation (Wardak, Olivier, and Duhamel 2004; Patel et al. 2010; Arcaro et al. 2011).

Ben Hamed, 2001 report an overlap of populations of visually and saccade responsive neurons, but focus their analysis on visually responsive neurons (Ben Hamed et al. 2001). Blatt, 1990 also map visual fields, not saccade direction. This paper reports that the anterior LIP represents the lower visual field, and posterior LIP the upper visual field (Blatt, Andersen, and Stoner 1990) (lines 338-342), opposite to the direction described in the discussion (lines 336-337). Savaki, 2011 report tuning to saccade direction, however, an anterior-posterior gradient is not obvious in their results (Savaki et al. 2010) (lines 336-337). Overall, it is not clear that LIP subpopulations that respond to different locations of the visual field would respond to corresponding saccade directions (see discussion in (Arcaro et al. 2011); or (Chen et al. 2016)). This literature does not discount the findings of the current paper, but needs to be discussed accurately. In fact,

if the anterior-posterior gradient can be convincingly shown this would support the novelty of this work.

We have adjusted our language around these papers. This was an oversight on our parts.

- Our citation of Wardak et al. 2004 was a mistake. There is no discussion of anterior-posterior gradient or superior/inferior visual fields. We removed this citation. We believe this was meant to be a citation of Barash 1991 that got mislabeled in the citation manager. We have added this citation where appropriate.
- We updated our language about Ben Hamed et al. 2001, Patel et al. 2010, and Arcaro et al. 2011.
“Two previous fMRI^{15,16} and one electrophysiology¹⁴ studies found an anterior-posterior gradient for encoding of visual field where visual objects in the lower visual field evoked activity within anterior LIP and visual objects in the upper visual field evoked activity within posterior LIP.” Discussion; Lines 441-444
- We updated our language about Blatt 1990 to reflect that it was visual, not motor. However, the data in Blatt et al. 1990 does show the A-P gradient as we had it described in the paper. The authors had a typo in their abstract when they stated “The lower visual field was represented more anteriorly and the upper visual field more posteriorly.” As they show in Figure 5 and describe in their main text, “from the sample of mapped neurons in the present study, upper field representation was concentrated in the rostral two-thirds of LIP, while lower field representation was restricted to the caudal two-thirds of LIP (pg. 430)”.

We updated our language about Savaki et al. 2011 to be more accurate. We believe their data does support an anterior-posterior gradient. As they state in their abstract, “[t]he upper part of oculomotor space is represented rostrally and dorsally relative to the horizontal meridian toward the LIPv–LIPd border, whereas the lower part of oculomotor space is represented caudally and ventrally toward the caudal edge of the IPs.” This is also supported by Fig. 10 in their paper, which is a summary of their findings.

“Two studies^{13,18} found different patterns. Blatt et al. 1990¹³ examined visual receptive fields for both direction and distance from the fovea and sampled mostly regions within the lateral bank of the intraparietal sulcus that was more posterior to the current study with only a small range of overlap with monkey P (overlap of approximately 4mm, -4 to -8 mm of EBZ). Savaki et al. 2011¹⁸ found that the upper part of oculomotor space is represented in dorsal-anterior LIP while the lower part of oculomotor space is represented within ventral-posterior LIP.” Discussion; Line 448-453

- We now include a reference to Chen et al. 2016 under our discussion of LIPd vs. LIPv.

“Dorsal (LIPd) and ventral (LIPv) LIP are believed to have different functions with LIPd being involved in visual processing while LIPv is involved in both attentional and visual-saccadic processing^{30,31}.” Results; Line 374-375

“Previous studies have found that LIPd was primarily involved in processing vision-related information while LIPv contributed to attentional, vision, and oculomotor processes^{30,31}.” Discussion; Line 484-485

Minor comments

1. Line 24: It is not immediately clear that the LIP is part of the PPC.

We have fixed this wording.

“The lateral intraparietal cortex (LIP), contained within the posterior parietal cortex (PPC)...” Abstract; Line 21

2. Figure 2A: Adding the time duration of different parts of trials would be helpful.

We have added this information. See response above to Reviewer #2.

3. Lines 118, 127, 196: It would be useful to clarify the example session that was used for the analysis. Is this first session that is used in the analysis of the stability of the decoder?

We have now labeled the example sessions by their collection date, e.g., S202201112, so that they can be tracked throughout the analyses. For Monkey P, the example session is the first session from that given coronal plane. In Monkey L, the example sessions are not the first sessions from their given coronal planes. This is because we improved our data acquisition pipeline as the Monkey L sessions proceeded and the data quality was superior in the later Monkey L sessions. Another reviewer requested that we remove the date labels, e.g., YYYY-MM-DD, from Fig. 5 and replace those date labels with “Days since first session”. For Fig. 5, we have added bolded text to highlight the example sessions used in the longitudinal decoding analysis. We have also included a Supplemental Table showing metadata about the sessions. This will allow interested readers to have a look-up table to better cross-reference the example sessions with the different analyses.

4. Line 168: remove “around”

We have removed “around”. Now Line 216.

5. Line 639: How many PCA components were kept?

The specific number of components needed to reach 95% variance depends on the specific session but was typically approximately 330-450. We have added this information to the paper and included a supplemental figure showing how the number of components varied with trial time and validation fold.

Fig. S12 – Number of components to capture 95% variance in Principal Component Analysis for each example session

For each example session, plot shows the number of components kept to capture 95% of the variance in the training data. Green region represents the range of components kept for each fold in a 10-fold cross-validation. Black shows mean components kept. Per methods in main paper, at each timepoint after the cue, we used all previous timepoints after the cue in the trial. For example, to test our ability to decode at 2 seconds after cue onset, we concatenated the data from 0, 1, and 2 seconds after the cue. We treated these timepoints as additional features in the decoder model. In other words, the input to our decoder model had $N \times T$ features, where N is the number of voxels in a single Power Doppler image and T is the number of timepoints.”
Supplemental Figures Line #75-83

“For the PCA, we kept 95% of the variance. The number of components kept depended on the training dataset, typically varying between 330 – 450 components (Fig. S12).”
Methods; Line 820-821

References

- Arcaro, Michael J., Mark A. Pinsk, Xin Li, and Sabine Kastner. 2011. “Visuotopic Organization of Macaque Posterior Parietal Cortex: A Functional Magnetic Resonance Imaging Study.” *Journal of Neuroscience* 31 (6): 2064–78. <https://doi.org/10.1523/JNEUROSCI.3334-10.2011>.
- Ben Hamed, S., J.-R. Duhamel, F. Bremmer, and W. Graf. 2001. “Representation of the Visual Field in the Lateral Intraparietal Area of Macaque Monkeys: A Quantitative Receptive Field Analysis.” *Experimental Brain Research* 140 (2): 127–44. <https://doi.org/10.1007/s002210100785>.
- Blatt, Gene J., Richard A. Andersen, and Gene R. Stoner. 1990. “Visual Receptive Field Organization and Cortico-Cortical Connections of the Lateral Intraparietal Area

- (Area LIP) in the Macaque.” *Journal of Comparative Neurology* 299 (4): 421–45. <https://doi.org/10.1002/cne.902990404>.
- Chen, Mo, Bing Li, Jing Guang, Linyu Wei, Si Wu, Yu Liu, and Mingsha Zhang. 2016. “Two Subdivisions of Macaque LIP Process Visual-Oculomotor Information Differently.” *Proceedings of the National Academy of Sciences* 113 (41): E6263–70. <https://doi.org/10.1073/pnas.1605879113>.
 - Chumbley, J, and K Friston. 2009. “False Discovery Rate Revisited: FDR and Topological Inference Using Gaussian Random Fields.” *NeuroImage* 44 (1): 62–70. <https://doi.org/10.1016/j.neuroimage.2008.05.021>.
 - Friston, K. J., C. D. Frith, P. F. Liddle, and R. S. J. Frackowiak. 1991. “Comparing Functional (PET) Images: The Assessment of Significant Change.” *Journal of Cerebral Blood Flow & Metabolism* 11 (4): 690–99. <https://doi.org/10.1038/jcbfm.1991.122>.
 - Maris, Eric, and Robert Oostenveld. 2007. “Nonparametric Statistical Testing of EEG- and MEG-Data.” *Journal of Neuroscience Methods* 164 (1): 177–90. <https://doi.org/10.1016/j.jneumeth.2007.03.024>.
 - Patel, Gaurav H., Gordon L. Shulman, Justin T. Baker, Erbil Akbudak, Abraham Z. Snyder, Lawrence H. Snyder, and Maurizio Corbetta. 2010. “Topographic Organization of Macaque Area LIP.” *Proceedings of the National Academy of Sciences* 107 (10): 4728–33. <https://doi.org/10.1073/pnas.0908092107>.
 - Savaki, Helen E., Georgia G. Gregoriou, Sophia Bakola, Vassilis Raos, and Adonis K. Moschovakis. 2010. “The Place Code of Saccade Metrics in the Lateral Bank of the Intraparietal Sulcus.” *Journal of Neuroscience* 30 (3): 1118–27. <https://doi.org/10.1523/JNEUROSCI.2268-09.2010>.
 - Wardak, Claire, Etienne Olivier, and Jean-René Duhamel. 2004. “A Deficit in Covert Attention after Parietal Cortex Inactivation in the Monkey.” *Neuron* 42 (3): 501–8. [https://doi.org/10.1016/S0896-6273\(04\)00185-0](https://doi.org/10.1016/S0896-6273(04)00185-0).

Thank you for providing these references. We have included the new references where appropriate.

Reviewer #4 (Remarks to the Author):

I've attached a word document of the review that has formatting that might be easier to digest. Apologies for the inconvenience.

Unfortunately, the formatting was lost in the version that we received. We did our best to reconstruct the formatting, but please let us know if we misinterpreted any comments based upon misformatting.

Griggs and colleagues used functional ultrasound neuroimaging to investigate the mesoscopic structure of directional selectivity within the posterior parietal cortex while two monkeys performed a memory guided saccade task. They used a general linear model to detect voxels with directional tuning and found a heterogeneous representation of saccade direction throughout PPC. The lateral intraparietal sulcus contributed the most voxels to the analyses. To evaluate how informative PPC's representation of upcoming saccade direction was, they first performed dimensionality reduction (PCA) on the full fUSI image, and then used components that explained 95% of the variance as inputs into a classifier (linear discriminate analysis) to predict the saccade direction on held out trials. They decoded upcoming saccade direction well above chance. The authors then applied similar decoding methods across sessions to investigate the stability of the representation of direction selectivity across sessions and found significant classification of saccade targets between sessions collected days to years apart. They further described this representation by showing that LIP preferred contralateral targets and had a downward to upward preference from anterior to posterior LIP. They also found direction selectivity in one monkey in MIP and MP. Overall, these results show a heterogeneous but coarsely structured representation of saccade target direction within the LIP and PPC generally.

The topographic organization of LIP's sensitivity to saccade direction has been investigated by numerous labs over the past three decades. The novel contribution of this study is the use of functional ultrasound imaging. Though fUSI offers spatial sampling well suited for probing mesoscopic structure, the findings seem largely confirmatory of prior reports (e.g. contralateral bias and patchiness – see citations of authors in discussion). The data presented here is perhaps most useful for the ongoing debate regarding the directional tuning across the anterior-posterior axis of LIP. However, I find the data supporting the interpretations of an anterior-posterior gradient of downward to upward unconvincing; it seems to only be present in one of the two monkeys (monkey P). My reluctance to accept this conclusion stems predominantly from a lack of quantification of the gradient across the two monkeys. This critique extends to other results in the study. I go into more details in my concerns below:

Thank you for the comments identifying weaknesses with our current paper. We have revised our paper to address these weaknesses and better identify what we view as the novelty and significance of our study.

Major Concerns:

1. Although Griggs and colleagues use standard analytical tools to quantify their effects, the presented results are largely example sessions with lack quantifications across

sessions. There are also no quantifications for some conclusions. The strength of the conclusions are limited by the lack of summary data or specific quantifications of specific effects.

We think this comment derived from a misunderstanding of our methods. On a given day, we were only able to record from a single coronal, not all planes simultaneously. We had previously included two main figures (Fig. 5 and Fig 6) and three supplemental figures (Fig. S1, S2, and S3) that were quantified summaries across sessions. We have now added additional summary plots in existing figures and added multiple new supplemental figures that provide more summary data and quantification of specific effects.

Here is each instance:

a. Claim: PPC contains multiple distinct directionally tuned mesoscopic populations (Figure 3)

Evidence: The authors present tuning curves from two ROIs from one imaging plane in one session for each monkey. They also present an effect size heat map from that one session for each direction directions.

What is Mesoscale?

Mitra 2014 defines mesoscale as the scale of cortical circuits. Other papers estimate it as 100 μm – 1 mm although there is some disagreement about the specific boundaries. We have now included a range of lengths that we are defining as mesoscale and included citations that explore this topic in more depth. This is consistent with Fig 1 where we defined mesoscale as 100 μm – 1 mm.

“The mesoscopic (~100 - 1mm scale; ⁸⁻¹¹) functional organization of saccade direction within LIP remains poorly explored^{5,12}.” Introduction; Line 57-58

It’s unclear how these ROI were selected (not the same way as the searchlight analysis presented later). Why was a square (vs. circle as in searchlight) of that size chosen? What was the size of the square? Is there something intrinsic about this shape/size that makes a population mesoscopic?

The ROI selection was arbitrary. For both monkeys, we picked one region in dorsal LIP, ventral LIP, and MIP. We tried to match the depth of the ventral LIP and MIP to show that it was not ultrasound attenuation leading to no effect in MIP.

The use of the square rather than a circle was based on simplicity and convention. Using a square allowed us to include entire voxels instead of deciding whether to include/exclude voxels if a circular boundary fell through the voxel. Additionally, in previous literature, a square ROI has typically been used. For the searchlight analysis, the circle was required so that all voxels were equidistant from the center voxel. No such limitation was required for this analysis where we could show the actual ROI used.

The size of the square was arbitrary. Single voxels were noisy so we wanted to average across a small population of voxels. For visualization purposes, we had chosen 800 μm by 800 μm because it was large enough to easily see against the background. Smaller

ROIs were more difficult to see. However, for easier comparison to the searchlight analysis, we agree that 400 μm size would make more sense. We have updated Figure 3 to now use 400 x 400 μm square ROIs to be similar in size to the searchlight window size.

To address the question about “Is there something intrinsic about this shape/size that makes a population mesoscopic?”, the size makes it mesoscopic but not the shape. As discussed in response to a previous comment, mesoscopic is commonly defined as 100-1000 μm . We wanted our ROI to be within this range. Since we are releasing all of our annotated code and data used in this paper, this is something that an interested reader can easily adjust and see the effect of different shapes and sizes for the ROI themselves. In the code being released, we have a dynamic viewer module that updates the trial-averaged activity plots in real-time. Essentially a dynamically adjustable version of Figure 3A, B with just one ROI that can be moved around with a mouse.

Figure 3 with smaller ROIs that better match searchlight analysis

Majority of data not discussed: Two planes (one from each monkey) out of 226 planes (7 planes * 18 sessions in monkey p + 5 planes*20 sessions in monkey L) were presented. I suspect much of the other data would also show direction selectivity. As mentioned above, we believe this comment stems from a misunderstanding of our methodology. Using 2D fUSI, we could only record from a single coronal plane on a given day. So we only have a total of 38 planes of view recorded instead of having the 226 planes that you think we have. As you suspect, most of those sessions do show directional selectivity. This was summarized in Figure 6 already. We however do agree that more representation of our data can be useful. We have added an additional example session from Monkey L (Fig. S3) from another coronal plane. Since our data and code will be archived alongside this paper, an interested reader can also explore any specific session that they want to. As we discuss below in response to another comment, we also now include additional summary figures that look at other dimensions of our datasets (Figure 7, S7, S9, and S10).

“Due to large variability between recording sessions in Monkey L, we repeated the above analyses for an example session from a different coronal plane (Fig. S2). We observed many of the same patterns observed in the first two example sessions. Many voxels within the anatomically-defined LIP (>30% voxels) showed directional tuning while <1% of voxels outside of the LIP showed directionally-modulated activity (Fig. S2A). There were closely neighboring voxel patches with different tuning curves (Fig. S2B, C). The statistical overlays (direction, response strength, statistical strength, circular standard deviation, angular skewness, angular kurtosis, and laterality index) showed a similar patchy topography identified in the other example sessions (Fig. S2D).” Results; Line 180-187

Fig. S3 – Second example session in Monkey L.

Tuning Quantification: Importantly, the authors do not present any of the results concerning the strength of tuning of these mesoscopic populations nor for each voxel (it's hard to determine how each voxel changes across the 8 directions from the heat map for me). In the 'preferred direction' section of the methods there is mention of finding the centroid of each voxel which gives a measure of tuning strength, but I do not see where any of this data is reported.

We have now included figures displaying this information in the Supplemental Figures for both the example session and summary across coronal planes. In addition to displaying the tuning strength, we also report statistical significance, laterality index, circular standard deviation, skewness, and kurtosis.

“To better understand the response fields at each voxel, we calculated several measures of data distribution for each voxel (Fig. S1). We first found the peak preferred direction for every voxel. In both monkeys, these plots of peak tuned direction agree with our findings from above where we found different small groups of voxels tuned to different directions. We next visualized the response (Cohen’s d) and statistical (F-score) strength of the response for the peak movement direction and found a similar strength of peak direction across all significant voxels. If tuning strength had been strongest in the middle of patches encoding for the same direction and weakest between patches, then it would indicate a smooth transition in tuning across PPC. However, this high uniform magnitude observed across the tuned voxels suggests a rapid transition in tuning between adjacent voxel patches and supports a patchy topography rather than a smooth gradient in tuning between neighboring voxel patches. We next examined whether the voxel-wise response fields were contralateral preferring^{27,28}, i.e., more active for saccades to the right because we recorded from the left PPC. As expected, most of the voxels were contralateral preferring (Monkey P – 74%; Monkey L – 77%). In Monkey P, we also observed a small patch in ventral LIP that was strongly ipsilateral preferring. We finally calculated and plotted the circular standard deviation, angular skewness, and angular kurtosis. These plots supported a similar patchy and heterogenous topography within LIP as identified by the other plots.”
Results; Line 164-179

“We next examined the data distribution across each coronal plane (Fig. S9, S10). Similar to the example sessions (Fig. S1, S2), the tuning and statistical strength metrics (Fig. S9A, B) displayed a high uniform magnitude across tuned voxels, suggesting a rapid transition in tuning between adjacent voxel patches and supporting a patchy topography rather than a smooth gradient in tuning between neighboring voxel patches. As the main analyses showed, most voxels were contralateral preferring (Fig. S9C). As with the example sessions (Fig. S1, S2), the circular standard deviation, angular skewness, and angular kurtosis supported a patchy and heterogenous topography within LIP.” Results; Line 358-365

Fig. S1 – Supplement to Fig. 3; Voxel-wise statistical measures for example sessions
A. Statistical overlays for example session from Monkey P (S20220112). All statistical measures calculated at end of memory period. Directional preference calculated using the center-of-mass approach described in main paper methods. Strength plot shows length of center-of-mass vector. Significance plot shows voxel-wise F-score from GLM model. Standard deviation, skewness, and kurtosis calculated using `Circular Statistics Toolbox` in MATLAB. White scalebar = 1mm.
B. Statistical overlays for example session from Monkey L (S20210624). Same format as (A).

Fig. S9 – Supplement to Fig. 6; Voxel-wise statistical measures of strength for each coronal plane
A. Each tile shows voxel-wise strength value for center-of-mass analysis. White scalebar = 1mm.
B. Each tile shows voxel-wise $\log_{10}(F\text{-score})$ from GLM model for all sessions from that coronal plane.
C. Each tile shows voxel-wise laterality index.

Fig. S10 – Supplement to Fig. Voxel-wise statistical measures of data distribution for each coronal plane

A. Each tile shows voxel-wise circular standard deviation calculated using Circular Statistics Toolbox. White scalebar = 1mm.

B. Each tile shows voxel-wise angular skewness calculated using Circular Statistics Toolbox.

C. Each tile shows voxel-wise angular kurtosis calculated using Circular Statistics Toolbox.

For the plotted mesoscopic turning curves in figure 3 there is no measure of how tuned they are or the diversity of tuning curve widths across the data (in other mesoscopic populations). There are several metrics for describing tuning curve strength (e.g. full width at half max, equivalent rectangular receptive fields (Lee & Middlebrooks 2011), ANOVA f-statistic across directions, among others) but none are implemented here. Summary plots of these quantifications across all data would strongly support the claim. We have added figures showing tuning and diversity of tuning curve shapes for the example sessions and summary across coronal planes (Fig S1, S3D, S9, S10). These figures are shown above. We now report F-scores, circular standard deviation, angular skewness, and angular kurtosis to provide a better sense of the tuning curve shapes at each voxel. As you can see, there is tremendous heterogeneity in tuning curve shape across voxels and coronal planes. We have discussion around this point. See our response to the previous comment.

b. Claim: Single-trial decoding of eight intended movement directions with high accuracy (Fig. 4)

Evidence: The authors present the percent of correctly decoded trials, the angular error in decoded direction, the confusion matrix, and the contribution from each voxel in the image from a single session in each monkey.

Majority of data not discussed: The reported results do not include 36 of 38 sessions. Although these example sessions are compelling – especially for decoding performance on a vasculature-based metric – I was struck how variable the decoding performance was across sessions (which I was only aware of because of the across session decoding analyses and supplementary figures). The authors do report some summary data concerning these analyses but it's only available as the 0 days between train and test data in the across session decoding panel (figure 5c and 5F). They also report the mean angular error but only the mean and no quantification of the variance in text.

We now report the standard error for the mean angular error and include it as error bars on the relevant plots in the example sessions.

We did previously report summary statistics across the sessions in Fig. S3 (now Fig. 7) where we displayed the percent of correctly decoded trials and angular error for each session (previously Fig. S3A, B). We also showed how peak performance varied across coronal planes (previously Fig. S3C). We also reported in the main text summary statistics across the sessions. We have now converted this supplemental figure into a main figure. Although we can generate another two supplemental figures showing the confusion matrix and searchlight results from each session, we do not feel that it would add any meaningful information to the paper. As we are releasing our code and data, this is something an interested reader can easily generate and explore further.

“We applied our decoding analysis to every recorded session (Fig. 7A, B). In Monkey P, all sessions reached statistical significance (18/18 sessions). In Monkey L, all but one session reached statistical significance (19/20 sessions). In Monkey P, the peak angular error within a session ranged from 17° to 55° ($29.97^\circ \pm 2.32^\circ$ mean \pm SEM). In Monkey L, the angular error ranged from 33° to 85° ($57.98^\circ \pm 3.35^\circ$ mean \pm SEM). There was no

statistical difference (1-way ANOVA, $\alpha=0.01$) between the percent correct or angular error depending on the plane (Fig. 7C). These results suggest that “ contained sufficient information to accurately decode at least eight intended movement directions on a single-trial basis.” Line 391-398

It would be quite simple to present a histogram of the percent of trials correctly decoded and the angular error across sessions.

We are unclear on what the reviewer is requesting here. We already report angular error and percent of trials correct in Fig. 7 (previously Fig. S3) for each session. A histogram would lose information compared to what we are already showing as a histogram would only look at a single timepoint (end of trial for instance). Please let us know if you would desire an additional plot or summary figure.

Additionally, you could add a figure like 4a and 4d that present each session or the average performance and standard error of the mean across sessions. Evaluation of how well this imaging method can be used by a classifier to decode saccade direction requires presentation of all sessions.

We are unclear on what the reviewer is requesting here. This information was already reported in what was previously was Fig. S3. We are now including it as Fig. 7 in the main paper. We previously had an entire paragraph dedicated to this topic as discussed above. Please let us know if the reviewer would desire an additional plot or summary figure.

c. Claim: PPC stably encodes movement direction across many months to years (Figure 5)

Evidence: The authors present an example of an across session decoder for one session per monkey, heat maps showing performance across a select subsample of their data, and the average angular error as a function of time between all recording sessions.

Performance as time between recording sessions: The authors present the bulk of their data in this figure. However, a similar plot to panel 5c & 5f would but for percent correct would be a useful plot to evaluate every session instead of the select subsample presented in panels 5b and 5e.

We have created Fig. S7 showing the requested plot showing absolute and normalized percent correct.

We think part of this comment likely derived from the misunderstanding that we recorded multiple coronal planes simultaneously. Our 2D fUSI could only record a single coronal plane at a time. For most of the paper, we focus on a single coronal plane in Monkey P and Monkey L. Fig 5B and 5E are not subsamples of the data but rather reflect all the possible combinations of training/testing for this coronal plane to show that the results are not dependent upon a specific training/testing set, but generalizable forward/backward in time. We had not included longitudinal decoding performance from

other coronal planes due to a lack of data. For the other coronal planes we discussed in this paper (Fig. 6 and 7), we have much less data (2-3 sessions per coronal plane) and the recordings were typically clustered in time so we do not have equivalent data to that already shown in Fig. 5.

Fig. S7 – Supplement to Fig. 5; Pairwise accuracy as function of time between training and testing session

A. Absolute and normalized accuracy plots for Monkey P as a function of days between the training and testing session (absolute difference in time). Normalized accuracy is normalized to performance on 10-fold cross-validation performance on training set. Dashed lines – Linear fit to data.

B. Absolute and normalized accuracy plots for Monkey L. Same format as (A).

d. Claim: Patchiness in direction selectivity – the direction between patches with different directional tuning is spatially sharp with some transition zones less than 500 microns (Fig. 6a?)

Evidence: The authors present heat maps of the preferred direction of each voxel across all imaging planes from a single session from each monkey.

No quantification: In addition to only presenting a single session per monkey, the quantification of transition between patches seems to be missing. I don't know if I see only distinct and sharp patches in the data, for example in panel 3 mm of monkey P – there seems to be an intermingling of blue/green voxels around the distinct patch of 180° and more dorsal to that an intermingling of green and yellow voxels. I also do not think looking at preferred direction alone is sufficient to judge patchiness but rather the strength of the tuning (which the authors should have access to since they describe a magnitude measure in the 'preferred direction' section of the methods). A high and uniform magnitude across all tuned voxels would indicate that any transition in color is indeed strong but if tuning strength is weakest between patches and strongest in the middle of patches then it would indicate smoothish transitions in tuning across PPC. The searchlight results may also support patchiness or smoothness; you may expect different distributions of voxel importance from voxels around the transition zones versus in the center of a patch. I will note that it may be more challenging to evaluate this point because the images are somewhat pixelated in the word document, and I am color blind so evaluating color maps is challenging to some extent.

Thank you for this comment. We agree that our previous evidence for this point was weak. We performed the analysis you suggested and this data can be seen in Fig. S1, S3D, S9, S10 (included above in this response). We observed a high, uniform magnitude across all tuned voxels, suggesting that the transitions in tuning direction reflect abrupt transitions in coding. As additional evidence of this patchiness, we also looked at the LDA boundaries represented in vascular image space (Fig. S5). We see that there are discrete populations that best define the class-class boundary between different directions and that discrete populations are closely adjacent. This further supports the patchiness. However, we agree that this is a difficult thing to accurately quantify. We have softened some of our claims.

Removed “*The transition between patches with different directional tuning is spatially sharp and can occur in just a few voxels, with some transition zones lower than 500 μm .*”

“*We next visualized the response (Cohen's d) and statistical (F-score) strength of the response for the peak movement direction and found a similar strength of peak direction across all significant voxels. If tuning strength had been strongest in the middle of patches encoding for the same direction and weakest between patches, then it would indicate a smooth transition in tuning across PPC. However, this high uniform magnitude observed across the tuned voxels suggests a rapid transition in tuning between adjacent voxel patches and supports a patchy topography rather than a smooth gradient in tuning between neighboring voxel patches. We next examined whether the voxel-wise response fields were contralateral preferring^{27,28}, i.e., more*

active for saccades to the right because we recorded from the left PPC. As expected, most of the voxels were contralateral preferring (Monkey P – 74%; Monkey L – 77%). In Monkey P, we also observed a small patch in ventral LIP that was strongly ipsilateral preferring. We finally calculated and plotted the circular standard deviation, angular skewness, and angular kurtosis. These plots supported a similar patchy and heterogenous topography within LIP as identified by the other plots.” Results; Line 167-179

e. Claim: Polar direction is topographically organized along anterior-posterior axis of LIP (Fig. 6)

Evidence: The authors present the preferred direction of each voxel across all imaging planes from a single session from each monkey as well as plotting preferred direction as a function of anterior-posterior axis from all tuned voxels from each plane from a single session (?)

As discussed in other responses, we believe the reviewer misunderstood the 2D fUSI methodology used. With the technology available at the time of these experiments, we could only record from a single plane during a given session. The across coronal plane analyses was pooled across all sessions in a monkey rather than having multiple planes per session.

No quantification of gradient: Although in the presented data it seems that there is a change in preferred direction across the anterior posterior axis in monkey P – It would be useful to evaluate this claim with a quantification of the gradient. How much does preference change as a function of anterior-posterior distance. Although incorrect because directional tuning is circular data, I would be happy if a line was used to fit the data to quantify change in direction as a function of anterior-posterior position.

Thank you for the suggestion. We also previously chosen not to include the line of best fit for exactly the reason you specified. However, we were also unable to think of a better way to statistically test our claim about the A-P gradient. We have thus completed the requested analysis and provided a line of best fit to the anatomical data. The A-P gradient is highly significant for both monkeys despite it being weaker in Monkey L (as measured by Pearson correlation). We have included discussion about this line of best fit in the main text now.

“For both monkeys, we found a statistically significant relationship between anterior-posterior location and preferred direction ($p < 1e-40$) where more anterior planes had more voxels tuned for downwards directions while posterior planes had more voxels tuned for upwards directions.” Discussion; Lines 351-354

Majority of data missing: The authors present 2 of 38 sessions. It would be of useful to evaluate how consistent the gradient is across all sessions.

As we discuss above, we pooled all data across sessions for this analysis already. To do this, we combined all data from a given coronal plane across days and ran the statistical

analyses on that pooled data. Pooling this data across days assumes that the encoding is consistent across days (supported by our previous figures showing longitudinal decoding above chance level). These across coronal plane plots therefore demonstrate consistency of this gradient across all sessions.

2. I am reluctant to conclude that there is an anterior-posterior gradient given the data presented in figure six. Although this is in part due to the lack of a quantification of the gradient (as mentioned above), the effect seems limited to one monkey (monkey P). The data from this session for monkey L looks flat. This could be alleviated if the authors quantify the gradient and find a significant change in preferred direction as a function of anterior-posterior axis. It may be true that there is a consistent mapping across animals and the explanation for the reported differences are due to sampling differences like the authors suggest. Alternatively, there may not be consistent mappings across animals. For example, in Patel et. al. 2010 (ref. # 8) there seems to be 1 of the 4 imaged LIPs with altered mappings (Fig 3, monkey Y, right hemisphere). We agree that the A-P gradient appears to be weaker in Monkey L. As requested above, we performed the statistical analysis to formalize our claim and there is a highly significant linear relationship between coronal plane and peak preferred direction. We have included discussion of this relationship.

“For both monkeys, we found a statistically significant relationship between anterior-posterior location and preferred direction ($p < 1e-40$) where more anterior planes had more voxels tuned for downwards directions while posterior planes had more voxels tuned for upwards directions.” Discussion; Lines 351-354

Minor concerns

1. Abstract:

a. Line 28-29: “A rough topography emerged where anterior LIP represented more contralateral and upward movements and posterior LIP represented more contralateral downward movements” – This is opposite of the reported anterior-posterior gradient. We have fixed this mistake.

“A rough topography emerged where anterior LIP represented more contralateral downward movements and posterior LIP represented more contralateral upward movements.” Abstract; Line 28-30

2. Introduction:

a. It may be worth noting some modern electrophysiology methods designed to access mesoscale organization: Dotson, Hoffman, Goodell & Gray, Neuron, 2017 and Tiechert et. Al. bioRxiv 2024 (<https://doi.org/10.1101/2024.05.13.593946>). Although these have less continuous spatial sampling they have better temporal resolution. Though maybe this would be a discussion point.

Thank you for bringing these papers to our attention. We have included these papers in both the introduction and discussion now.

“Recent advances in distributed microelectrode arrays, such as the Gray Matter array¹⁹ or MePhys system²⁰, now allow simultaneous multi-region recordings across primate hemispheres. While these distributed microelectrode array platforms achieve unprecedented temporal resolution (<1 ms) and volumetric coverage, the spacing between electrode shafts (2–5 mm) remains an order of magnitude coarser than the ~100–500 μm functional organization observed in cortical columns and microcircuits²¹. This spacing fundamentally limits continuous spatial sampling of fine-grained population activity patterns. These trade-offs in existing recording techniques highlight the need for a sensitive technique to bridge the gap in spatial resolution between microscopic (e.g., single neurons) and macroscopic (e.g., whole brain) views of the primate cortex.”
Introduction; Lines 69-78

“Recently, there have been development of new electrophysiology methods^{18,19} designed to access mesoscale organization that would aid greatly in understanding the relationship between the mesoscopic neurovascular populations identified in this paper and the underlying neuronal activity of neurons distributed across the PPC.” Discussion; Line 632-635

b. Fig. 1 Panel C: Are the spaces between the planes just for visualization or does fUSI have large gaps between planes? Is this related to the 1.66 mm between transducer slots?

This question I think arises from the earlier confusion where you thought we record all planes simultaneously (3D fUSI) instead of only being able to record from a single plane during a given session (2D fUSI). The 1.66 (or 5/3) mm spacing between recording slots was designed as a compromise between complete coverage of the PPC and small distance between planes. The spaces between planes in Fig 1C represents these different planes that we recorded from across several years. We added a sentence to the main text and methods to help clarify this point.

“During each session (Table S1), we recorded from a single coronal plane..” Line 112-113

“During each recording session (Table S1), we placed the ultrasound transducer on the dura with sterile ultrasound gel and acquired images from a single imaging plane.” Line 733-734

3. Methods:

a. Memory guided saccade task:

i. What was the size of the fixation dot, the target, the fixation window, and the target window?

We have added this information to the methods.

“Monkeys performed a memory-guided saccade task (Fig. 2A) where they fixated on a center dot of radius 2° (fixation state), maintained fixation while a peripheral cue (2° radius) was flashed for 400 ms ... During each task state, the monkey had to keep their

eyes within 6° of the fixation or target, i.e., the tolerance window was 6°.” Methods; Line 704-711

ii. Did the visual cue always appear in the middle of the fixation/memory phase? For example, if the fix/memory phase for 1 trial was 5 seconds would it appear at 2.5 seconds and if in another trial the fix/memory period was 4 seconds the cue would appear at 2 seconds? Or was there some other distribution of cue presentation times? We have clarified this point in the methods. The cue always appears right after the fixation period. The cue precedes the memory phase. This was not necessarily in the exact middle since the fixation and memory periods were independently variable in length per trial.

“For each task state, the duration was randomly drawn from the uniform distribution, meaning that the fixation and memory lengths might be different in the same trial, e.g., fixation length of 4.15 seconds and memory length of 3.9 seconds for one trial in Monkey L.” Methods; Line 717-719

iii. Why were the intertrial intervals different between the two monkeys – was there something intrinsically different about monkey P’s vasculature that they needed 2-4 more seconds between trials? Could the longer intertrial interval contribute to the better performance of decoding in Monkey P?

The change in intertrial interval (ITI) between monkeys was for behavioral training purposes. Monkey L did not tolerate as long of recording sessions and would also get frustrated with longer ITIs. To handle this, we used the shorter ITI. In Monkey P, he got slightly larger amounts of juice per trial to maintain his motivation, so we lengthened the ITI so that he would not become satiated with juice as fast and work for more trials. In our analyses, we never saw a previous trial’s activity influence the next trial. Although not formally tested since it would require collecting new data, we do not think that the longer ITI directly contributes to better performance in Monkey P.

b. General Linear Model (GLM)

i. I’m unfamiliar with vasculature-based methodologies, but convolution of a time-series can lead to leakage of signal into earlier bins. Is there potential for movement related signal to leak into the memory time bin? I think this is of little concern given the sluggishness of the blood volume responses.

Thank you for raising this point. There is no potential for the movement related signal to leak into early bins during the convolution of our task with a canonical hemodynamic response function (HRF; **Reviewer Figure 2**). We use a causal HRF with an absolute delay term. As a result, no activity can precede the start of the designated task state. In other words, the convolution can cause a temporal blur into the future, but not leakage into previous time bins.

Reviewer Figure 2 – Hemodynamic response function used for GLM

c. Preferred Direction

i. What was the time-bin used at the end of the memory period? Was it the last second of the memory period?

We have clarified this point in the main text and methods.

“To quantify the example regions’ tuning, we averaged the response to each of the directions in the last second of the memory period across all the voxels in each ROI to create regional tuning curves (Fig. 3C).” Results; Method 139-141

“For each voxel, we first calculated the Cohen’s d measure of effect size by comparing the response in the last second of the memory period (single timepoint within ± 0.5 seconds of memory end) to the baseline (-1 to 1 seconds relative to cue onset).” Methods; Lines 774-777

d. Within-Session Decoding

i. Line 618 Typo: fUS > fUSI

We have fixed this typo.

ii. How many repeats of the leave-one-out validation were done? Or was every trial left out once?

Every trial was left out once. We did not add any extra details to the methods about this since the standard definition of a leave-one-out cross-validation is to leave each trial out exactly once.

4. Results:

a. General

i. Line 96-97: ‘high spatial resolution 100-micron x 100 micron in-plane’ – I found this to be a bit misleading. It is important to know the voxels used in all analysis will be done on are 100x100x400 microns. Especially for interpretation of voxel-by-voxel heat maps and anterior-posterior gradients.

We have clarified this point.

“We used a miniaturized linear ultrasound transducer array capable of high spatial resolution (100 μm x 100 μm in-plane) and a large field of view (12.8 mm width, 128

elements, 100 μm spatial pitch, 16 mm depth penetration, 400 μm plane thickness)^{16,17}, i.e., each voxel is approximately 100 x 100 x 400 μm .” Line 108-110.

ii. Fig. 2a: The dashed circle isn't defined in the legend. I assume it is the target window. We see how this was confusing. We have revised the figure to just show the possible target locations around the fixation point. The dashed circle and the lines had no meaning. They were there to show that the targets were symmetrically distributed around a circle.

From Fig. 2

iii. Fig. 2c: Medial Parietal (MP) is not defined in the figure legend. We have added definitions for LIP, VIP, MIP, and MP to the figure legend.

“LIP: lateral intraparietal area; VIP: ventral intraparietal area; MIP: medial intraparietal area; MP: medial parietal area; Is: lateral sulcus; cis: cingulate sulcus; ips: intraparietal sulcus.” Figure 2 Legend; Lines 127-129

b. Figure 3:

i. Panel B/E: Does ‘Move’ correspond to the go cue or the actual start of the saccade? Move corresponds to the end of the memory period, i.e. the “go cue” when the center fixation point disappears. Here are the cumulative distribution function plots for time between the go cue and approximate start of saccade, showing that there is approximately 0.5 second delay between end of memory period and start of actual saccade (**Reviewer Figure 1**). We included information about this in the legend of Figure 2.

“Example session median [IQR] saccade latency Monkey P-399 [363, 430] ms, Monkey L-530 [500, 569] ms.” Figure 2 Legend; Line 122-123

Reviewer Figure 1 – Latency between move cue and saccade execution

ii. Which time bin is used for the analysis? From the methods I thought it was the last bin of the memory phase, but “analysis” appears aligned with “Move” on the x-axis. It would be nice to have as shaded area of the plot for the analysis bin (like for the baseline).

You are correct. The last bin of the memory phase is what was used for the analysis. This is what is already shown on the plot. The baseline period shows the shaded area for 3 timepoints. The analysis line shows that we used a single timebin. To better clarify this point, we added width to the analysis period showing that the actual timebin width is all fUSI timepoints landing within 1 second of the end of the memory period.

Panel b from Fig. 3. Shows more accurate representation of length of analysis period.

iii. I found it interesting there was some ROIs that had much higher baseline volume. It made me think about the CBV response – are there non-linearities in the CBV response? Are there saturations if there’s a significantly larger visual response to a saccade target that could occlude an additional movement response?

These are interesting questions but we think they are out of scope for this current paper. There likely are non-linearities in the CBV response. Multiple papers have begun to

explore the CBV response as measured by fUSI (Claron et al. 2023, Nunez-Elizalde et al. 2022, Aydin et al. 2020). Our task was not designed to tease apart possible non-linearities in the CBV response. Since we think your question is related to separating visual from motor components in the CBV response, this would likely require a task with only visual, only motor, and combined trials.

Assessing signal saturation is also difficult to assess. In our experiments, we never witnessed signal saturation on the hardware side. We do not think it is currently known what the full dynamic range of CBV is, and how this depends on the diameter of the mesovasculature being measured. In other words, the dynamic range seen for CBV almost certainly depends on diameter of the blood vessels. It is possible that a significant large visual response might occlude additional movement response, but we do not have a method to assess for this in the current dataset. However, we think it is improbable because in Fig. 3 and Fig. S3, we see that the CBV continues to rise until several seconds after the movement period, suggesting that we are not saturating the signal. One future method to better answer your question might be to incorporate memory-guided saccades without visual components and visual-only trials without movements, but this is a difficult task to train monkeys to perform and due to the task complexity would still be difficult to interpret the results.

c. How consistent is the directional tuning within a session?

i. I'm interested in the components that explain the most variance of the CBV response across the PPC. How many components are needed to reach 95% variance? Also do the top PCs correspond to direction tuning? A quick way to look at this would be to correlate each voxel's directional tuning strength to their weight onto each component. It would be interesting to know if saccade direction explained a large amount of variance across all components or specific components.

The specific number of components needed to reach 95% variance depends on the specific session, but was typically approximately 330-450. We have added this information to the paper and included a supplemental figure showing how the number of components varied with trial time and validation fold.

Fig. S12 – Number of components to capture 95% variance in Principal Component Analysis for each example session

For each example session, plot shows the number of components kept to capture 95% of the variance in the training data. Green region represents the range of components kept for each fold in a 10-fold cross-validation. Black shows mean components kept. Per methods in main paper, at each timepoint after the cue, we used all previous timepoints after the cue in the trial. For example, to test our ability to decode at 2 seconds after cue onset, we concatenated the data from 0, 1, and 2 seconds after the cue. We treated these timepoints as additional features in the decoder model. In other words, the input to our decoder model had $N \times T$ features, where N is the number of voxels in a single Power Doppler image and T is the number of timepoints.”
Supplemental Figures Lines 75-83

“For the PCA, we kept 95% of the variance. The number of components kept depended on the training dataset, typically varying between 330 – 450 components (Fig. S12).”
Methods; Lines 820-821

We have included three figures here that show you some characteristics of the PCA performed as part of the decoding analysis. In Reviewer Fig. 3, we plot the analysis you requested for the example session in Monkey P. As you can see, there is no clear relationship between the top PCs and directional tuning strength or statistical significance of the tuning. We also show you the principal component coefficients of the decoding analyses for two example sessions (Reviewer Fig. 4, 5). Although the first 20 principal components do appear to differentiate some of the mesovasculature, there was no clear relationship to task variables.

In this paper, we used PCA as a method to reduce redundant dimensionality to help with the decoder performance. We would thus like to not include these figures in the main or supplemental figures. We feel that these figures distract from the main paper since we do not currently make claims about relationships between the top principal components

and task variables in our paper and further analysis of PCs for relationship to task variables would require several additional figures that may distract readers.

Reviewer Figure 3– Correlation between PCA coefficients and GLM metrics

A. Voxel-wise correlation between PCA coefficients and tuning strength (Cohen's d). Each dot represents one voxel's directional tuning strength (from center-of-mass calculation) plotted against its weight onto each component.

B. Voxel-wise correlation between PCA coefficients and tuning significance (F-score). Each dot represents one voxel's F-score from GLM analysis strength plotted against its weight onto each component.

Reviewer Figure 4 – Coefficients of first N principal components for Monkey L example session

Each row represents one component. Each column represents one timepoint. At each timepoint after the cue, we used all previous timepoints after the cue in the trial. For example, to test our ability to decode at 2 seconds after cue onset, we concatenated the data from 0, 1, and 2 seconds after the cue. We treated these timepoints as additional features in the decoder model. In other words, the input to our decoder model had $N \times T$ features, where N is the number of voxels in a single Power Doppler image and T is the number of timepoints. This training input is what defines the PCA transform. For this figure, we took the final timepoint in the trial (i.e., used all timepoints from cue start to trial end).

Reviewer Figure 5– Coefficients of first N principal components for Monkey P example session

Each row represents one component. Each column represents one timepoint. At each timepoint after the cue, we used all previous timepoints after the cue in the trial. For example, to test our ability to decode at 2 seconds after cue onset, we concatenated the data from 0, 1, and 2 seconds after the cue. We treated these timepoints as additional features in the decoder model. In other words, the input to our decoder model had $N \times T$ features, where N is the number of voxels in a single Power Doppler image and T is the number of timepoints. This training input is what defines the PCA transform. For this figure, we took the final timepoint in the trial (i.e., used all timepoints from cue start to trial end).

ii. Although potentially out of scope for this study, an interesting future direction for this data would be to use a continuous decoder to see if fUSI could predict single-trial variance in movement direction (E.g. decoded endpoint error correlates with actual saccade endpoint error on single trials).

We agree this is a very interesting question and also agree it is out of scope for this particular study for one main reason. We used a standard saccade distance (20° eccentricity) so although there is small variation in the saccade endpoint, this variability is small relative to the saccade magnitude and is insufficient to train a continuous decoder. It would require a different task design. We have added this as a future direction to our discussion section.

“Furthermore, implementing continuous fUSI decoding (as opposed to predicting discrete directions or eccentricities) could extend our findings by examining whether trial-to-trial variations in mesoscopic LIP activity predicts endpoint error distributions in saccadic eye movements. This would enable investigation of how mesoscopic LIP populations encode movement precision in addition to categorical direction or eccentricity.” Discussion; Lines 608-612

iii. Line 179-181: “LIP contained the most informative...” – It may be a nice addition to this analysis to correlate the GLM weight for each voxel with the angular error from the searchlight analysis.

Thank you for the suggestion. This is similar to an analysis requested by Reviewer 1 where we showed the overlap between the GLM and searchlight analyses. This new analysis is shown in Fig. 3G, S2H, 7D. We also performed the specific analysis you requested for the example sessions and it is Fig. S3. As we can see, there is a high correlation between the GLM statistical significance (F-score) and the angular error for each voxel. This is discussed in the main text

“Fig. S3 – Correlation between GLM weight and searchlight accuracy for each example session

Voxel-wise relationship between GLM weights and searchlight accuracy. Different dot colors represent voxel-wise statistical significance for searchlight and/or GLM value ($q < 0.001$).” Lines 17-21

“Additionally, voxels in all of the example sessions displayed a strong and significant correlation between F-score (GLM analysis) and angular error (searchlight analysis) (Fig. S3).” Results; Lines 232-233

d. Figure 4:

i. ‘Decode movement direction’: With the sluggishness of the CBV response is it possible to differentiate the visual response to the saccade target versus the movement preparation for the saccade? Are you decoding the saccade target location or are you decoding the saccade direction?

Thank you for the question. We had designed our task to help differentiate the visual response from the movement preparation by having a cue that is short (400 ms) and a memory period that is long (>4 seconds). Although CBV is slow, the hemodynamic response function from previous primate fMRI papers is 1-2 seconds, so we should see a rise and then fall in the signal if we were solely seeing the visual response. Similarly, if we were solely seeing the movement preparation, then the signal would not rise until later in the memory period. We saw a combination of these two extremes, where the CBV would increase early after the visual stimuli and then stay elevated throughout the memory period suggesting that we have both an early visual response and a later movement preparation response. We have added this point to our discussion section.

“Visual versus motor planning – While our task design temporally separated visual cue processing and saccade preparation phases by having a cue that is short (400 ms) and a memory period that is long (>4 seconds), fUSI faces inherent constraints in fully dissociating these components. Neurovascular coupling introduces an ~2-4 second smoothing effect on CBV signals^{50–52}, creating a temporal overlap between the visual and motor components. We saw the CBV increase early after the visual stimuli and then stay elevated throughout the memory period suggesting that we may have both an early visual response and a sustained movement preparation response. Future experiments with modified tasks or simultaneous electrophysiology will be needed to fully disentangle the visual and motor components.” Discussion; Line 585-592

ii. I know the circle corresponds to the ‘center’ position which emerged from the multicoder approach, but it’s not listed anywhere in the main text or figure legend. It turns out to be a nice internal control that ‘center’ never seems to be decoded.

1. Is the chance listed in panel A and D $1/8$ (the directions) or $1/9$ (the directions + center)?

Thank you for the question. The chance listed is $1/8$. Although 9 classes are possible from the decoder architecture, due to the class imbalance (i.e. no center options in the training set), the empiric chance level is closer between $1/9$ and $1/8$. Because of this class imbalance in the multicoder architecture, we chose the more conservative

threshold of greater than 1/8 as our chance level. We have included a brief discussion of this point in the methods.

“Although the multi-coder architecture can generate 9 possible classes, due to class imbalances in the training sets, the empiric chance level of the decoder was between [1/8 1/9]. We therefore picked the more conservative threshold of 1/8 chance level for the binomial test.” Methods; line 833-835

iii. What time bin corresponds to the decoder for the confusion matrix?

We realized this was never clarified in the paper. For the confusion matrix, we used the performance during the last timepoint in the trial. We have clarified this point in the figure legend.

“B. Confusion matrix of decoding from last timepoint in trial.” Fig 4 legend; Line 266

iv. What time bin corresponds to the searchlight analysis? Additionally, it would be interesting to look at what timepoint the voxels are most informative.

We used the last timepoint in the trial for the searchlight analyses. We have also now included a supplemental figure that examines how the most informative voxels evolve across the trial (Fig. S4). We have included a brief discussion of this evolution in the main text (#add).

“C. Searchlight analysis at end of trial.” Fig. 4 legend, Line 268

“To understand how these searchlight results evolved across trial time, we repeated the searchlight analysis for each timepoint after fixation (Fig. S4). Similar to the decoder performance using the entire image, the searchlight plots began showing significant voxels within LIP within three seconds after fixation and then plateaued in number of significant voxels and decoder performance by the end of the movement period.” Results; line 238-242

Fig. S4 – Supplement to Fig 4; Searchlight results across decoding time windows

A. Example session in Monkey P (S20220112). Each tile shows searchlight analysis for that timepoint. All significant decoding ($p < 0.001$) shown. Per methods in main paper, at each timepoint after the cue, we used all previous timepoints after the cue in the trial. For example, to test our ability to decode at 2 seconds after cue onset, we concatenated the data from 0, 1, and 2 seconds after the cue. White circle – 200 μm searchlight radius. White line – 1 mm scalebar.

B. Example session in Monkey L (S20210624). Same format as **(A)**.

C. Example session in Monkey L (S20200921). Same format as **(A)**.

e. Are the mesoscopic populations stable across multiple days? And Figure 5:

i. It is impressive to get above chance decoding 900 days later.

We agree. We were pleasantly surprised to see how well it still worked.

ii. I found the explanation that the implant likely shifted and correlating image similarity with accuracy and angular error convincing.

Thank you.

iii. This section is when I started to realize that, especially for monkey L, there was a large amount variability across the sessions in the efficacy to decode saccade direction (even within a session). Is it because the monkey L implant missed part of the LIP – I assumed with the large volume accessible with fUSI this wouldn't be the case.

We never figured out why there was such high variability. We explored a number of factors including anatomical plane (which gets at your point about missing LIP), number of error trials in a given session as a proxy for motivation, day of the week, image stability, etc, but never found anything that explained the day-to-day variability in Monkey L.

We included several of these discussions in the paper already, such as examining anatomical plane and image stability.

“There was no statistical difference (1-way ANOVA, $\alpha=0.01$) between the percent correct or angular error depending on the plane (Fig. 7C). These results suggest that all anatomical LIP planes we sampled contained sufficient information to accurately decode at least eight intended movement directions on a single-trial basis.” Results; line 395-398

“In Monkey L, the performance was clumped into two temporal groups (before and after Day 900). Was this change in performance due to physical changes in the imaging plane or due to changes in subpopulation function? ... This supports our hypothesis that the decrease in decoder performance resulted from changes in the imaging plane rather than drift in each subpopulation's tuning.” Results; line 298-314

1. The angular error in monkey L seems to be averaging above 60 degrees (for the 0-day decoder – which I assume is the cross validated performance). I found this concerning that the example session in figure 4 does not seem that representative of monkey L's decoding performance.

Thank you for highlighting this point. We have added an additional example session in Monkey L to the supplemental figures (shown above in reviewer response). We would like to keep the new example session in the supplemental figures because we want the example session in the main text to be the same coronal plane as the majority of the rest of the main figures for logical flow of the story. As mentioned elsewhere in review, all of the code and data behind this paper will be archived online and interested readers can easily explore the variability across sessions.

f. How does the mesoscopic population tuning change across anterior to posterior portions of PPC?

i. Line 270-271: “respond most strongly to ipsilateral movements” – Was this an increase or decrease in the CBV?

Thank you for highlighting this sentence. We have clarified to mean an increase in CBV.

“some voxels that responded most strongly, i.e., a CBV increase, for ipsilateral movements.” Results; line 355

g. Figure 6:

i. Are the data on the beeswarm plot from only the session presented in panel A?

This is all the data pooled across all the sessions. I think this comment likely is related to your misunderstanding that we recorded from every plane simultaneously on a given day. On each day, we only could record one plane, so we pooled all the data across days for the across-plane analyses.

ii. What time point is used for these data?

The end of the memory period. We have clarified this point in the methods.

“For each coronal plane, we would concatenate all sessions recorded from that plane. We then performed the same methods for statistical analyses as described above. We examined the last timepoint when calculating the preferred tuning and other statistical measures of data distribution at each voxel.” Methods; Lines 791-794

h. Discussion:

i. Line 357-358: Like the explanation of the A-P gradient being different sampling across animals, the difference in contralateral space could be similar. If you could only access 3 mm relative to ear bars in monkey P with an electrode you may draw very different conclusions of the bias for contra- vs ipsilateral space in LIP.

We are interpreting this comment to mean that Platt and Glimcher may have found their results due to a low sampling of the anterior-posterior extent of LIP. Please correct us if this is the wrong interpretation of your comment. We are not comfortable adding a sentence to the paper discussing this point because there is insufficient data within the Platt and Glimcher 1988 paper to make this inference. Their histological study only shows the recording location of 11 of their 99 neurons and was from a 2-week recording period in only 1 animal so we do not know the true A-P extent of their recordings. Our sentence “The reasons for the apparent discrepancy with Platt and Glimcher 1998 remain unclear.” is an accurate statement and we feel it does not warrant further discussion.

ii. Line 372-373: With the sluggish response of the CBV I am reluctant to draw any conclusions on the separability of saccade preparation and eye movement.

I am unclear on what the comment is about. We do not make any claims near lines 372-373 about the separability of saccade preparation and eye movements. We were discussing that we saw no difference between LIPd and LIPv in our study, which would seem to agree with your comment. We also say that our study was not designed to separate effects of attention from oculomotor planning, which would also seem to agree with your comment. Can you please clarify your concern?

We have a similar discussion about separability of visual processing vs motor preparation. See our comment above about “visual vs motor planning”.

iii. Line 457: ‘record anterior-posterior synchronously’ – This seems to be the ideal way to tackle the anterior-posterior gradient question.

We agree but unfortunately it was not possible with the existing ultrasound transducer due to the size of the recording chamber and its positioning relative to the LIP. This will have to be left for future studies sadly.

REVIEWER COMMENTS

Reviewer #1 (Remarks to the Author):

I believe the authors have adequately addressed my concerns. No further questions.

Thank you for the earlier comments and taking the time to review our paper.

Reviewer #2 (Remarks to the Author):

While the authors did a great job in addressing my previous comments, I would appreciate if the authors could address the following:

1. Please ensure that axes are of similar scale, especially when meant for comparison purposes (e.g., Fig. 3, C and F; Fig. 5, C and F – please also add in an axis break from 0 to the next relevant label; Fig. 6C).

Thank you for comment. This was an oversight on our part. We have now ensured the axes are a similar scale within and between figures whenever possible. Specifically, we standardized the **y**-axes in:

- Fig 3B, 3C, 3E, 3F, S2B, and S2C
- Fig. 5C, 5F
- Fig. S8C, F
- Fig. 6C

We standardized the **x**-axes in Fig. S8C, F.

We elected to not standardize the x-axes in Fig. 5C, F, and S7 due to the vastly different date ranges covered (~100 vs ~900 days). We can standardize this if the reviewer would like but we think it would make it more difficult for readers to appreciate the spread of the data for Fig. 5C.

We additionally standardized the colorbar range in Fig 5B, E and in Fig. S8B, E. We elected to not standardize the colorbar range in other figures (Fig 3A, D for example) since a standardized colorbar range would compress the informative colors to a narrower range of colors and make it more difficult for the reader to ascertain useful details.

We are unclear on where the reviewer wants us to add an axis break for Fig. 6C. For the y-axis, an axis break is unnecessary since it is a linear scale from 0 to 0.01. For the x-axis, it is continuous from -180 to 180. We are happy to add an axis break once we understand where the reviewer wants it.

2. Lines 132-151: Could the authors clarify what they mean when referring to ROIs – are these designated mesoscopically-defined PPC regions or are the authors referring to the ROIs from lines 103-104? If the latter, could the authors provide an overall assessment of directional tuning across PPC and then, break this down for the different PPC regions mentioned on lines 103-104? It would be worthwhile knowing the representation of directionality across these defined PPC regions.

We see how this was confusing. We have added a new analysis (Fig. S1) to aid with answering this comment. Previously we had examined LIP vs non-LIP because there was minimal activation outside of the LIP. We have now defined the other subregions (e.g., Area 7, VIP, MIP, Area 5, and MP) and quantified the percent of voxels within each PPC region that display directional tuning. We have revised the paragraph to describe the findings of this revised analysis. We start with an overall assessment of directional tuning across PPC regions (Fig. S1) and then explicitly define what we mean by “ROI”.

“[R]egions outside of the LIP, such as VIP, MIP, Area, 5, Area 7, and MP, contained fewer voxels with directional tuning (<10% of voxels within a given region) (Fig. S1). All region boundaries were estimated based upon a histological rhesus monkey atlas²⁷. Within LIP, we observed heterogenous response patterns in different subpopulations, where we define “subpopulation” as a group of one or more adjacent voxels with highly similar responses to the different directions, i.e., highly similar tuning properties... To highlight this diversity of directional responses, we defined several example 400 x 400 μm regions-of-interest (ROIs) within LIP and MIP (Fig. 3A-white boxes, Fig. 3B) ... Regions outside of the LIP, such as VIP, MIP, and Area 7, contained much fewer voxels with directional tuning (<10% of voxels within a given region).” Results; lines 135-154

3. Moreover, in this section, the authors jump from stating results for the whole area (and voxels within) to highlighting specific LIP results. Could the authors please describe the results across all regions mentioned before or indicate why/when they are mentioning specific regions in the corresponding results subsection?

Similar to the response above, we have now gone through the paper and added clarity about the scope of the findings (PPC overall vs LIP), with a focus on the Results section. Since most of our results show our entire field of view inclusive of many PPC regions or were based upon the entire field-of-view, we have kept it generalized to PPC when our results were not specific to LIP. We have added the word “LIP” in multiple locations to clarify when the analyses and results are more restricted. When we observed findings in other PPC subregions, we explicitly mention those results and the

pertinent PPC region. We include several examples below of places we improved clarity.

*“However, this high uniform magnitude observed across the tuned voxels **in LIP** suggests a rapid transition in tuning between adjacent voxel patches and supports a patchy topography rather than a smooth gradient in tuning between neighboring voxel patches”* Results; Line 179-180

*“**In LIP**, there were closely neighboring voxel patches with different tuning curves (Fig. S3B, C).”* Results; Line 191-192

*“we extracted the directional-modulated voxels **within LIP** and created a beeswarm chart containing each voxel’s directional preference”* Results, Lines 356-358

*“more anterior planes had more **LIP** voxels tuned for downwards directions while posterior planes had more **LIP** voxels tuned for upwards directions. Most **LIP** voxels encoded for contralateral movements (-90° to +90°) although there were some voxels that responded most strongly, i.e., a CBV increase, for ipsilateral movements.”* Results; Lines 361-362

*“These results suggest that all anatomical **PPC** planes we sampled contained sufficient information to accurately decode at least eight intended movement directions on a single-trial basis.”* Results; Lines 390-392

a. For example, lines 240-242 mention decoding results for LIP, but not the other regions. Does this mean only significant decoding performance was found in LIP and none of the other regions?

We have revised this section to describe the searchlight decoding results in other regions.

*“In both animals, voxels in regions outside of LIP were within the 10% most significant voxels (**Fig. 4C, F**). In Monkey P, these regions included Area 7 (14% of the most significant voxels), MIP (11%), VIP (5%), and MP (2%). In Monkey, these regions included MIP (8%), VIP (5%), and Area 7 (4%).”* Results; Lines 239-242

“Like the decoder performance using the entire image, the searchlight plots began showing significant voxels within LIP within three seconds after fixation. Significant

voxels appeared within other regions, including Area 7, VIP, MIP, and MP, in later trial timepoints. The number of significant voxels and decoder performance plateaued by the end of the movement period.” Results; Lines 246-249

b. In keeping with this idea, could the authors state what they define as ‘PPC subpopulations’ (lines 274-277)?

We have added a definition for ‘subpopulation’. We now define subpopulation to mean “a group of one or more adjacent voxels with highly similar responses to the different directions, i.e., highly similar tuning properties”. We do not have a rigorous, quantitative way to define the boundaries of different subpopulations because the responses observed were so variable so this definition is unfortunately a subjective definition. We believe that this definition helps clarify our usage of the word ‘subpopulation’ regardless.

“[W]e define ‘subpopulation’ as a group of one or more adjacent voxels with highly similar responses to the different directions, i.e., highly similar tuning properties.”
Results; Lines 140-141

4. For the consistency in directional tuning results section, could the authors explain or comment on the reasons for decoding on entire fUSI data and not on anatomically-subdivided data? Could there be inter-region differences that could skew the results one way or another?

Thank you for this question. We have added two sentences to the methods explaining our motivation for decoding on the entire fUSI data rather than anatomically-subdivided data. In essence, we felt that decoding on the entire dataset introduced less bias than imposing our approximations of region boundaries. Without detailed post-mortem histology, we cannot confidently know the specific boundaries of each anatomical region.

With regards to your question about inter-region differences that could skew the results one way or another, we do not believe this is a significant concern when using the entire image. The whole-image decoder using PCA+LDA finds the most informative features to decode intended movement direction. By including the entire image, we do not actively bias our decoder to rely upon specific regions but rather lets the data itself tell us the most important features. The PCA-LDA weight visualization (Fig. S5) shows these features. Additionally, the searchlight analysis, which uses a small 400- μm ‘searchlight’, shows that different 400- μm circular subregions do have different ability to decode

intended movement direction. With the archived data and code, it is possible for a reader to explore any subregions they want with the decoder, for example exploring the decoder performance when only including voxels from Area 7.

“We used the entire image rather than decoding from specific anatomically-defined subregions to avoid introducing any bias by our approximation of anatomical boundaries. By including the entire image, we did not actively bias the decoder to rely upon specific regions but rather let the data itself tell us the most important features.”
Methods; Lines 836-840

5. This is merely a comment – it is notable that the authors chose to check the consistency of the images as a later step instead of using this first to ensure consistent anatomical mapping across sessions and then, perform further analyses.

Thank you for the comment. To clarify the process a little bit, we did qualitatively check the anatomical similarity before each session to maximize the similarity by eye. Due to how the transducer is attached to the recording chamber and how the acquisition software worked, it was very difficult to finetune the transducer positioning in real-time and gross similarity by eye was the best we could obtain for given recording sessions. As we analyzed the variability of decoding performance across sessions, we implemented the CW-SSIM measure to detect more subtle differences and have a quantitative measure of image similarity rather than qualitative similarity. With advances in recording software since the acquisition of this paper’s data, it would now be possible to implement the quantitative image similarity metric into a real-time app that helps the user maximize image similarity before each recording session.

6. Lines 374-387: Why do the authors discuss dorsal and ventral aspects of LIP in a section addressing the anterior-posterior axis? Could the authors clarify this?

Thank you for pointing this out. This was poor organization on our part. We have now moved that paragraph to be under its own subsection.

“Do dorsal and ventral LIP display different tuning properties?”

Dorsal (LIPd) and ventral (LIPv) LIP are believed to have different functions with LIPd being involved in visual processing while LIPv is involved in both attentional and visual-saccadic processing^{31,32} Results; Lines 399-413

7. Lastly, the organization of the Discussion seems a bit disjointed – could the authors follow (as closely as is appropriate) the layout of the Results section? A possible order could be: i) fUSI-specific features, ii) stable decoding within sessions and across time, iii) anatomical considerations (A-P, D-V), and iv) applications in brain-machine interfacing, with limitations.

Thank you for pointing out that our organization felt confusing to the reader. We have now reorganized our discussion similar to the schema you suggested:

1. fUSI-specific features
 - a. Sensitivity of fUSI
 - b. Mesoscopic functional imaging – balancing spatial resolution and field-of-view
2. Stability of directional tuning across time
3. Preferences for contralateral space
4. Anatomical considerations
 - a. Anterior-posterior gradient
 - b. Differences between dorsal and ventral LIP
 - c. Directional saccadic activity outside of LIP
5. Applications to ultrasonic brain-machine interfaces
6. Limitations of study
7. Future studies
8. Conclusion

We kept limitations and future studies at the end rather than integrating it into each Discussion section because we felt that it was easiest for the reader to see all of the limitations together followed by our suggestions for future studies to help address these limitations but also the new questions raised by our paper.

Reviewer #3 (Remarks to the Author):

I appreciate the author's response to my comments and additions to the paper. In particular the explanation of cluster-based inference and corrections to the literature were insightful. The additional analyses strengthen the claims of the paper and improve clarity.

Thank you for the earlier comments and taking the time to review our paper.

Reviewer #3 (Remarks on code availability):

GitHub README and data organization are easy to follow. Code is easy to set up and run. 'PercentChangeColormap.mat' file seems to be missing.

Thank you for trying out our code. We identified the error that led to that file (and a few others) being missing from the online repo. We had an overly broad .gitignore file and have now fixed the bug that led to the colormaps not being present in the GitHub repository. We pushed the changes to the GitHub repository and will archive the newest version to Zenodo after all comments have been addressed in case of other necessary changes.

Reviewer #4 (Remarks to the Author):

The authors sufficiently addressed all my concerns. I list a few minor comments they may or may not want to address:

1. Line 140-141 “Last second of the memory period”: To me this phrase indicates the last 1 second of timepoints within the memory period and would have no time points in the move period. The phrase used in the rebuttal seems a better description of the actual time bin “timepoints landing within 1 second of the end of the memory period” or using the parenthetical from the methods “(single timepoint within +/-0.5 seconds of memory end)”

We see how this is confusing. We have revised the language as you suggested for clarity.

“last second of the memory period (single timepoint within +/-0.5 seconds of memory end)” Results; Lines 146-147

2. Line 245 “fUSI images (Fig. 4A, B, D, E): Not sure why these figures are referenced here.

We see how this is confusing. We had previously had the references to Fig. 4 to reflect the whole-image fUSI decoding results, but we see how this is confusing and not necessary. We have removed this reference and put the appropriate figure reference.

“weights of the PCA-LDA decoder trained on full fUSI images (Fig. S6)” Results; Line 252

3. Line 500 “We do not know why we see activity”: Could it be visual responsiveness to the cue?

Thank you for raising this point. We agree that it could be visual responsiveness. We have revised the language accordingly.

“We also observed activity within the deeper layers of posterior-ventral MIP in both monkeys, agreeing with previous literature that found visual and saccade-related activity within MIP^{46,47}.” Discussion; Lines 543-545

4. The F-Score color scale bars across Fig. S1, S2D, S9 are not consistent (seems opposite in S1 compared to the other two).

Thank you for finding this error. This was an error in S1 that I had already fixed in the main figure file but somehow escaped making it into the submitted version. We have now ensured it is fixed in the submitted version.

5. Fig S9 has Lothar and Pumu instead of Monkey L and Monkey P.

Thank you for identifying this error. We have now corrected the figure to say Monkey L and P instead.

6. Could the variability in Monkey L be due to a difference in fixational eye movements compared to monkey P? Maybe this will be left to the interested reader.

This is an excellent question. This is not something we analyzed in our dataset but it is something that could be examined in the future. We have added a short paragraph to the manuscript discussing this variability and how it might be related to fixational eye movements.

“In Monkey L, we observed high variability in decoder performance across sessions (Fig. 7). We explored several factors including anatomical plane, number of error trials in a given session as a proxy for motivation, day of the week, and amount of brain movement within a session, but never identified any factors that explained the day-to-day variability in Monkey L. Future work will be needed to identify the causes of this across session variability, such as whether differences in fixational eye movements

between the two monkeys might explain the decoder variability.” Discussion; Lines 591-596